# Sign Lock-In: Randomly Initialized Weight Signs Persist and Bottleneck Sub-Bit Model Compression

**Akira Sakai** [1] [2]    **Yuma Ichikawa** [1] [3]

## Abstract

Sub-bit model compression targets storage below one bit per weight; as magnitudes are aggressively compressed, the sign bit becomes a fixed-cost bottleneck. Across Transformers, CNNs, and MLPs, learned sign matrices resist low-rank approximation and are spectrally indistinguishable from an i.i.d. Rademacher baseline. This randomness gives rise to the lower bound of sub-bit model compression—the one-bit wall. Despite this apparent randomness, most weights retain their initialization signs; flips primarily occur via rare near-zero boundary crossings, **suggesting that sign-pattern randomness is largely inherited from initialization.** We formalize this behavior with *sign lock-in theory*, a stopping-time analysis of sign flips under SGD noise. Under bounded updates and a rare re-entry condition into a small neighborhood of zero, the number of effective sign flips exhibits a geometric tail. Building on this mechanism, we introduce a *from-scratch* low-rank sign-template training method that prevents the emergence of this one-bit wall.

## 1. Introduction

The sign is the minimal discrete attribute of a real-valued weight; it maps $w \in \mathbb{R}$ to a binary state $\text{sign}(w) \in \{\pm 1\}$, carrying one bit of information per scalar weight. Historically, most practical compression pipelines focused on the few-bit regime, where the sign bit constituted a small and nearly constant overhead relative to magnitude storage and, therefore, rarely emerged as a bottleneck. Our study demonstrates that the sub-bit regime is qualitatively distinct. As indicated in Figure 1, once magnitudes are compressed to approximately one bit per weight, the remaining sign be-

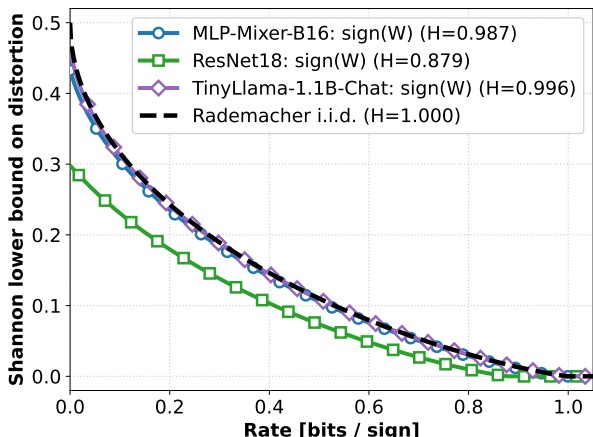

*Figure 1.* **One-bit wall.** Shannon's rate-distortion lower bound for binary sign patterns under Hamming distortion evaluated using an entropy-rate proxy estimated from pretrained weights. Across all models, this proxy is close to one, and the bound is nearly indistinguishable from that of an i.i.d. Rademacher baseline, indicating that sign patterns contain little redundancy.

comes a fixed-cost barrier referred to as the one-bit wall. Furthermore, learned sign patterns across architectures are close to i.i.d. Rademacher; they are nearly uniform and only weakly correlated, leaving little redundancy for further compression. However, tracking sign dynamics has revealed that this apparent randomness is largely inherited from initial random weight signs. This empirical picture points to a paradoxical dynamical regime: although the *marginal* distribution of trained signs is nearly indistinguishable from i.i.d. Rademacher noise, the *trajectory* of each sign is highly persistent throughout training.

Our approach follows the stochastic-process viewpoint for analyzing SGD beyond asymptotic linearization, aiming to explain this *sign lock-in phenomenon*. When noise drives the dynamics through rare events such as boundary hits and escapes, tracking only the mean flow can miss the key mechanism; localization via stopping times is therefore a central but challenging approach. This stopping-time perspective connects several classical frameworks: the ODE method controls stability by stopping-time localization (Kushner & Yin, 2003; Benaïm, 1999), diffusion approximations turn

[1]Fujitsu Limited [2]Tokai University [3]Riken Center for AIP. Correspondence to: Akira Sakai <akira.sakai@fujitsu.com>, Yuma Ichikawa <ichikawa.yuma@fujitsu.com>.

boundary crossings into first-passage problems (Li et al., 2017), and Freidlin–Wentzell theory explains exponentially rare boundary hits via metastable exit times (Freidlin & Wentzell, 1998). Recent ML work adopts SDE/Markov-process lenses to study hitting-time-like behavior under realistic schedules and noise levels (Mandt et al., 2017). Following this line, we formalize sign dynamics under schedule-aware SGD as an effective minimal theory built around stopping times.

**Contributions.** The main contributions are as follows:

- **Empirical discovery**. The learned weight signs are much harder to compress than magnitudes in various representative pretrained architectures. In particular, sign matrices exhibit low-rank approximation error decay and behave like an i.i.d. Rademacher baseline. Moreover, sign patterns remain largely inherited from initialization throughout training. The practical implications of this phenomenon are formalized in the sub-bit regime, which is referred to as a one-bit wall.

- **Sign lock-in theory.** An excursion-based effective framework is introduced to characterize sign dynamics. Under two verifiable conditions, a *geometric* tail law is proved for the effective outer-to-outer sign flip count, which provides a mechanistic explanation for the persistence of noise-like signs in experiments. This theory is also numerically validated.

- **Lock-in enhancement.** Building on the theory, we propose a from-scratch low-rank sign-template method: a re-generable template $T = \text{sign}(GH^\top)$ is selected before training, and training is biased to preserve it. Gap initialization and early-phase outer-drift regularization reduce boundary visits and re-entry, preserving the structured sign template during training.

Related work is discussed in Appendix B.

## 2. One-Bit Wall of Model Compression

We empirically find that learned weight signs are particularly difficult to compress. Across diverse pretrained architectures (MLP, CNN, Transformer) and layers, the sign component exhibits (i) weak low-rank compressibility, (ii) spectral statistics close to i.i.d. Rademacher noise, and (iii) strong persistence during training. In contrast, the magnitude component is significantly more compressible. Figure 2(a–c) summarizes these signatures. Notation is given in Appendix A. The experimental settings and additional analyses can be found in Appendix C.1–C.3.

Modern compression pipelines leverage structure learned during training. For signs, our evidence indicates that training does not yield exploitable structure: the trained pattern remains close to an initialization-level random template, and optimization seldom modifies it.

### 2.1. Phenomenon: Signs Look Like Noise Yet Persist

We investigate both *compressibility* and *randomness* of learned weight signs. Our evaluation includes representative pretrained models: MLP (MLP-Mixer-B16), CNN (ResNet18), and Transformer (TinyLlama-1.1B-Chat). To investigate training-time dynamics, we track sign drift in a scratch-trained multi-layer Transformer language model designed for next-token prediction.

**Sign-magnitude decomposition.** Let $W \in \mathbb{R}^{m \times n}$ denote a weight matrix and define the components of sign and magnitude as follows:

$$ S := \text{sign}(W) \in \{\pm 1\}^{m \times n}, \ \ A := |W| \in \mathbb{R}_{\geq 0}^{m \times n}, $$

so that $W = S \odot A$. This isolates the discrete sign pattern $S$ from the nonnegative magnitudes $A$ and allows us to test their structure separately.

**Compressibility probe: low-rank approximation error.** To assess low-rank compressibility, the optimal rank-$r$ approximation error in the Frobenius norm is measured. For a matrix $M$, let

$$ M_r \in \underset{\text{rank}(X) \leq r}{\text{argmin}} \|M - X\|_F $$

denote the best rank-$r$ approximation, given by truncated SVD, and define $E_r(M) := \|M - M_r\|_F / \|M\|_F$. Since layers have different shapes, set $d := \min(m, n)$ and parameterize $r$ using the rank ratio $q := r/d \in (0, 1]$ with $r = \lfloor qd \rfloor$. Across architectures, $E_r(S)$ decays substantially slower than $E_r(A)$ at matched $q$, and the raw matrix error $E_r(W)$ tracks the sign-side behavior more closely than the magnitude-side behavior. Thus, the sign matrix is far more resistant to low-rank compression than the magnitudes and largely explains why direct low-rank approximation of $W$ is difficult.

**Randomness probe.** Low-rank error alone does not distinguish structured matrices from those that behave like random noise. To test whether $S$ is spectrally consistent with a random baseline, we sample multiple $s \times s$ submatrices $S_{\text{sub}}$ from $S$, compute their singular values, and normalize by $\sqrt{s}$ to remove trivial scaling. As a baseline, we use i.i.d. Rademacher matrices $R \in \{\pm 1\}^{s \times s}$. Let $F_{\text{sign}}$ and $F_{\text{rand}}$ denote the pooled empirical cumulative distribution functions (ECDFs) of the normalized singular values from $S_{\text{sub}}$ and $R$, respectively, and measure the discrepancy using the two-sample Kolmogorov–Smirnov statistic $D := \sup_x |F_{\text{sign}}(x) - F_{\text{rand}}(x)|$. We observe small KS distances $D$ across layers: the singular value statistics of

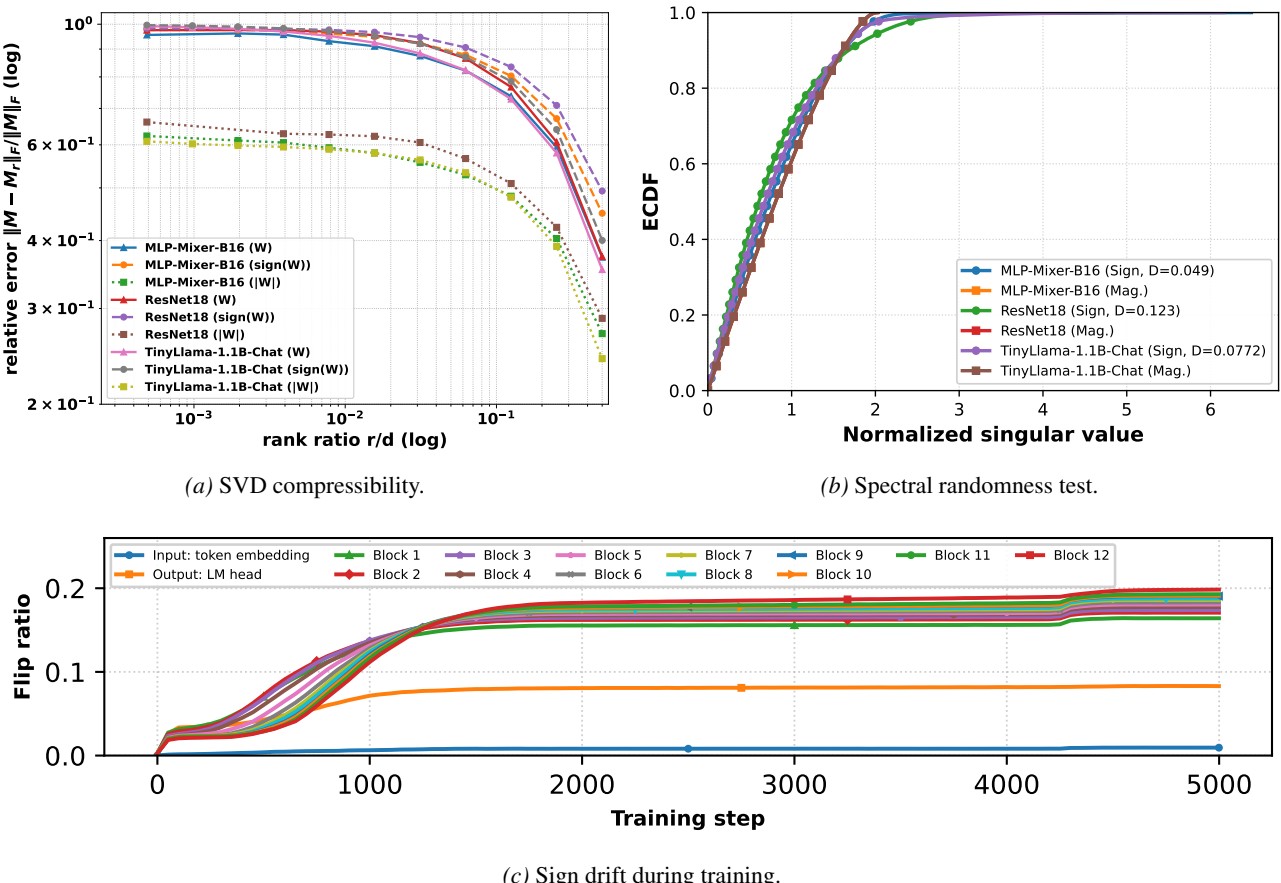

*(a)* SVD compressibility.

*(b)* Spectral randomness test.

*(c)* Sign drift during training.

*Figure 2.* **Empirical validation of one-bit wall. (a)** SVD compressibility (best rank-$r$ approximation error) of the raw weight matrix $W$, sign matrix $S = \text{sign}(W)$, and magnitude matrix $A = |W|$ as a function of rank ratio $r/d$ ($d = \min(m,n)$). **(b)** Spectral fit of sign matrices to an i.i.d. Rademacher baseline using a two-sample KS test on normalized singular values. **(c)** Initialization-to-trained sign drift in a Transformer trained on next-token prediction: flip ratio vs. initialization across layers (input→output) and pooled.

learned sign submatrices closely track the Rademacher baseline. This supports the interpretation that the trained sign patterns exhibit *noise-like behavior at the spectral level*, consistent with their poor low-rank compressibility. Additional randomness diagnostics are provided in Appendix E.2.

Furthermore, Figure 1 presents an information-theoretic evaluation. For a binary sign source with entropy rate $H_{\text{RD}}$ under Hamming distortion, Shannon's rate–distortion lower bound implies $\mathcal{D}^{\text{lb}}_{\text{RD}}(\mathcal{R}_{\text{RD}}) = h_2^{-1}(\max\{0, H_{\text{RD}} - \mathcal{R}_{\text{RD}}\})$. Estimated $H_{\text{RD}}$ for $\text{sign}(W)$ in representative pretrained models yields a near-Rademacher regime ($H_{\text{RD}} \approx 1$), leaving little room for sub-bit sign storage without incurring non-negligible sign distortion. The details of this analysis are provided in Appendix C.2.

**Dynamics Probe: Sign Drift during Training.** To assess whether the noise-like behavior of $S$ is produced during optimization or inherited from initialization, we track the sign mismatch ratio relative to the initial sign pattern during training. Let $\{W^{(l)}(t)\}_{l=1}^{L}$ denote the collection of matrix-

shaped weight tensors at training step $t$, where $W^{(l)}(t) \in \mathbb{R}^{m_l \times n_l}$ and $N_l := m_l n_l$. Define the sign flip ratio

$$\text{flip}(t) := \frac{1}{\sum_l N_l} \sum_{l,i,j} \mathbf{1}\left[\text{sign}(W^{(l)}_{ij}(t)) \neq \text{sign}(W^{(l)}_{ij}(0))\right].$$

We track matrix-shaped weights and report $\text{flip}(t)$ over time for token embeddings (input), each Transformer block, and the LM head (output). $\text{flip}(t)$ increases during the early phase but typically remains well below $0.5$ throughout training, indicating that most signs are inherited from initialization and remain stable. This connects the two observations above: signs appear noise-like because they remain close to a random initial template, and training rarely alters them.

> **Takeaway: empirical phenomenon**
>
> Empirically, weight signs are spectrally noise-like yet largely persistent during training.

## 2.2. Why Noise-Like Signs Create the One-Bit Wall

Sub-bit compression targets an average storage cost of less than one bit per parameter. Many successful schemes can reduce the *magnitude A* to below one bit per weight on average through quantization, low-rank factorization, pruning, and entropy coding. However, if the sign pattern $S$ closely resembles i.i.d. noise, it offers little exploitable structure for such compressors. In that regime, storing $S$ alone costs one bit per weight under any coordinatewise representation, establishing the **one-bit wall** even when the magnitudes are highly compressible.

The empirical phenomenon described translates into a concrete bottleneck: *in the sub-bit regime, sign storage becomes the dominant and potentially irreducible cost.* This motivates the remainder of the paper: we seek a mechanistic understanding of why signs persist as random-like and stable, and how to transform signs from a bottleneck into a controllable component. Section 3 provides a minimal theory that explains sign persistence as a consequence of rare boundary excursions. Section 4 then converts this understanding into practical interventions that actively promote compressible sign structure.

# 3. Sign Lock-In Theory

The empirical results suggest that optimization rarely changes signs: most coordinates keep the sign they received at initialization. This section formalizes a simple mechanism behind this persistence. A sign flip of a scalar coordinate can occur only if the trajectory crosses the boundary at $0$. If typical training dynamics keep coordinates at magnitudes bounded away from $0$, then sign flips must be initiated by rare excursions into a narrow boundary neighborhood. We show that, under the standard training setting of deep learning, the number of effective outer-to-outer sign flips admits a geometric-tail bound.

## 3.1. Problem Setting

We analyze a single coordinate because sign patterns are defined entrywise and sign-storage cost is inherently coordinatewise. The theory is stated for a one-dimensional adapted process; in experiments, we aggregate per-coordinate statistics across layers.

Let $(\Omega, \mathcal{F}, (\mathcal{F}_t)_{t \geq 0}, \mathbb{P})$ be a filtered probability space, and let $(w_t)_{t \geq 0}$ be a one-dimensional $(\mathcal{F}_t)$-adapted process that represents the discrete-time evolution of a scalar parameter over a finite horizon $T \in \mathbb{N}$. We introduce the radii $0 < \epsilon < \rho$ that separate a sign-stable outer region from a sign-ambiguous boundary neighborhood.

**Definition 3.1** (Regions). *Fix an outer threshold $\rho > 0$ and select a base boundary radius $\epsilon_0 \in (0, \rho)$. Let $\Delta$ be as stated in Assumption 3.3 and define*

$$\epsilon := \max\{\epsilon_0, \Delta\} \in (0, \rho),$$

*assuming $\rho > \Delta$ such that $\epsilon < \rho$. The outer region $\mathsf{Outer}(\rho)$ and boundary neighborhood $\mathsf{Bd}(\epsilon)$ are defined as*

$$\mathsf{Outer}(\rho) := \{|w_t| \geq \rho\}, \ \ \mathsf{Bd}(\epsilon) := \{|w_t| \leq \epsilon\}.$$

**Definition 3.2** (Stopping time). *With the convention $\inf \emptyset = \infty$, define recursively*

$$\sigma_0 := \inf\{t \geq 0 : \ |w_t| \geq \rho\},$$
$$\tau_k := \inf\{t > \sigma_{k-1} : \ |w_t| \leq \epsilon\}, \ \ k \geq 1,$$
$$\sigma_k := \inf\{t > \tau_k : \ |w_t| \geq \rho\}, \ \ k \geq 1.$$

## 3.2. Assumption: Bounded Update and Re-Entry

The sign lock-in bound is stated for the abstract adapted process $(w_t)$ and depends on two key components. The first excludes "pathological" one-step sign flips. The second controls the likelihood that the process returns to the boundary neighborhood after moving back into the outer region.

**Assumption 3.3** (Bounded update). *Fix $T \in \mathbb{N}$. There exist deterministic constants $\Delta > 0$ and $\delta_{\mathrm{upd}} \in [0, 1)$ such that, for every stopping time $\theta$ satisfying $\theta \leq T - 1$,*

$$\mathbb{P}\left[\mathcal{E}_\Delta(\theta)|\mathcal{F}_\theta\right] \geq 1 - \delta_{\mathrm{upd}} \ \text{a.s.},$$

*where the good event $\mathcal{E}_\Delta(\theta)$ is*

$$\mathcal{E}_\Delta(\theta) := \left\{ \max_{\theta \leq t \leq T-1} |w_{t+1} - w_t| \leq \Delta \right\}.$$

*In particular, on $\mathcal{E}_\Delta := \mathcal{E}_\Delta(0)$ we have $|w_{t+1} - w_t| \leq \Delta$ for all $0 \leq t \leq T - 1$, and the special case $\delta_{\mathrm{upd}} = 0$ yields an almost-sure uniform increment bound.*

Assumption 3.3 prevents "pathological" one-step sign flips that jump across the origin while remaining in the outer region. With $\epsilon \geq \Delta$ (Definition 3.1), any outer-to-outer sign flip must pass through $\mathsf{Bd}(\epsilon)$.

**Assumption 3.4** (Re-entry bound condition). *There exists $g_T \in (0, 1)$ such that for all $k \geq 0$,*

$$\mathbb{P}[\tau_{k+1} \leq T|\mathcal{F}_{\sigma_k}] \leq g_T \ \text{a.s. on } \{\sigma_k \leq T\}.$$

Rather than being a technical convenience, Assumption 3.4 delineates the boundary between a stable training regime, where meaningful weights persist, and a degenerate floating regime (Appendix D.6) in which the boundary becomes attractive and weights collapse toward zero. The latter requires inward drift strong enough to distort task loss and is therefore non-standard. Under standard smoothness and bounded-noise conditions, this assumption is satisfied for

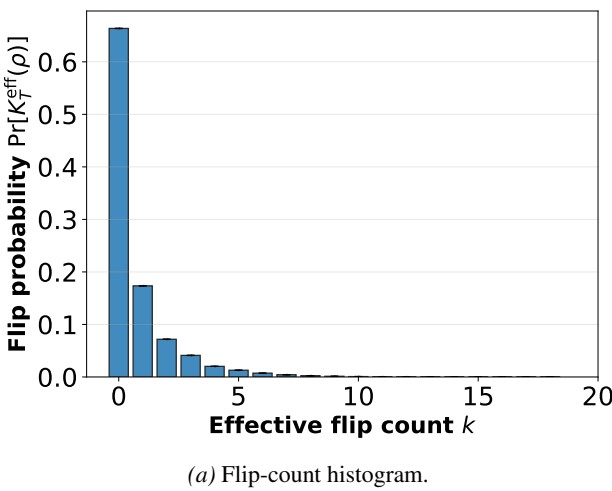

*(a)* Flip-count histogram.

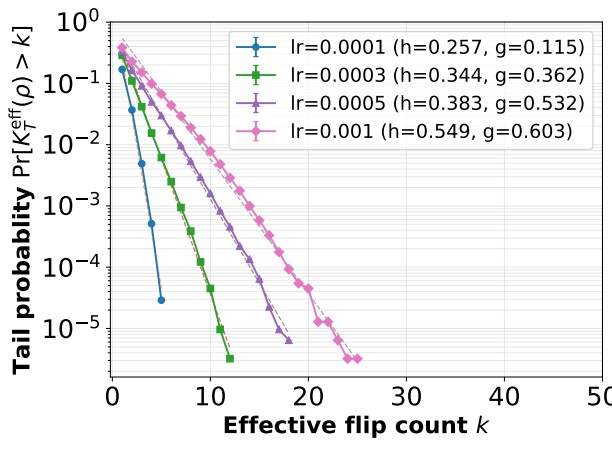

*(b)* Tail probability and geometric fit.

*Figure 3.* **Sign lock-in validation. Left:** Histogram of the effective outer-to-outer flip count $K_T^{\text{eff}}(\rho)$ across scalar weights; see Appendix C.4 for $T$, $\rho$, and $\epsilon$. **Right:** Tail probability $\mathbb{P}[K_T^{\text{eff}}(\rho) \geq k]$ on a log scale for multiple learning rates, with dashed geometric fits of the form $\hat{h} \, \hat{g}^{\,k-1}$.

scheduled SGD (Proposition D.10). The proof is based on Lemma D.8, which shows that the inward drift toward the sign boundary—even if present—is bounded by the finite cumulative gradient norm ensured by standard descent theory.

**Proposition 3.5** (Informal version of Proposition D.10: re-entry bound in SGD)**.** *Under the standard bounded-update and descent/noise conditions, there exists an explicit upper bound $g_T^{\text{SGD}}$ such that for all $k \geq 0$,*

$$\mathbb{P}[\tau_{k+1} \leq T \mid \mathcal{F}_{\sigma_k}] \leq g_T^{\text{SGD}}.$$

*Moreover, $g_T^{\text{SGD}}$ decreases when (i) the boundary margin $\rho - \epsilon$ grows, (ii) step sizes decay so that $\sum_{t<T} \eta_t^2$ is small, and (iii) mini-batch noise is moderate. Consequently, Assumption 3.4 holds with $g_T := g_T^{\text{SGD}}$, yielding the geometric-tail bound in Theorem 3.6.*

### 3.3. Geometric Tail for Effective Sign Flips

An outer-to-outer sign flip can occur only if the trajectory (i) exits the outer region, (ii) enters a neighborhood of the boundary, and (iii) re-enters the outer region from the opposite side. Assumption 3.4 uniformly controls step (ii) over time, while Assumption 3.3 prevents single update transitions from (i) to (iii). Taken together, these conditions yield a geometric-tail bound on the number of effective outer-to-outer sign flips.

**Theorem 3.6** (Sign Lock-in Theorem)**.** *Under Assumptions 3.3–3.4, define the effective outer-to-outer flip count up to time $T$ by*

$$K_T^{\text{eff}}(\rho) := \sum_{k \geq 1} \mathbf{1}\Big[\sigma_k \leq T,\ \text{sign}(w_{\sigma_k}) \neq \text{sign}(w_{\sigma_{k-1}})\Big],$$

*which is well-defined since $|w_{\sigma_k}| \geq \rho > 0$ on $\{\sigma_k \leq T\}$. Let $h_T := \mathbb{P}[\tau_1 \leq T] \in [0, 1]$. Then for all integers $k \geq 1$,*

$$\mathbb{P}[\tau_k \leq T] \leq h_T \, g_T^{k-1},$$

*and consequently,*

$$\mathbb{P}\big[K_T^{\text{eff}}(\rho) \geq k\big] \leq h_T g_T^{k-1} + \delta_{\text{upd}}.$$

**Remark 3.7.** *The sign lock-in theorem relies solely on Assumptions 3.3 and 3.4, making it optimizer-agnostic. The uniform re-entry condition of Proposition 3.5 is also not tied to SGD. SGD variant optimizers follow the same arguments underlying Proposition D.10.*

The theorem formalizes the excursion picture: sign changes are initiated only by boundary hits, and repeated effective outer-to-outer sign flips are exponentially unlikely. To connect this excursion-based perspective to the most common empirical notion of sign persistence, we next demonstrate a complementary fact: if the sign at time $T$ differs from the initial sign, then the trajectory must have entered a small $\Delta$-neighborhood of the origin at least once, provided that one-step updates are bounded.

**Proposition 3.8.** *Assume Assumption 3.3. Let $(w_t)_{t=0}^T$ be a real-valued process and $s_t := \text{sign}(w_t) \in \{\pm 1\}$ with a fixed tie-break at 0. Define the first hit time of the $\Delta$-band*

$$\tau_\Delta := \inf\{t \in \{0, 1, \dots, T\} : |w_t| \leq \Delta\},$$

*with the convention $\inf \emptyset = \infty$. We have the deterministic implication*

$$\mathbf{1}\{s_T \neq s_0\} \leq \mathbf{1}\{\tau_\Delta \leq T\}.$$

*Consequently,*

$$\mathbb{P}[s_T \neq s_0] \leq \mathbb{P}[\tau_\Delta \leq T] + \delta_{\text{upd}}.$$

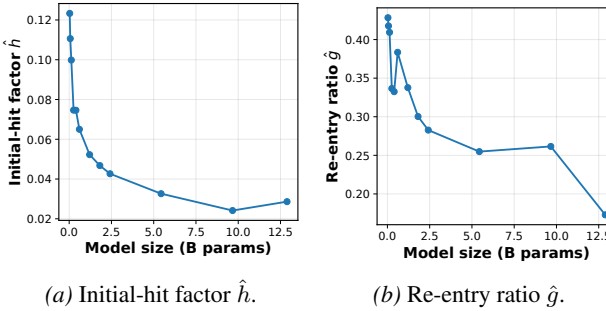

*(a)* Initial-hit factor $\hat{h}$.      *(b)* Re-entry ratio $\hat{g}$.

*Figure 4.* **Billion-scale sweep of the lock-in parameters** $(\hat{h}, \hat{g})$.

> **Takeaway: main result of sign lock-in theory**
>
> Even if a weight sign flips, its effective flip count decays exponentially.

### 3.4. Empirical Validation of Sign Lock-In

We test the core prediction of Theorem 3.6: the effective outer-to-outer sign flip count $K_T^{\text{eff}}(\rho)$ should have a rapidly decaying tail. Figure 3 reports (i) the histogram of $K_T^{\text{eff}}(\rho)$ in baseline training, and (ii) the tail probability $\mathbb{P}[K_T^{\text{eff}}(\rho) \geq k]$ on a semi-log scale. The distribution is sharply concentrated near small values of $K_T^{\text{eff}}(\rho)$, and the tail exhibits an approximately linear trend on the log scale, consistent with geometric decay. We overlay the fitted geometric form $\hat{h}\,\hat{g}^{\,k-1}$, where $(\hat{h}, \hat{g})$ are estimated from the empirical distribution under a zero-inflated geometric model. As predicted by the theory, varying the learning rate changes the effective update scale $\Delta$ and hence the prefactor and decay rate of the flip-count distribution, while preserving its geometric form. Accordingly, we observe the same qualitative behavior across learning rates, confirming sign lock-in as a robust baseline phenomenon. An analogous experiment on a vision task is also provided in Appendix E.3.

### 3.5. Practical Insights of Sign Lock-In Theory

**Modern deep learning models tend to show stronger lock-in.** Appendix D.4 details the theoretical properties of the initial-hit factor $h$ and the re-entry ratio $g$, derived from sign lock-in theory, and provides experimental validation. (i) The lock-in parameters vary substantially depending on the learning rate and its schedule. Under a fixed training horizon and peak step size, lock-in becomes progressively *weaker* in the order of inverse decay $\rightarrow$ cosine decay $\rightarrow$ exponential decay $\rightarrow$ constant learning rate. (ii) Scale-invariance mechanisms, such as ReLU positive homogeneity and normalization layers, further strengthen lock-in by suppressing effective boundary re-entry. (iii) Increasing the batch size or model size enhances lock-in due to reduced stochastic noise and width-induced stabilization effects. *Consequently,*

*sign lock-in weakens when small models are trained with a constant learning rate and small batch size, whereas models following modern standard architectures and training recipes, including LLM, tend to exhibit strong sign lock-in.*

**Billion-scale validation.** We sweep the size of the model from the $\sim 30\text{M}$ to over 10B weight parameters and estimate the lock-in parameters $(\hat{h}, \hat{g})$. Appendix C.5 details experiments. As shown in Figure 4, both the initial-hit factor $\hat{h}$ and the re-entry ratio $\hat{g}$ decrease monotonically with scale. In the largest models, $\hat{h}$ becomes very small and $\hat{g}$ approaches zero, indicating that sign flips are rarely initiated and almost never repeated. These results demonstrate that sign lock-in is systematically strengthened with model size, consistent with the prediction of sign lock-in theory.

> **Takeaway: effectiveness of sign lock-in theory**
>
> Sign lock-in theory robustly predicts the histogram of sign flips and demonstrates that lock-in effects are strengthened in modern neural networks.

## 4. Sign Lock-In Enhancement

The sign lock-in theory suggests that effective outer-to-outer sign flips are rare and exhibit a geometric tail. This turns the empirical one-bit wall into a constructive opportunity: if the initial sign pattern is chosen to be compressible and training preserves it, then the trained model can reuse that sign structure rather than store an arbitrary learned sign matrix. Our resulting procedure is therefore a *from-scratch* training method. It selects the sign structure before optimization begins and trains the model around that structure; it is not intended as a post-training compressor that takes an arbitrary pretrained model and rewrites its signs after the fact. We therefore ask the following question:

> *Can sign lock-in preserve a compressible initialization template while maintaining task quality?*

To probe this question, we introduce a theory-guided approach that controls the two quantities governing boundary excursions in Theorem 3.6: (i) the probability of reaching the sign boundary, the initial-hit factor $h_T$, and (ii) the probability of returning to the boundary after escaping, the re-entry ratio $g_T$. The soft version of this approach tests whether a structured template can survive ordinary training with little quality loss. The hard template-constrained version then tests the compression implication directly: if signs are re-generable, the bit budget can be spent on magnitudes.

Figure 5 illustrates the key point with the example above. If training begins from ordinary random signs, sign lock-in preserves a Rademacher-like, high-rank sign matrix, which

$$S_{\text{rand}} = \begin{bmatrix} 1 & 1 & -1 & 1 \\ -1 & 1 & 1 & -1 \\ 1 & -1 & 1 & -1 \\ -1 & -1 & 1 & 1 \end{bmatrix}$$

*(a) A random sign pattern is not low-rank-friendly.*

$$S_{\text{temp}} = \begin{bmatrix} 1 & 1 & -1 & -1 \\ 1 & 1 & -1 & -1 \\ -1 & -1 & 1 & 1 \\ -1 & -1 & 1 & 1 \end{bmatrix} = \begin{bmatrix} 1 \\ 1 \\ -1 \\ -1 \end{bmatrix} \begin{bmatrix} 1 & 1 & -1 & -1 \end{bmatrix}$$

*(b) A deliberately chosen template can be rank one.*

*Figure 5.* **Re-generable sign templates drive the effective sign cost to zero.** A random sign matrix such as $S_{\text{rand}}$ has no compact description and costs about one bit per weight. In contrast, $S_{\text{temp}}$ is the elementwise sign of a rank-1 outer product and is therefore *re-generable* from a small specification (a random seed and the rank). Storing only this specification amortizes the per-weight sign cost to $\approx 0$ bits as the model grows, freeing the entire bit budget for the magnitudes.

is precisely the one-bit wall. Fixing such signs after training would not help. The proposed method changes the initial condition: it starts from a structured, low-rank-friendly sign template such as $S_{\text{temp}}$ and uses sign lock-in enhancement to preserve it. Thus sign persistence becomes an advantage rather than a bottleneck.

### 4.1. Low-Rank Sign Templates for Suppressing the One-Bit Wall

Beyond describing a naturally emerging phenomenon, sign lock-in can be *actively controlled* through artificial interventions. We first choose a low-rank, compressible sign template as an initialization prior, and then use two lightweight mechanisms to keep trajectories away from the sign boundary. Gap initialization reduces initial boundary hits, while outer-drift regularization reduces later re-entry. Appendix D.5 gives the corresponding theoretical support. The complete details of the end-to-end pipeline implementation are provided in Appendix E.4.

**Low-rank sign template.** We begin by specifying a compressible sign template before training starts. For a layer $l$ with a weight matrix $W^{(l)} \in \mathbb{R}^{m \times n}$, we create a low-rank template by sampling two factor matrices $G \in \mathbb{R}^{m \times r}$ and $H \in \mathbb{R}^{n \times r}$, where $r \ll \min(m, n)$, and then taking the elementwise sign of their product:

$$G_{ik} \overset{\text{i.i.d.}}{\sim} \mathcal{N}(0, 1), \; H_{jk} \overset{\text{i.i.d.}}{\sim} \mathcal{N}(0, 1), \; T^{(l)} := \text{sign}(GH^{\top}),$$

The rank parameter $r$ controls the intrinsic degrees of freedom of the template, making $T^{(l)}$ clearly specified and straightforward to reuse. We treat $T^{(l)}$ as the initial distribution template for the weights in layer $l$. Let $\mathcal{D}$ be any distribution supported on $\mathbb{R}_{>0}$ and draw magnitudes $A_{ij}^{(l)} \overset{\text{i.i.d.}}{\sim} \mathcal{D}$. We initialize the weights as $W^{(l)} := T^{(l)} \odot A^{(l)}$. This design establishes *a priori* a sign structure that is easy to store and reuse; in the experiments, we use $r = 2$ as a representative case. Because the template is chosen before optimization, the method targets new training runs and is distinct from post-training quantization or pruning methods that operate on already-trained weights.

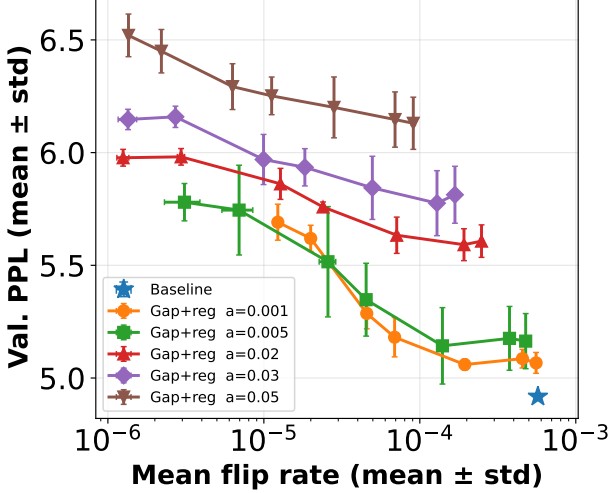

*Figure 6.* **Flip–quality trade-off with a compressible sign template** ($r{=}2$). Validation perplexity mean±std over three seeds vs. the mean per-step sign flip rate (`flip_mean`). Each curve fixes the gap threshold $a_{\text{init}}$ and sweeps the log-barrier weight $\lambda$ from left to right ($0.5, 0.3, 0.1, 0.05, 0.01, 0.001, 0.0001$). Stronger stabilization suppresses flips but can worsen perplexity, while intermediate $a_{\text{init}}$ and $\lambda$ achieve large flip reduction with little loss in validation quality.

**Gap Initialization.** To reduce early sign flips caused by weights drifting near zero, we initialize each weight with a margin explicitly away from the origin. Let $\sigma_{\text{init}} > 0$ denote the base initialization scale and define a gap threshold, $a_{\text{init}} := c_{\text{gap}}\sigma_{\text{init}}$, $c_{\text{gap}} > 0$, where $c_{\text{gap}}$ is a user-chosen constant controlling the gap size. For each entry, we sample $z \sim \mathcal{N}(0, \sigma_{\text{init}}^2)$ and reject the draw if $|z| < a_{\text{init}}$, repeating until $|z| \geq a_{\text{init}}$. Let $Z^{(l)} \in \mathbb{R}^{m \times n}$ be the resulting matrix with entries that are independently and identically distributed according to this rejection-sampling procedure. Equivalently, each $Z_{ij}^{(l)}$ follows a two-sided truncated Gaussian supported on $\mathbb{R} \setminus [-a_{\text{init}}, a_{\text{init}}]$. We initialize the layer-$l$ weights as $W_0^{(l)} := Z^{(l)}$. By construction, this initialization suppresses early excursions into the near-zero region, where sign changes are most likely.

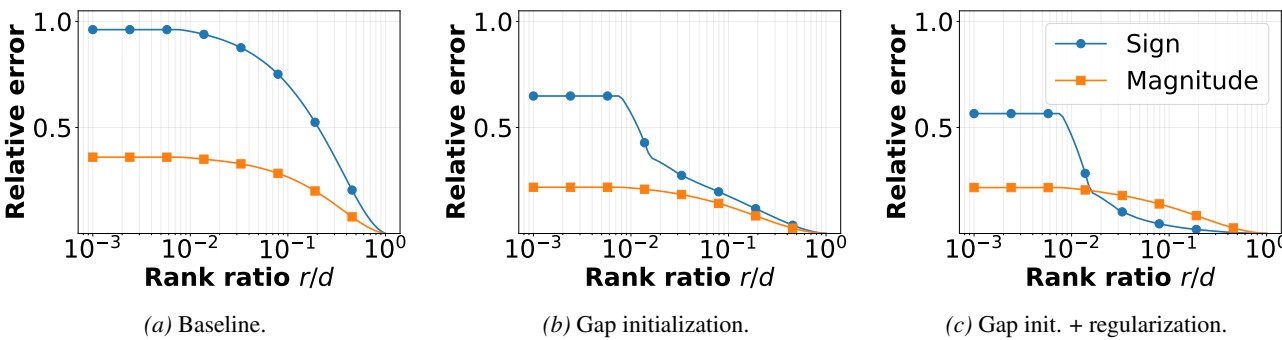

*(a)* Baseline.  *(b)* Gap initialization.  *(c)* Gap init. + regularization.

*Figure 7.* **Sign vs magnitude low-rank compressibility example.** Relative Frobenius error $E_r(M) = \|M - M_r\|_F/\|M\|_F$ as a function of rank ratio $r/d$ (log scale), for the sign matrix $S = \text{sign}(W)$ and magnitude matrix $A = |W|$, evaluated on final trained weights under baseline, gap initialization only, and gap initialization with regularization.

**Outer-drift regularization.** Even with a gap at initialization, individual weights may later drift back to zero, where sign changes are most probable. To discourage such re-entries during early optimization, we introduce a lightweight log-barrier that penalizes small magnitudes. For a weight matrix $W \in \mathbb{R}^{m \times n}$, a gap threshold $a_{\text{init}} > 0$, and a numerical stabilizer $\epsilon_{\text{lb}} > 0$, we define the following:

$$R_{\text{LB}}(W; a_{\text{init}}, \epsilon_{\text{lb}})$$
$$:= \frac{1}{mn} \sum_{i,j} \log\left(\max\left\{1, \frac{a_{\text{init}}}{|W_{ij}| + \epsilon_{\text{lb}}}\right\}\right).$$

This penalty is 0 whenever $|W_{ij}|$ is safely outside the near-zero band, i.e., when $|W_{ij}| + \epsilon_{\text{lb}} \geq a_{\text{init}}$, and it increases smoothly as $|W_{ij}|$ approaches 0. Let $\theta$ denote all model weight parameters, and let $\mathcal{L}_{\text{task}}(\theta)$ be the task loss, e.g., negative log-likelihood. We apply the barrier to a selected set of layers $\mathcal{M}$, resulting in the time-dependent training objective at the optimization step $t$:

$$\mathcal{L}_{\text{total}}(\theta; t) := \mathcal{L}_{\text{task}}(\theta) + \lambda(t) \sum_{l \in \mathcal{M}} R_{\text{LB}}(W^{(l)}; a_{\text{init}}, \epsilon_{\text{lb}}),$$

where $\lambda(t) \geq 0$ controls the strength of the regularizer. In practice, we keep $\lambda(t)$ constant during an initial warmup phase and then reduce it to 0, so that the penalty primarily influences early dynamics. Overall, this regularizer biases optimization away from the near-zero region after weights have moved into the outer region $|W_{ij}^{(l)}| \gtrsim a_{\text{init}}$, reducing repeated boundary excursions and helping to preserve the intended sign structure.

### 4.2. Empirical Validation of Sign Lock-In Enhancement

We now validate the mechanism in three steps. First, we test whether gap initialization and outer-drift regularization reduce sign flips without substantially degrading task quality. Second, we check whether the resulting trained weights preserve the intended low-rank sign structure while keeping magnitudes compressible. Third, we evaluate the end-to-end sub-bit compression consequence of using a regenerable sign template and magnitude-only SVD storage. All template-based experiments train models from scratch with the template specified at initialization; they should not be interpreted as post-training conversion of an arbitrary pretrained model. The first two tests use a Transformer trained for next-character prediction; detailed settings are provided in Appendix C.4, with an additional vision task in Appendix E.3.

**Do initial gap and outer drift enhance lock-in?** We empirically examine whether gap initialization and outer-drift regularization effectively enhance sign lock-in. Figure 6 shows the trade-off between task quality and the mean sign-flip rate. While the two mechanisms reinforce each other, a combination of a smaller gap and stronger outer-drift regularization consistently lies on the Pareto frontier, suppressing sign flips to $\sim 10^{-3}$ with only about a one-point increase in perplexity. These results agree fully with the theoretical prediction. We also report direct control effect of initial hit factor and re-entry ratio in Appendix C.4.

**Low-rank structure of sign and magnitude preserved?** We next investigate whether sign lock-in enhancement preserves the low-rank structure of the magnitude, which is crucial for effective compression. As shown in Figure 7, the magnitude matrix maintains a low-rank structure comparable to the baseline, even under strong sign stabilization. In contrast, the sign matrix becomes substantially more amenable to low-rank approximation due to the preservation of its structured initialization. This result confirms that sign lock-in enhancement stabilizes sign patterns without compromising the compressibility of magnitudes.

**Does the low-rank sign template method improve sub-bit compression?** We finally evaluate the end-to-end sub-bit compression consequence of using a low-rank sign template and magnitude-only SVD storage. When signs are fixed by

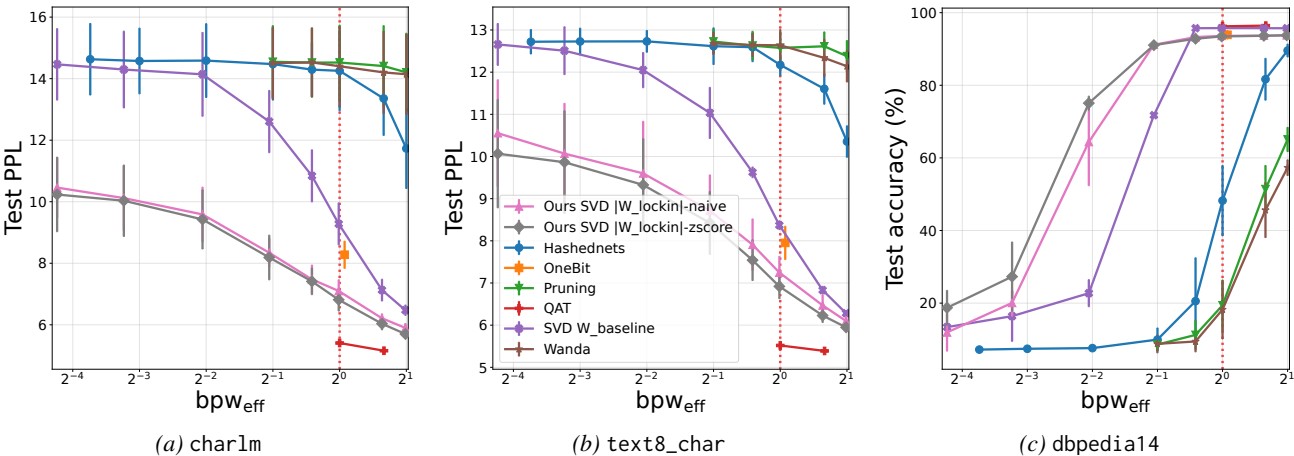

*Figure 8.* **Performance vs. effective bits per weight** ($\mathrm{bpw}_{\mathrm{eff}}$) **on benchmark tasks.** Markers indicate the mean over three seeds and error bars show one standard deviation. Lower is better for perplexity (CharLM, Text8-Char), while higher is better for accuracy (DBPedia14).

the template, the sign component incurs no storage cost, so the remaining budget can be used to compress the nonnegative magnitudes with truncated SVD and quantized factors. Figure 8 reports the resulting performance–bit trade-off on three representative benchmark tasks. In contrast, vanilla SVD on raw weights and explicit 1-bit baselines either degrade sharply near or stall at one bit per weight, directly visualizing the one-bit wall. Across language modeling and classification tasks, SVD $|W_{\mathrm{lockin}}|$ is consistently stronger than applying the same SVD budget directly to raw weights in the sub-bit region, and it remains competitive against hashing, one-bit, and pruning-based baselines. This supports the practical implication of sign lock-in: once a structured sign template is preserved, magnitudes become the main compressible object and the one-bit sign wall can be bypassed. This comparison evaluates the benefit of choosing and preserving a low-rank sign template during training, not a post-training replacement of the sign matrix in pretrained checkpoints. Additional distillation-based variants and bit-accounting details are provided in Appendix E.4.4.

## 5. Conclusion

Learned signs are harder to compress than magnitudes, exhibit near-random spectral statistics, yet remain strongly aligned with initialization. We explain these phenomena through a sign lock-in theory predicting a geometric flip-count tail, and validate it empirically. Based on this mechanism, we propose gap initialization and outer-drift regularization to suppress boundary visits and sign flips. Our results indicate that stabilizing sign structure is a practical prior for sub-bit compression, and that surpassing the one-bit wall requires explicit control and reuse of sign structure. Leveraging these insights to develop a post-training method that extends sign lock-in to arbitrary pretrained checkpoints is an attractive direction for future work.

## 6. Limitations

Our study has several limitations. (i) The low-rank sign-template method is a from-scratch training approach: the template must be selected before optimization and then preserved during training. It is not a post-training compressor for arbitrary pretrained checkpoints, and extending sign-template control to that setting remains future work. (ii) While sign lock-in typically emerges under natural training dynamics, the system can be driven into a *sign floating mode* under extraordinarily strong magnitude-side regularization that induces persistent attraction toward the sign boundary. We also analyze sign floating mode in Appendix D.6. (iii) We focus on simple enforcement methods; other strategies remain unexplored. (iv) We have conducted extensive experiments including a billion-scale validation, but broader empirical coverage is left to future work. (v) While we show that sign degrees of freedom are rarely exploited by optimization, we do not analyze the representational role of signs when treated as fixed parameters, nor their potential contribution to expressivity.

## Acknowledgements

This work was partially supported by JST BOOST, Japan (Grant No. JPMJBY24D0).

## Impact Statement

This work aims to improve the scientific understanding and practical efficiency of sub-bit neural-network compression; the proposed method reduces model storage and memory traffic and can lower deployment cost and energy consumption. At the same time, aggressive compression may degrade rare or boundary-case behavior, which is particularly important in safety-critical applications.

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

# A. Notation

We summarize the key symbols that appear throughout the paper. Auxiliary variables that are introduced only within a particular proof, such as shifted or stopped processes and intermediate martingales, are defined locally in the corresponding appendix.

| Symbol | Description |
|---|---|
| **Weights, signs, and low-rank structure** | |
| $W^{(\ell)} \in \mathbb{R}^{m \times n}$ | weight matrix at layer $\ell$ |
| $S := \text{sign}(W) \in \{\pm 1\}^{m \times n}$ | sign matrix ties at 0 mapped to $+1$ |
| $A := |W| \in \mathbb{R}_{\geq 0}^{m \times n}$ | magnitude matrix; $W = S \odot A$ |
| $\mathbf{T}^{(\ell)} \in \{\pm 1\}^{m \times n}$ | re-generable sign template used for sign-template enforcement |
| $E_r(M)$ | relative rank-$r$ truncated-SVD error: $E_r(M) = \|M - M_r\|_F / \|M\|_F$ |
| $d := \min(m, n), \ q := r/d$ | effective dimension and rank ratio |
| **Rate–distortion quantities (Figure 1 / Appendix C.2)** | |
| $\mathcal{R}_{\text{RD}}, \mathcal{D}_{\text{RD}}$ | rate (bits/sign) and Hamming distortion $\mathcal{D}_{\text{RD}} = \mathbb{P}[S \neq \hat{S}]$ |
| $\mathcal{D}_{\text{RD}}^{\text{lb}}(\mathcal{R}_{\text{RD}})$ | Shannon lower bound on distortion (inverse view) |
| $H_{\text{RD}}, \widehat{H}_{\text{RD}}$ | entropy-rate parameter and its empirical proxy (estimated from selected layers) |
| $h_2(p), h_2^{-1}(y)$ | binary entropy and its inverse on $[0, 1/2]$ |
| $L, N_\ell, \hat{p}_\ell$ | number of selected layers, layer sizes, and empirical sign frequency used to form $\widehat{H}_{\text{RD}}$ |
| **Sign flips and stopping-time framework** | |
| $\mathbf{1}\{\cdot\}, N$ | indicator function and number of tracked scalar entries |
| $\text{flip}(t), \text{flip\_mean}$ | mismatch-to-initialization ratio and mean step-wise flip rate |
| $\mathcal{E}_\Delta, \Delta, \delta_{\text{upd}}$ | bounded-update event $|w_{t+1} - w_t| \leq \Delta$ and its failure probability $\mathbb{P}[\mathcal{E}_\Delta^c] \leq \delta_{\text{upd}}$ |
| $\rho, \epsilon$ | outer threshold and boundary-neighborhood radius. We set $\epsilon = \max\{\epsilon_0, \Delta\}$. |
| $\sigma_k, \tau_k$ | $k$-th outer-entry time and boundary-hit time |
| $K_T^{\text{eff}}(\rho)$ | effective (outer-to-outer) sign-flip count up to time $T$ |
| $h_T, g_T$ | lock-in parameters: $h_T = \mathbb{P}[\tau_1 \leq T]$ and an upper bound on re-entry probability |
| **Quantization and bit accounting** | |
| $Q_b(x; \alpha), b, \alpha$ | $b$-bit symmetric uniform quantizer with scale $\alpha$ |
| $\text{bpw}_{\text{eff}}, \text{bpw}_{\text{target}}$ | effective bits-per-weight and target budget grid |
| **Selected appendix-only bundles (defined where used)** | |
| $\mathcal{L}, v_t, g_t, \eta_t, \xi^2$ | objective / iterate / stochastic gradient / step size / noise proxy (App. D.3) |
| $u_t, r_t, \widetilde{w}_t, (\Delta_{\text{blk}}, \delta_{\text{blk}})$ | BN+ReLU scale-invariance block notation and its bounded-update parameters (App. D.3) |
| $\zeta, \omega, \epsilon_{\text{zs}}, D_{\text{zs}}, (\cdot)^{\text{zs}}$ | magnitude z-score preconditioning notation (App. E.4.6) |
| $\mathcal{M}_{\text{float}}, \rho_f, R_{\text{ID}}(W; \rho_f), \mathcal{L}_{\text{float}}$ | floating-mode regularization objects (App. D.6) |

To avoid confusion between time indices and discrete counters, we denote time decrements as $(t - 1)$, e.g., $w_{(t-1)}$, while counter decrements are indicated using subscripts, e.g., $\sigma_{k-1}, \tau_{k-1}$.

# B. Additional Related Work

Recent work on compressing pretrained models has evolved from few-bit quantization to sub-bit regimes. One line of work represents weight matrices using a small number of binary bases with learned or compactly encoded coefficients, enabling effective storage at one or sub-bit levels and efficient reconstruction (Xu et al., 2024; Boža & Macko, 2025; Bulat et al., 2024; Lee et al., 2025; Gu et al., 2025; Ichikawa et al., 2025). Pruning and sparsification reduce the number of stored parameters and are often combined with quantization (Han et al., 2015; Frantar & Alistarh, 2023; Sun et al., 2024). However, these approaches largely treat the sign pattern as a given object to represent, rather than analyzing what makes signs compressible or incompressible. In particular, there is limited research directly studying the statistical structure and training-time dynamics of weight signs, the underlying object that 1-bit bases must ultimately represent. We address this gap by empirically characterizing sign incompressibility and persistence across architectures, providing a minimal stopping-time

theory that explains sign stability and offers actionable methods to control sign evolution.

Ultra-low-bit training includes binary and ternary networks that constrain weights and/or activations to small discrete sets, typically relying on the straight-through estimator (Bengio et al., 2013; Courbariaux et al., 2015; Hubara et al., 2016; Li et al., 2016; Zhu et al., 2017). For 1-bit inference, accuracy has improved through scaling strategies and architectural refinements, including XNOR-type formulations and specialized training recipes (Rastegari et al., 2016; Zhou et al., 2016; Lin et al., 2017; Liu et al., 2018; Qin et al., 2020; Liu et al., 2020). These studies discuss the optimization of sign-only networks, and several studies (Xiang et al., 2024; Lin et al., 2025; Xu et al., 2021) report that typical binary networks do not reach a sign-flip rate of 50%. However, these studies are observation based and there is still no unified explanation of sign dynamics. Our goal is also different from classical binary or ternary QAT such as DoReFa-Net and TTQ: those methods learn low-bit weights or activations but still store the resulting sign or ternary states explicitly, often together with auxiliary scales. In contrast, our sign-template method targets the near- or sub-one-bit storage regime by choosing a compressible sign prior before training, preserving it through sign lock-in, and reallocating the remaining bit budget to magnitudes.

More broadly, quantization research includes early analyses of training with limited numerical precision (Gupta et al., 2015), practical inference pipelines that utilize only integer-arithmetic (Jacob et al., 2018), and methods for quantization-aware training (QAT) or post-training quantization (PTQ) that learn or parameterize clipping thresholds and step sizes (Choi et al., 2018; Esser et al., 2020). For Transformers and large language models, PTQ has matured within the 8-bit regime (Dettmers et al., 2022; Xiao et al., 2023) and has recently advanced significantly for settings that use only 4-bit weights (Frantar et al., 2023; Lin et al., 2024). Complementary compression mechanisms reduce the cost of storing magnitudes via codebooks, hashing, or vector quantization (Chen et al., 2015; Gong et al., 2014; Ullrich et al., 2017). In parallel, communication-efficient optimization compresses gradients into signs or ternary values (Seide et al., 2014; Alistarh et al., 2017; Wen et al., 2017; Lin et al., 2018; Bernstein et al., 2018), providing further evidence that sign information remains useful even under extreme discretization.

Pruning and sparsity methods range from early second-order criteria (LeCun et al., 1989; Hassibi & Stork, 1993) and magnitude-based pruning (Han et al., 2015) to more recent techniques tailored for large language models (Sun et al., 2024; Frantar & Alistarh, 2023). Structured compression via low-rank and tensorized parameterizations exploits correlations in weights and has a long history in deep learning (Denil et al., 2013; Denton et al., 2014; Jaderberg et al., 2014; Sainath et al., 2013; Novikov et al., 2015). These lines of work complement our approach, which effectively reallocates bits to magnitude factors after rendering sign information free by construction. Theoretical perspectives on near-initialization training in wide networks, including the neural tangent kernel, over-parameterized convergence analyses, and "lazy training," predict limited parameter movement and therefore offer a plausible explanation for sign persistence during training (Jacot et al., 2018; Du et al., 2019; Allen-Zhu et al., 2019; Lee et al., 2019; Chizat et al., 2019). Finally, our observation that sign matrices exhibit spectral behavior close to i.i.d. Rademacher baselines naturally connects to classical random matrix theory (Marchenko & Pastur, 1967; Wigner, 1955) and to spectral viewpoints on deep networks (Pennington et al., 2017; Martin & Mahoney, 2018; Sakai et al., 2022).

## C. Additional Experimental Details

This appendix documents experimental settings and implementation details that support the empirical results in the main text. Specifically:

- **Appendix C.1** reports the hardware and software environment used across experiments.
- **Appendix C.2** describes the experimental settings for estimating the entropy-rate proxy.
- **Appendix C.3** details the protocol used for the randomness tests (Figure 2(a–c)).
- **Appendix C.4** summarizes the full experimental details for the language-task sign lock-in experiments.
- **Appendix C.5** reports the billion-scale LLM validation setup.

### C.1. Hardware and Software

To ensure reproducibility, we set a global random seed in both NumPy and PyTorch and report the hardware and software configurations used in our experiments. All runs were conducted on a single NVIDIA A100 GPU. Our software environment

included Python `3.12.12`, PyTorch `2.9.0`, torchvision `0.24.0`, timm `1.0.22`, transformers `4.57.3`, and SciPy `1.16.3`.

## C.2. Experimental Settings for Estimating Entropy Proxy

We consider three pretrained models: TinyLlama-1.1B-Chat (Zhang et al., 2024), ResNet18 (torchvision; `resnet18` with `weights="DEFAULT"`), and MLP-Mixer-B16 (timm; `mixer_b16_224` with `pretrained=True`). From each model, we extract two-dimensional internal weight tensors and convert them into matrices suitable for unified analysis. For linear layers, we use the native parameter matrix $W \in \mathbb{R}^{m \times n}$. For convolutional layers, we flatten each Conv2d kernel into a matrix of shape $W \in \mathbb{R}^{C_{\mathrm{out}} \times (C_{\mathrm{in}} \, k_h \, k_w)}$. For Transformer models, we exclude tensors with parameter names that contain `embed` or `lm_head` to avoid including embedding tables and output heads that may exhibit distinct statistical structures.

To focus computation on the most informative layers, we prioritize large matrices by sorting all extracted tensors according to $d = \min(m, n)$ in descending order and selecting up to $L = 6$ matrices per model. Let $N_l$ denote the number of entries in the selected matrix for layer $l$. Whenever we compute a statistic per layer, we aggregate it across layers using an entry-weighted average,

$$\widehat{H}_{\mathrm{RD}} := \frac{\sum_{l=1}^{L} N_l \, \widehat{H}_{\mathrm{RD},l}}{\sum_{l=1}^{L} N_l},$$

so that larger matrices contribute proportionally to the overall estimate.

Given each weight matrix $W$, we form the sign matrix $S = \mathrm{sign}(W)$ elementwise, adopting the convention $\mathrm{sign}(0) = +1$ so that $S \in \{\pm 1\}^{m \times n}$. To capture local dependencies beyond the marginal frequency of positive and negative signs, we estimate the entropy of small contiguous sign patches. For each selected sign matrix $S^{(l)} \in \{\pm 1\}^{m \times n}$, we sample contiguous $3 \times 3$ patches $P \in \{\pm 1\}^{3 \times 3}$ by choosing a top-left corner $(i, j)$ uniformly at random, subject to $1 \le i \le m - 2$ and $1 \le j \le n - 2$, and then taking $P = S^{(l)}[i : i + 2, \, j : j + 2]$. We map each patch to a 9-bit pattern index by converting $\pm 1$ to $0, 1$ and packing the resulting bits, yielding a categorical variable over $2^9 = 512$ possible patterns. Let $\hat{p}^{(l)}(u)$ denote the empirical frequency of pattern $u$ among the sampled patches in layer $l$. We compute the plug-in Shannon entropy of the patch distribution,

$$\widehat{H}_{\mathrm{patch},l} := -\sum_{u=1}^{512} \hat{p}^{(l)}(u) \log_2 \hat{p}^{(l)}(u),$$

and define the corresponding entropy-rate proxy by normalizing per site, $\widehat{H}_{\mathrm{RD},l} := \widehat{H}_{\mathrm{patch},l}/9$. Under an i.i.d. Rademacher field, this quantity approaches 1 (up to finite-sample effects), whereas it decreases when local structure induces predictable, low-entropy patch patterns. We report the entry-weighted aggregate $\widehat{H}_{\mathrm{RD}}$ in Figure 1. For scalability, when a full enumeration of patch locations is computationally expensive, we estimate $\hat{p}^{(l)}(u)$ using a uniform random subsample of patch locations within each layer.

## C.3. Experimental Details for Randomness Test

This appendix describes the experimental protocol used to produce Figure 2(a–c), including the sources of the pretrained models, the procedures for selecting and preprocessing weight matrices, and the hyperparameters utilized in each experiment.

### C.3.1. PRETRAINED MODELS AND SOURCES

We study representative pretrained architectures, including MLPs, convolutional networks, and Transformers, to probe whether the observed phenomena are consistent across model families. Specifically, we utilize MLP-Mixer-B16 with pretrained weights from `timm`, ResNet18 with pretrained weights from `torchvision`, and TinyLlama-1.1B-Chat from `HuggingFace Transformers`. To ensure numerical stability and consistent linear-algebra behavior across libraries, we cast all extracted matrices to `float32` before performing SVD computations.

### C.3.2. WEIGHT-MATRIX EXTRACTION AND LAYER FILTERING

For each model, we iterate over the modules and extract two-dimensional internal weight tensors, converting each into a matrix $W \in \mathbb{R}^{m \times n}$. For linear layers, we use the native weight matrix directly. For Conv2d layers, we flatten the convolutional kernel into a matrix with $m$ representing the number of output channels and $n$ denoting the product of the number of input channels, the kernel height, and the kernel width. To focus on internal transformation matrices in

Transformer models, we exclude tensors with parameter names containing embed or lm_head, thereby omitting token embeddings and the output head, which may exhibit different statistical regularities.

To control runtime while emphasizing the most informative tensors, we prioritize larger matrices by sorting all candidates according to $d = \min(m, n)$ in descending order. We then limit the number of selected matrices per model based on the requirements of each panel: for Figure 2(a), we use up to 40 matrices per model to obtain smooth average curves, whereas for Figure 2(b), we use up to 6 matrices per model because each layer contributes additional submatrix sampling and spectral computations.

### C.3.3. SIGN–MAGNITUDE DECOMPOSITION AND THE SIGN CONVENTION

Given a weight matrix $W \in \mathbb{R}^{m \times n}$, we perform a decomposition.

$$S := \text{sign}(W), \quad A := |W|, \quad \text{so that} \quad W = S \odot A.$$

Throughout the paper, we use the convention $\text{sign}(0) = +1$, ensuring that $S \in \{\pm 1\}^{m \times n}$ is always maintained. Even if $\text{sign}(0) = +1$, we have $W = S \odot A = 0$ when $A = 0$, so this tie-break convention does not affect the reported statistics.

### C.3.4. FIGURE 2(A): SVD COMPRESSIBILITY ERROR VS. RANK RATIO

For a matrix $M \in \{W, S, A\}$, let $\sigma_1^{\text{SVD}}(M) \geq \cdots \geq \sigma_d^{\text{SVD}}(M) \geq 0$ denote its singular values ($d = \min(m, n)$). The optimal rank-$r$ approximation error in the Frobenius norm can be expressed as

$$E_r(M) := \frac{\|M - M_r\|_F}{\|M\|_F} = \sqrt{\frac{\sum_{i>r} \sigma_i^{\text{SVD}}(M)^2}{\sum_{i \geq 1} \sigma_i^{\text{SVD}}(M)^2}}.$$

We report $E_r(W)$, $E_r(S)$, and $E_r(A)$ as functions of the **rank ratio** $q := r/d \in (0, 1]$. In our plots, we vary $q$ over a log-spaced grid implementation.

$$q \in \left\{ \frac{1}{2048}, \frac{2}{2048}, \frac{4}{2048}, \ldots, \frac{1024}{2048} \right\}, \quad r = \lfloor qd \rfloor, \quad r \in [1, d],$$

and average $E_r(\cdot)$ over selected layers within each model.

**Exact vs. randomized SVD.** When $d \leq 2048$, we compute the singular values precisely. For larger matrices, we utilize a randomized low-rank SVD routine (e.g., torch.svd_lowrank) to estimate the tail energy necessary for evaluating $E_r(M)$ efficiently at the ranks in our sweep.

### C.3.5. FIGURE 2(B): SPECTRAL FIT TO A RADEMACHER BASELINE (KS TEST)

Figure 2(b) tests whether sign matrices are spectrally consistent with an i.i.d. Rademacher random matrix. For each selected sign matrix $S \in \{\pm 1\}^{m \times n}$, we set $d = \min(m, n)$ and choose $s = \min(256, d)$. If $s < 32$ for a layer, we exclude it from Figure 2(b).

**Submatrix sampling.** For each layer, we sample $N_{\text{sub}} = 20$ contiguous submatrices $S_{\text{sub}} \in \mathbb{R}^{s \times s}$ by drawing uniform offsets $(i_0, j_0)$ and taking the block slice $S[i_0 : i_0 + s, \ j_0 : j_0 + s]$.

**Normalization and pooling.** For each sampled submatrix, we compute the singular values and normalize them by $\sqrt{s}$ to eliminate trivial scaling:

$$\tilde{\sigma}_i^{\text{SVD}} := \sigma_i^{\text{SVD}}(S_{\text{sub}})/\sqrt{s}.$$

We pool $\tilde{\sigma}_i^{\text{SVD}}$ across sampled submatrices and layers to form an empirical CDF $F_{\text{sign}}$. We construct the baseline CDF $F_{\text{rand}}$ by repeating the same procedure on i.i.d. Rademacher matrices $R \in \{\pm 1\}^{s \times s}$ with equal probability.

**Two-sample KS statistic.** We report the two-sample Kolmogorov–Smirnov statistic

$$D := \sup_x \big| F_{\text{sign}}(x) - F_{\text{rand}}(x) \big|,$$

and optionally the corresponding $p$-value from the KS test implementation.

C.3.6. FIGURE 2(C): TRANSFORMER LM INIT-SIGN DRIFT ON A SYNTHETIC LANGUAGE TASK

Figure 2(c) measures how much the sign pattern deviates from initialization during scratch training of a multi-layer Transformer language model (LM) on a simple next-token prediction task.

**Task (synthetic next-token prediction).** We generate a synthetic token corpus with vocabulary size $V = 256$ and sequence length $L = 128$. Training uses teacher forcing: given input tokens $(x_1, \ldots, x_L)$, the target is $(x_2, \ldots, x_{L+1})$. We use 4096 sequences and mini-batches of size 64.

**Model.** We employ a causal Transformer language model with $N_{\text{layer}} = 12$ Transformer blocks, a model width of $d_{\text{model}} = 256$, $h = 4$ attention heads, a feed-forward hidden size of $d_{\text{ff}} = 1024$, and a dropout set to 0.0. Each block features explicit linear projections for $q/k/v/o$ and a two-layer MLP with GELU nonlinearity.

**Optimization.** We train for 5000 steps with Adam, learning rate $3 \times 10^{-4}$, and cross-entropy loss. We log metrics every 50 steps.

**Sign-flip (mismatch) ratio vs. initialization.** For a tracked parameter tensor $W(t)$ during training step $t$, define

$$\text{flip}(t) := 1 - \frac{1}{N} \sum_{i=1}^{N} \mathbf{1}\{\text{sign}(w_i(t)) = \text{sign}(w_i(0))\},$$

where $N$ denotes the count of entries in the tensor. We report $\text{flip}(t)$ for the token embedding weights, for each Transformer block after averaging the linear weight tensors within that block, for the LM head weights, and for a pooled average over all tracked tensors; the corresponding curves are shown in Figure 2(c).

## C.4. Experimental Details for Language Task

C.4.1. DATASET AND MODEL

This appendix summarizes the settings used for the CharLM sign lock-in experiments.

**Data.** Tiny Shakespeare; contiguous 90/10 train/validation split; character vocabulary from the corpus.

**Task, batching, and evaluation.** Next-character prediction with teacher forcing on random contiguous blocks. We use sequence length $L = 64$ and batch size $B = 64$. Validation loss is averaged over 20 randomly sampled validation mini-batches; PPL $= \exp(\text{val\_loss})$ (loss in nats).

**Model.** TinyCharLM: causal Transformer with $d_{\text{model}} = 128$, $n_{\text{layers}} = 2$, $n_{\text{heads}} = 4$, $d_{\text{ff}} = 256$, maximum context length 256, and no dropout.

**Optimization and regularization.** AdamW; baseline initialization scale $\sigma_{\text{init}} = 0.02$. When $\lambda > 0$, add the log-barrier regularizer as in Section 4 (held constant for the first half of training, then linearly decayed to 0).

**Initial weights.**

- **Baseline:** GPT-like initialization.

- **Proposed method:** apply the low-rank sign template (Section E.4.1, Eq. (21)) and gap initialization to *all* matrix-shaped parameters, with template rank $r = 2$ and a fixed global seed 1234.

**Training horizons and learning rates.** All results use three seeds (0/1/2).

- **Sign lock-in validation** $T = 2000$ steps. Tail plot sweeps learning rate in $\{10^{-4}, 3 \cdot 10^{-4}, 5 \cdot 10^{-4}, 10^{-3}\}$; the histogram uses $3 \cdot 10^{-4}$.

- **Sign lock-in enhancement:** $T = 12000$ steps, learning rate $3 \cdot 10^{-4}$, and $a_{\text{init}} \in \{0.001, 0.005, 0.02, 0.03, 0.05\}$, $\lambda \in \{10^{-4}, 10^{-3}, 10^{-2}, 0.05, 0.1, 0.3, 0.5\}$.

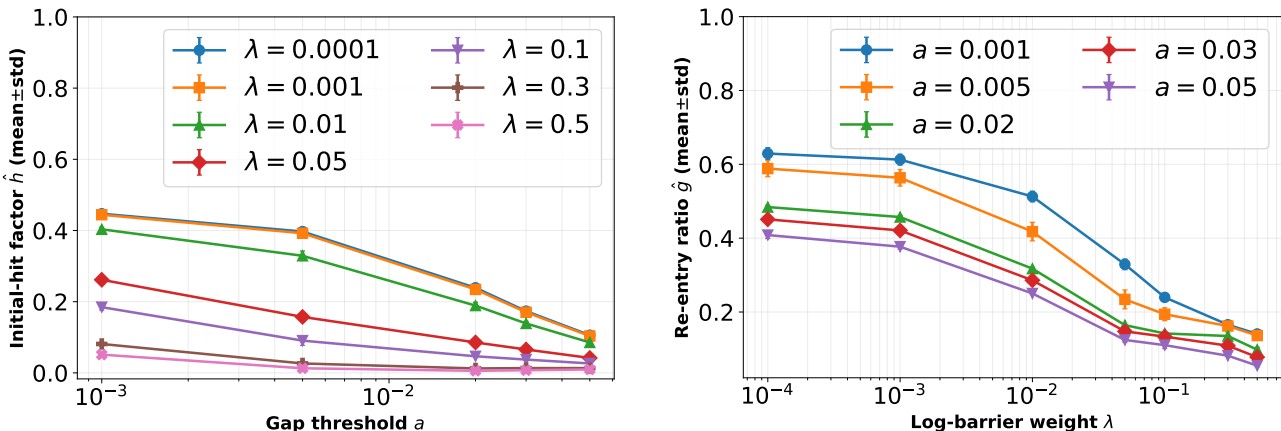

*Figure 9.* **Estimated lock-in parameters over gap-init and regularization. Left:** $\hat{h}$ (initial-hit factor) as a function of the gap threshold $a_{\text{init}}$ for different log-barrier weights $\lambda$. **Right:** $\hat{g}$ (re-entry ratio) as a function of $\lambda$ (log scale) for different $a_{\text{init}}$. Points show mean±std over three seeds; see Appendix C.4 for the full hyperparameters (including $T$, $\rho$, and $\epsilon$).

### C.4.2. ANALYSIS TOOLS

**Lock-in statistics.** We track all matrix-shaped parameters (dim$\geq 2$). For $K_T^{\text{eff}}(\rho)$, we use $\rho = 10^{-3}$ and $\epsilon = 10^{-4}$.

**Zero-inflated geometric fit.** We model the empirical distribution of $K := K_T^{\text{eff}}(\rho)$ by a zero-inflated geometric distribution:

$$\mathbb{P}[K = 0] = 1 - h, \quad \mathbb{P}[K = k] = h(1-g)g^{k-1} \quad (k \geq 1).$$

The maximum-likelihood estimates are $\hat{h} = \mathbb{P}[K > 0]$ and $\hat{g} = 1 - 1/\mathbb{E}[K \mid K > 0]$ (with $\hat{g} = 0$ when $\mathbb{E}[K \mid K > 0] \leq 1$). We interpret $\hat{h}$ as the *initial-hit* factor and $\hat{g}$ as the *re-entry* propensity.

**Step-wise flip-rate proxy `flip_mean`.** We report the step-wise flip-rate estimation `flip_mean`:

$$\text{flip\_mean} := \frac{1}{T} \sum_{t=0}^{T-1} \left( \frac{1}{N} \sum_{i=1}^{N} \mathbf{1}\{\text{sign}(w_i(t+1)) \neq \text{sign}(w_i(t))\} \right).$$

Here $w_i(t)$ is the $i$-th tracked scalar at step $t$, and $N$ is the number of tracked scalars.

**SVD method.** This setting is the same as Appendix C.3.

**Example of weight trajectory.** To illustrate the underlying mechanism, Figure 10 shows representative 1D weight trajectories under baseline, gap-dominant, and strongly regularized regimes, together with the outer/boundary bands. The regularized regime keeps trajectories away from the boundary neighborhood, reducing repeated re-entry events.

### C.4.3. ADDITIONAL FIGURES FOR THE LANGUAGE TASK

**Lock-in parameter control.** We quantify how the lock-in parameters vary with enhancement: gap initialization (threshold $a_{\text{init}}$) and log-barrier outer-drift regularization (weight $\lambda$). For each $(a_{\text{init}}, \lambda)$, we fit a zero-inflated geometric model to the empirical per-weight distribution of $K_T^{\text{eff}}(\rho)$ and extract $(\hat{h}, \hat{g})$. Figure 9 shows the fitted $(\hat{h}, \hat{g})$ over the sweep. As predicted by Proposition D.27, increasing the gap threshold $a_{\text{init}}$ reduces $\hat{h}$, indicating fewer initial boundary hits. Moreover, increasing the log-barrier weight $\lambda$ reduces $\hat{g}$ (Proposition D.28), consistent with suppressed re-entry into the boundary neighborhood once weights have moved into the outer region. Together, these results support the interpretation that gap initialization controls $h_T$ while outer-drift regularization controls $g_T$.

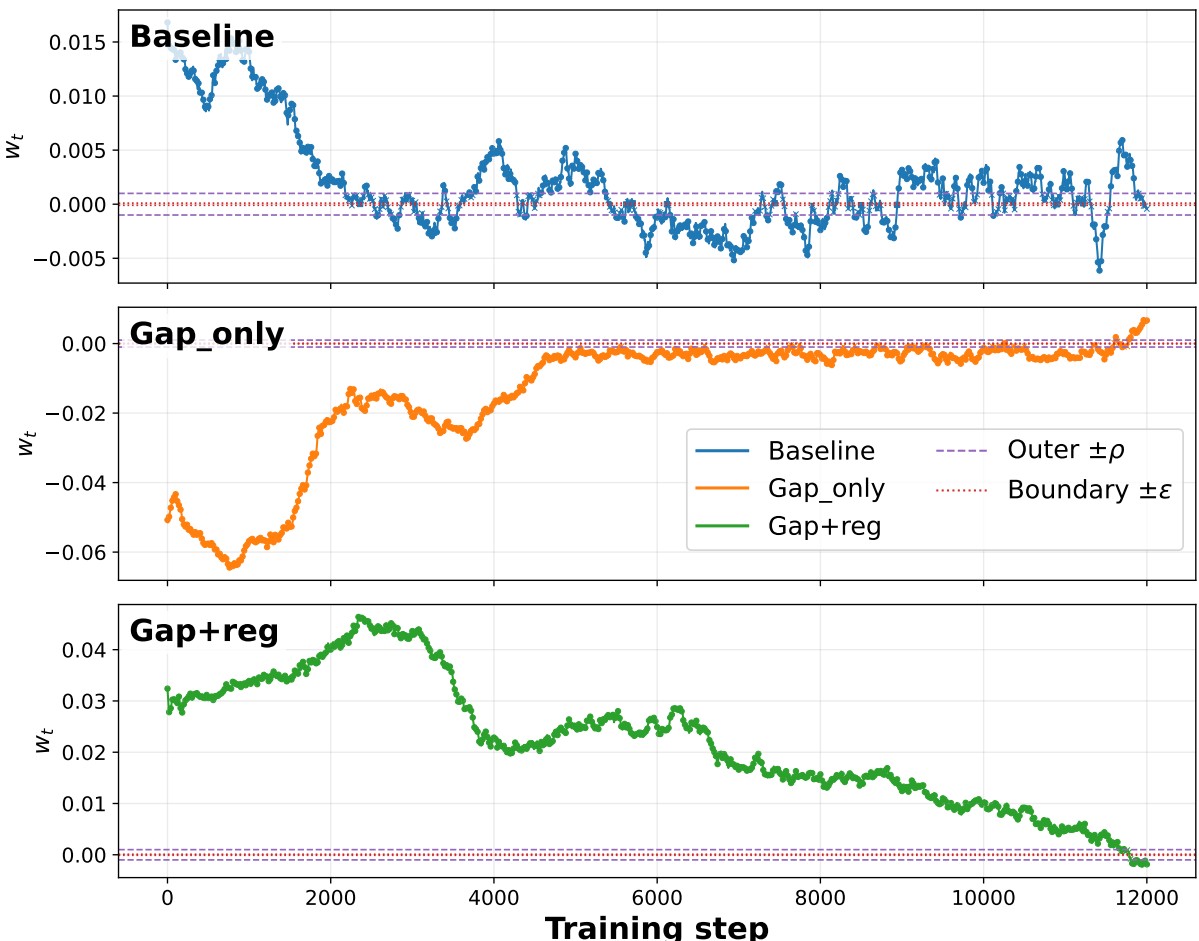

*Figure 10.* **Representative one-dimensional weight trajectories (down sampled for printing).** We plot a representative 1D weight trajectory (down-sampled) $w_t$ under baseline ($a_{\mathrm{init}}$=0, $\lambda$=0), a gap-dominant setting, and a strongly regularized setting, together with the outer/boundary bands used to define $(\sigma_k, \tau_k)$ and $K_T^{\mathrm{eff}}(\rho)$. Regularization keeps trajectories away from the boundary neighborhood, suppressing re-entry.

## C.5. Billion-Scale LLM Validation

This appendix reports a parameter-scale sweep of the lock-in parameters $(\hat{h}, \hat{g})$, estimated via the zero-inflated geometric fit described in Appendix C.4.2.

**Data.**  Tiny Shakespeare; contiguous 90/10 train/validation split; character vocabulary from the corpus.

**Task and optimizer.**  Next-character prediction with teacher forcing on random contiguous blocks. We use sequence length $L = 64$ and training micro-batch size $B = 1$. Each configuration is trained for $T = 1000$ optimizer steps.

**Model.**  Char-level causal Transformer LM with learned token embeddings and learned absolute positional embeddings of length $L = 64$ (maximum context length 64), and no dropout. Each Transformer block uses pre-LayerNorm, SDPA attention (PyTorch `scaled_dot_product_attention` with is_causal=True, dropout probability 0), and a 2-layer MLP with GELU. For each configuration we set: (i) $d_{\mathrm{ff}} = 4\, d_{\mathrm{model}}$, (ii) the head dimension fixed to 64 by using $n_{\mathrm{heads}} = d_{\mathrm{model}}/64$ (when divisible). To reduce activation memory, we enable per-block gradient checkpointing. Weights are stored in bf16; LayerNorm parameters and computations are kept in fp32 for stability. The model configurations are reported in Table 1.

**Optimization and regularization.**  We optimize the cross-entropy loss using AdamW with an 8-bit optimizer state (`bitsandbytes` AdamW8bit) to reduce memory. We use a constant learning rate $\eta = 3 \times 10^{-4}$ for $T = 1000$ steps, weight

*Table 1.* **Parameter-scale sweep.** We report the configurations of Transformers

| Model | Params (B) | $d_{\mathrm{model}}$ | Layers | Heads |
|-------|-----------|----------------------|--------|-------|
| 31M   | 0.032     | 512   | 10 | 8  |
| 64M   | 0.064     | 768   | 9  | 12 |
| 126M  | 0.126     | 1024  | 10 | 16 |
| 260M  | 0.260     | 1344  | 12 | 21 |
| 399M  | 0.399     | 1664  | 12 | 26 |
| 604M  | 0.604     | 2048  | 12 | 32 |
| 1.2B  | 1.209     | 2048  | 24 | 32 |
| 1.8B  | 1.813     | 2048  | 36 | 32 |
| 2.4B  | 2.417     | 2048  | 48 | 32 |
| 5.4B  | 5.437     | 3072  | 48 | 48 |
| 9.7B  | 9.665     | 4096  | 48 | 64 |
| 12.9B | 12.887    | 4096  | 64 | 64 |

decay 0.01, and global-norm gradient clipping at 1.0. We use bf16 autocast for the forward pass. Unless otherwise specified by the library defaults, AdamW uses $\beta_1 = 0.9$, $\beta_2 = 0.999$, and $\epsilon = 10^{-8}$. We do not use dropout or additional regularizers in this sweep.

**Initialization.** All Linear and Embedding weights are initialized i.i.d. from $\mathcal{N}(0, \sigma_{\mathrm{init}}^2)$ with $\sigma_{\mathrm{init}} = 0.02$; positional embeddings are initialized to zero.

**Lock-in statistics.** We track matrix-shaped parameters (dim$\geq 2$), using the tie-break convention $\mathrm{sign}(0) = +1$. To keep memory bounded at large scale, for each tracked tensor we sample 2048 scalar coordinates uniformly at random (fixed seed) and compute the effective flip count $K := K_{\mathrm{eff},T}(\rho)$ on these coordinates with $\rho = 10^{-3}$ and $\epsilon = 10^{-4}$. We fit the zero-inflated geometric model and report the MLE $\hat{h} = \mathbb{P}[K > 0]$ and $\hat{g} = 1 - 1/\mathbb{E}[K \mid K > 0]$ (with $\hat{g} = 0$ when $\mathbb{E}[K \mid K > 0] \leq 1$).

**Evaluation.** Figure 4 shows that both estimated initial-hit factor $\hat{h}$ and re-entry ratio $\hat{g}$ decrease monotonically with model scale. As the parameter count grows from tens of millions to ∼10B parameters, the estimated initial-hit factor $\hat{h}$ drops by nearly an order of magnitude, indicating that an increasingly small fraction of weights ever approach the near-zero boundary. At the same time, the re-entry ratio $\hat{g}$ also decreases and remains close to zero for the largest models, suggesting that once a weight exits the boundary neighborhood, repeated re-entries are exceedingly rare. These trends imply that sign lock-in strengthens with scale: larger models exhibit fewer initial boundary hits and substantially lower probability of repeated sign flips. This observation is consistent with the width-scaling predictions of the sign lock-in theory, and suggests that extreme-scale models naturally operate deeper in the lock-in regime.

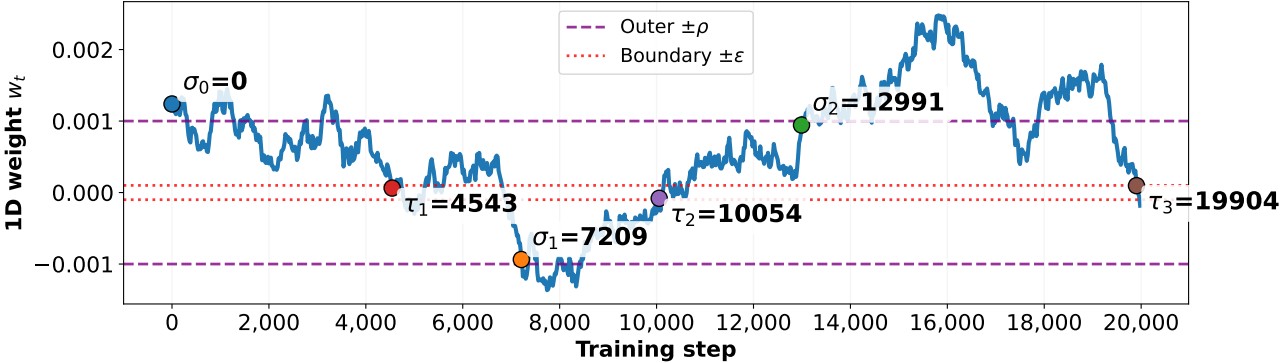

*Figure 11.* **Sign lock-in trajectory and stopping times.** We illustrate the one-dimensional weight trajectory (down-sampled) $w_t$ together with the outer region and the sign-boundary neighborhood used in the sign lock-in theory. The outer region is defined as $\{|w_t| \geq \rho\}$ and the boundary neighborhood as $\{|w_t| \leq \epsilon\}$. Starting from an outer-entry time $\sigma_0$ (outer region, positive sign), the trajectory hits the boundary neighborhood at $\tau_1$ and exits to the opposite outer side at $\sigma_1$ (negative sign), then returns to the boundary at $\tau_2$ and re-enters the original outer side at $\sigma_2$ (positive sign). This outer-to-outer sign evolution is counted by the effective flip count $K_T^{\text{eff}}(\rho)$.

## D. Proofs and Extensions of Sign Lock-In Theory

This appendix provides a self-contained theoretical analysis underlying the sign lock-in phenomenon introduced in Section 3. For completeness, we restate the stopping-time construction and clarify how each part of this appendix connects to the main theoretical statements.

Specifically:

- Appendix D.1 provides a complete proof of Theorem 3.6 based solely on Assumptions 3.3 and 3.4.

- Appendix D.2 details initial hit factor and sign flips results under the bounded update condition.

- Appendix D.3 derives a schedule-aware SGD-based sufficient condition for Assumption 3.4 related to re-entry ratio, we note that this is a representative proof of SGD family optimizers.

- Appendix D.4 analyzes implications for learning-rate schedules, batch size, and model width.

- Appendix D.5 studies gap initialization and outer-drift regularization.

- Appendix D.6 discusses a contrasting sign floating regime in the extreme experimental setting.

**Stopping times and regions (restated).** We briefly restate the stopping-time construction used throughout this appendix, so that the arguments below can be read independently of the main text. Let $(w_t)_{t \geq 0}$ be a real-valued adapted process, and fix constants $0 < \epsilon < \rho$ and a finite horizon $T$.

We define the *outer region* by $|w_t| \geq \rho$ and the *boundary neighborhood* by $|w_t| \leq \epsilon$. Starting from the first time the trajectory enters the outer region, we define a sequence of stopping times recursively as follows. Let

$$\sigma_0 := \inf\{t \geq 0 : |w_t| \geq \rho\},$$

and for $k \geq 1$,

$$\tau_k := \inf\{t > \sigma_{k-1} : |w_t| \leq \epsilon\}, \qquad \sigma_k := \inf\{t > \tau_k : |w_t| \geq \rho\}.$$

These stopping times decompose the trajectory into excursions between the outer region and the boundary neighborhood and form the basis of all proofs in Appendix D.1. The image of the region and stopping time setting is illustrated in Figure 11. Throughout the sign lock-in theory and its proofs, we assume the following bounded-update condition.

**Assumption 3.3 (restated).** Fix $T \in \mathbb{N}$. There exist deterministic constants $\Delta > 0$ and $\delta_{\mathrm{upd}} \in [0, 1)$ such that, for every stopping time $\theta$ satisfying $\theta \leq T - 1$,

$$\mathbb{P}\left[\mathcal{E}_\Delta(\theta)|\mathcal{F}_\theta\right] \geq 1 - \delta_{\mathrm{upd}} \quad \text{a.s.,} \tag{1}$$

where the *good event* $\mathcal{E}_\Delta(\theta)$ is

$$\mathcal{E}_\Delta(\theta) := \left\{ \max_{\theta \leq t \leq T-1} |w_{t+1} - w_t| \leq \Delta \right\}.$$

In particular, on $\mathcal{E}_\Delta := \mathcal{E}_\Delta(0)$ we have $|w_{t+1} - w_t| \leq \Delta$ for all $0 \leq t \leq T - 1$, and the special case $\delta_{\mathrm{upd}} = 0$ yields an almost-sure uniform increment bound.

## D.1. Proofs for Sign Lock-In Theory

In the proof of Theorem 3.6, we assume the following Re-entry bound condition. We show in Appendix D.3 that this assumption is justified under standard SGD settings. Moreover, although we do not provide an explicit proof, similar arguments apply to SGD-family optimizers such as Adam. Therefore, this assumption is well justified under typical training settings.

**Roadmap of the proof.** To keep Appendix D easy to follow, we first show in Lemma D.1 that an outer-to-outer sign change necessarily passes through the boundary neighborhood. Next, Lemma D.2 lifts this pathwise fact to an event-level statement: frequent effective flips imply frequent boundary hits. Finally, Theorem 3.6 combines Assumption 3.4 with this implication to obtain a geometric tail bound. Overall, the logic is flip $\Rightarrow$ boundary hit $\Rightarrow$ geometric decay.

**Assumption 3.4 (restated).** There exists $g_T \in (0, 1)$ such that for all $k \geq 0$,

$$\mathbb{P}[\tau_{k+1} \leq T|\mathcal{F}_{\sigma_k}] \leq g_T \quad \text{a.s. on } \{\sigma_k \leq T\}.$$

**Lemma D.1** (Outer-to-outer sign flip forces a boundary visit). *On the good event $\mathcal{E}_\Delta$ from Assumption 3.3, on $\{\sigma_{k-1} \leq T, \sigma_k \leq T\}$, if $\mathrm{sign}(w_{\sigma_k}) \neq \mathrm{sign}(w_{\sigma_{k-1}})$ then $\tau_k < \sigma_k$ and in particular $\tau_k \leq T$.*

*Proof.* Work on the good event $\mathcal{E}_\Delta$ (Assumption 3.3). Assume $\mathrm{sign}(w_{\sigma_k}) \neq \mathrm{sign}(w_{\sigma_{k-1}})$ on $\{\sigma_{k-1} \leq T, \sigma_k \leq T\}$. Then $w_{\sigma_{k-1}}$ and $w_{\sigma_k}$ have opposite signs with $|w_{\sigma_{k-1}}| \geq \rho$ and $|w_{\sigma_k}| \geq \rho$. Without loss of generality, take $w_{\sigma_{k-1}} \geq \rho > 0$ and $w_{\sigma_k} \leq -\rho < 0$. Let

$$t^\star := \inf\{t > \sigma_{k-1} : w_t \leq 0\}.$$

Since $w_{\sigma_{k-1}} > 0$ and $w_{\sigma_k} < 0$, we have $t^\star \leq \sigma_k \leq T$. That is, $t^\star$ is the first post-$\sigma_{k-1}$ contact with the sign boundary.

Moreover, $t^\star < \sigma_k$: if $t^\star = \sigma_k$, then $w_{(\sigma_k-1)} > 0$ and $w_{\sigma_k} \leq -\rho$, so $|w_{\sigma_k} - w_{(\sigma_k-1)}| \geq \rho$. But on $\mathcal{E}_\Delta$ we have $|w_{\sigma_k} - w_{(\sigma_k-1)}| \leq \Delta < \rho$ (as $\epsilon = \max\{\epsilon_0, \Delta\} < \rho$), a contradiction.

By definition, $w_{(t^\star-1)} > 0$ and $w_{t^\star} \leq 0$, hence

$$|w_{t^\star}| = -w_{t^\star} \leq w_{(t^\star-1)} - w_{t^\star} = |w_{t^\star} - w_{(t^\star-1)}| \leq \Delta \leq \epsilon \quad (\text{on } \mathcal{E}_\Delta).$$

Therefore $|w_{t^\star}| \leq \epsilon$, and by the definition of $\tau_k$ we get $\tau_k \leq t^\star < \sigma_k$. In particular, $\tau_k \leq T$. $\square$

**Lemma D.2** (Many effective flips imply many boundary hits). *For all $k \geq 1$,*

$$\{K_T^{\mathrm{eff}}(\rho) \geq k\} \subseteq \{\tau_k \leq T\} \cup \mathcal{E}_\Delta^c.$$

*Consequently,*

$$\mathbb{P}[K_T^{\mathrm{eff}}(\rho) \geq k] \leq \mathbb{P}[\tau_k \leq T] + \delta_{\mathrm{upd}}.$$

*Proof.* On the good event $\mathcal{E}_\Delta$, Lemma D.1 holds. If $K_T^{\mathrm{eff}}(\rho) \geq k$, then there exist $k$ indices $1 \leq j_1 < \cdots < j_k$ such that $\sigma_{j_m} \leq T$ and $\mathrm{sign}(w_{\sigma_{j_m}}) \neq \mathrm{sign}(w_{\sigma_{j_m-1}})$ for each $m$. In other words, we can explicitly list $k$ distinct outer-to-outer sign-change episodes before time $T$. Fix such an index $j = j_m$. Since $(\sigma_l)$ is nondecreasing, we have $\sigma_{j-1} \leq \sigma_j \leq T$, so on $\mathcal{E}_\Delta$ Lemma D.1 applies and yields $\tau_j \leq T$. Because $(\tau_l)$ is nondecreasing and $j_k \geq k$, we have $\tau_k \leq \tau_{j_k} \leq T$.

Thus we have shown $\{K_T^{\mathrm{eff}}(\rho) \geq k\} \cap \mathcal{E}_\Delta \subseteq \{\tau_k \leq T\}$, equivalently $\{K_T^{\mathrm{eff}}(\rho) \geq k\} \subseteq \{\tau_k \leq T\} \cup \mathcal{E}_\Delta^c$. Taking probabilities and using $\mathbb{P}(\mathcal{E}_\Delta^c) \leq \delta_{\mathrm{upd}}$ gives Lemma D.2. $\square$

**Theorem 3.6 (restated).** Under Assumptions 3.3–3.4. Define the effective outer-to-outer flip count up to time $T$ by

$$K_T^{\text{eff}}(\rho) := \sum_{k \geq 1} \mathbf{1}\Big[\sigma_k \leq T, \ \text{sign}(w_{\sigma_k}) \neq \text{sign}(w_{\sigma_{k-1}})\Big], \tag{2}$$

which is well-defined since $|w_{\sigma_k}| \geq \rho > 0$ on $\{\sigma_k \leq T\}$. Let $h_T := \mathbb{P}[\tau_1 \leq T] \in [0,1]$. Then for all integers $k \geq 1$,

$$\mathbb{P}[\tau_k \leq T] \leq h_T \, g_T^{k-1}, \tag{3}$$

and consequently,

$$\mathbb{P}\big[K_T^{\text{eff}}(\rho) \geq k\big] \leq h_T g_T^{k-1} + \delta_{\text{upd}}. \tag{4}$$

*Proof.* We prove Eq. (3) by induction. For $k = 1$, Eq. (3) reduces to $\mathbb{P}[\tau_1 \leq T] = h_T$.

Assume $\mathbb{P}[\tau_k \leq T] \leq h_T \, g_T^{k-1}$ for some $k \geq 1$. Then, by the tower property and Assumption 3.4, we first note that $\tau_{k+1} = \inf\{t > \sigma_k : |w_t| \leq \epsilon\}$ implies $\{\tau_{k+1} \leq T\} \subseteq \{\sigma_k \leq T\}$. Therefore

$$\begin{aligned}
\mathbb{P}[\tau_{k+1} \leq T] &= \mathbb{E}\big[\mathbb{P}[\tau_{k+1} \leq T \mid \mathcal{F}_{\sigma_k}] \, \mathbf{1}_{\{\sigma_k \leq T\}}\big] \\
&\leq \mathbb{E}\big[g_T \, \mathbf{1}_{\{\sigma_k \leq T\}}\big] = g_T \, \mathbb{P}[\sigma_k \leq T] \leq g_T \, \mathbb{P}[\tau_k \leq T] \\
&\leq g_T \cdot h_T \, g_T^{k-1} = h_T \, g_T^k.
\end{aligned}$$

Here we used $\mathbb{P}[\sigma_k \leq T] \leq \mathbb{P}[\tau_k \leq T]$, which follows from the stopping-time order $\tau_k < \sigma_k$ whenever $\sigma_k < \infty$.

The bound for $K_T^{\text{eff}}(\rho)$ follows from Lemma D.2, which introduces the additional failure probability term $\delta_{\text{upd}}$ under Assumption 3.3. $\square$

## D.2. Initial Hit Factor and Deterministic Foundations of Sign Flips

This subsection isolates a deterministic prerequisite for sign changes that follows solely from the bounded-update event in Assumption 3.3. In particular, Proposition 3.8 shows that any deviation of the final sign from the initialization sign forces the trajectory to enter the $\Delta$-band at least once, so sign drift can be upper bounded by a boundary-hit probability. We then lift this implication to the vector setting via Proposition D.3, which bounds the initialization-to-final mismatch rate $p_T^{\text{init}}$ by the average of per-coordinate $\Delta$-band hit probabilities, up to the failure term $\delta_{\text{upd}}$ from Assumption 3.3. Finally, Proposition D.4 turns this boundary-hit viewpoint into an initialization-dominated control of the initial-hit factor $h_T$ using the boundary radius in Definition 3.1. These deterministic foundations complement the excursion and re-entry analysis that controls repeated effective flips, and they will be used later when we design interventions that suppress boundary visits.

**Proposition 3.8 (restated).**

Assume the bounded-update condition. Let $(w_t)_{t=0}^T$ be a real-valued process and define $s_t := \text{sign}(w_t)$ with a fixed tie-breaking rule at zero. Define the first hitting time of the $\Delta$-band by

$$\tau_\Delta := \inf\{t \in \{0, 1, \ldots, T\} : |w_t| \leq \Delta\},$$

with the convention $\inf \varnothing = \infty$. Then the following deterministic implication holds:

$$\mathbf{1}\{s_T \neq s_0\} \ \leq \ \mathbf{1}\{\tau_\Delta \leq T\}.$$

Consequently,

$$\mathbb{P}(s_T \neq s_0) \ \leq \ \mathbb{P}(\tau_\Delta \leq T) + \delta_{\text{upd}}.$$

*Proof.* Work on the event $\mathcal{E}_\Delta$ and assume that $s_T \neq s_0$. Define the first instance where the sign differs from initialization:

$$t^\star := \min\{t \in \{1, 2, \ldots, T\} : s_t \neq s_0\}.$$

Then $s_{t^\star - 1} = s_0$ and $s_{t^\star} \neq s_0$, hence $s_{t^\star} \neq s_{t^\star - 1}$. With a fixed tie-break at 0, $s_{t^\star} \neq s_{t^\star - 1}$ implies $w_{t^\star} w_{t^\star - 1} \leq 0$ (opposite signs or one is 0). Therefore,

$$|w_{t^\star} - w_{t^\star - 1}| = |w_{t^\star}| + |w_{t^\star - 1}|.$$

On $\mathcal{E}_\Delta$, we also have $|w_{t^\star} - w_{t^\star - 1}| \leq \Delta$, hence

$$|w_{t^\star}| + |w_{t^\star - 1}| \leq \Delta,$$

which implies $\min\{|w_{t^\star}|, |w_{t^\star - 1}|\} \leq \Delta$. Thus there exists $t \in \{t^\star - 1, t^\star\}$ such that $|w_t| \leq \Delta$, so $\tau_\Delta \leq t^\star \leq T$. This proves $\mathbf{1}\{s_T \neq s_0\} \leq \mathbf{1}\{\tau_\Delta \leq T\}$ on $\mathcal{E}_\Delta$. Finally,

$$\mathbb{P}[s_T \neq s_0] \leq \mathbb{P}[\tau_\Delta \leq T] + \mathbb{P}[\mathcal{E}_\Delta^c] \leq \mathbb{P}[\tau_\Delta \leq T] + \delta_{\text{upd}},$$

where the last inequality employs Assumption 3.3 at $\theta = 0$. $\qquad\square$

**Proposition D.3** (Initialization-to-final sign drift bound). *Assume Assumption 3.3. Let $w_t \in \mathbb{R}^N$ be the parameter vector and define $s_t(i) := \text{sign}(w_t(i)) \in \{\pm 1\}$ entrywise with a fixed tie-break at 0. Let*

$$p_T^{\text{init}} := \frac{1}{N} \sum_{i=1}^N \mathbf{1}\left[s_T(i) \neq s_0(i)\right] \in [0, 1].$$

*Define the per-coordinate first hit time of the $\Delta$-band by*

$$\tau_{\Delta,i} := \inf\{t \in \{0, 1, \ldots, T\} : |w_t(i)| \leq \Delta\} \in \{0, 1, \ldots, T\} \cup \{\infty\},$$

*with $\inf \emptyset = \infty$. Then:*

*(A) On the event $\mathcal{E}_\Delta := \{\max_{0 \leq t \leq T-1} \|w_{t+1} - w_t\|_\infty \leq \Delta\}$,*

$$\mathbf{1}\left[s_T(i) \neq s_0(i)\right] \leq \mathbf{1}\left[\tau_{\Delta,i} \leq T\right] \quad \text{for all } i \in [N].$$

*Consequently,*

$$p_T^{\text{init}} \leq \frac{1}{N} \sum_{i=1}^N \mathbf{1}\left[\tau_{\Delta,i} \leq T\right] \quad \text{on } \mathcal{E}_\Delta.$$

*(B) Let $\bar{h}_T := \frac{1}{N} \sum_{i=1}^N \mathbb{P}[\tau_{\Delta,i} \leq T]$. Then*

$$\mathbb{E}[p_T^{\text{init}}] \leq \bar{h}_T + \delta_{\text{upd}}.$$

*Proof.* (A) Fix an index $i \in [N]$ and work on the event $\mathcal{E}_\Delta$. Assume that $s_T(i) \neq s_0(i)$. Define the first time at which the $i$-th coordinate changes its sign relative to initialization:

$$t^\star := \min\{t \in \{1, 2, \ldots, T\} : s_t(i) \neq s_0(i)\}.$$

Then $s_{t^\star - 1}(i) = s_0(i)$ and $s_{t^\star}(i) \neq s_0(i)$, so $s_{t^\star}(i) \neq s_{t^\star - 1}(i)$. With the fixed tie-break at 0, this implies $w_{t^\star}(i) w_{t^\star - 1}(i) \leq 0$, and hence

$$|w_{t^\star}(i) - w_{t^\star - 1}(i)| = |w_{t^\star}(i)| + |w_{t^\star - 1}(i)|.$$

On the event $\mathcal{E}_\Delta$, we have $|w_{t^\star}(i) - w_{t^\star - 1}(i)| \leq \Delta$, which implies

$$|w_{t^\star}(i)| + |w_{t^\star - 1}(i)| \leq \Delta.$$

Therefore $\min\{|w_{t^\star}(i)|, |w_{t^\star - 1}(i)|\} \leq \Delta$, and hence there exists $t \in \{t^\star - 1, t^\star\}$ such that $|w_t(i)| \leq \Delta$. By definition of $\tau_{\Delta,i}$, this yields $\tau_{\Delta,i} \leq t^\star \leq T$. Thus,

$$\mathbf{1}\{s_T(i) \neq s_0(i)\} \leq \mathbf{1}\{\tau_{\Delta,i} \leq T\} \qquad \text{on } E_\Delta.$$

Averaging over $i$ gives the stated bound for $p_T^{\text{init}}$ on $\mathcal{E}_\Delta$.

(B) By part (A),

$$p_T^{\text{init}} \leq \frac{1}{N} \sum_{i=1}^{N} \mathbf{1}\{\tau_{\Delta,i} \leq T\} \qquad \text{on } E_\Delta.$$

Taking expectations and using $\mathbb{P}(E_\Delta^{\text{c}}) \leq \delta_{\text{upd}}$ from Assumption 3.3, we obtain

$$\mathbb{E}[p_T^{\text{init}}] \leq \frac{1}{N} \sum_{i=1}^{N} \mathbb{P}[\tau_{\Delta,i} \leq T] + \mathbb{P}(E_\Delta^{\text{c}}) \leq \bar{h}_T + \delta_{\text{upd}}.$$

This completes the proof. $\qquad\square$

**Proposition D.4** (Initialization-dominated bound for the initial-hit factor). *Assume Assumption 3.3. Let $\epsilon$ be the boundary radius in Definition 3.1 and define*

$$b_T := \epsilon + T\Delta.$$

*Then the initial-hit factor $h_T := \mathbb{P}[\tau_1 \leq T]$ satisfies*

$$h_T \leq \mathbb{P}\big[|w_0| \leq b_T\big] + \delta_{\text{upd}}. \tag{5}$$

*Proof.* Let $\mathcal{E}_\Delta$ be the bounded-update good event from Assumption 3.3. On $\mathcal{E}_\Delta$, if $\tau_1 \leq T$, then there exists $t \leq T$ such that $|w_t| \leq \epsilon$. By the triangle inequality and bounded updates,

$$|w_0| \leq |w_t| + \sum_{s=0}^{t-1} |w_{s+1} - w_s| \leq \epsilon + T\Delta = b_T.$$

Hence $\{\tau_1 \leq T\} \cap \mathcal{E}_\Delta \subseteq \{|w_0| \leq b_T\}$, which implies Eq. (5). $\qquad\square$

### D.3. An SGD-based Sufficient Condition for Assumption 3.4 That Controls Re-entry Ratio

This appendix provides a typical **sufficient condition** for Assumption 3.4 by combining: (i) bounded increments (Assumption 3.3), (ii) an *expected* descent-lemma argument for scheduled SGD that yields cumulative inward drift control in expectation, and (iii) a variance-aware martingale concentration argument (Freedman's inequality). **Importantly, the proof below does not rely on any SGD-specific property beyond bounded increments and a descent-type energy budget, and thus serves as a template argument for a broad class of SGD-family optimizers as discussed in Remark D.11.**

**Proof sketch (roadmap for Proposition D.10).**   Proposition D.10 provides a schedule-dependent sufficient condition for the re-entry control in Assumption 3.4 under scheduled SGD (Assumption D.5) together with bounded increments (Assumption 3.3). **The crucial point is that, under standard training settings, even if a local inward drift toward the sign boundary exists, it is budget-limited by the finite gradient-energy available for task optimization via telescoping expected descent over a bounded loss range.** One can violate this condition by introducing an artificial and sufficiently strong inward drift, in which case the training dynamics undergo a qualitative transition into the sign floating mode D.6. The logical dependencies among the auxiliary lemmas are:

$$\text{Lemma D.6} \implies \text{Lemma D.7} \implies \big(\text{Lemma D.8 \& Lemma D.9}\big) \implies \text{Proposition D.10.}$$

- **Expected descent $\Rightarrow$ gradient-energy budget.** Lemma D.6 establishes the one-step expected descent inequality (Eq. (6)). Summing and telescoping this inequality over a (possibly random) time window yields Lemma D.7, which upper bounds the conditional gradient energy $\sum_t \eta_t \|\nabla\mathcal{L}(v_t)\|^2$ by schedule-dependent cumulative quantities and the loss range.

- **Gradient-energy budget $\Rightarrow$ drift budget.** Lemma D.8 converts the gradient-energy control in Lemma D.7 into a bound on the cumulative *inward* predictable drift of the oriented coordinate process. In parallel, Lemma D.9 provides a closely related (and proof-wise parallel) bound for the drift-budget proxy $\tilde{U}$, which is tailored for a subsequent Markov-inequality step.

- **Drift-good reduction on an excursion.** In the proof of Proposition D.10, we condition on $\mathcal{F}_{\sigma_k}$ (Definition 3.2) and orient the excursion via $z_t = s_k w_t$. We then decompose the excursion dynamics into predictable drift $d_t$ and martingale noise. We introduce a drift-good event $\mathsf{G}_{\mathrm{drift}}$ ensuring that the total inward drift is at most $B_T^{\mathrm{SGD}}$ (defined in Eq. (7)); Lemma D.9 combined with Markov's inequality yields $\mathbb{P}(\mathsf{G}_{\mathrm{drift}}^c \mid \mathcal{F}_{\sigma_k}) \leq \delta$.

- **Noise control via Freedman + union bound.** On $\mathsf{G}_{\mathrm{drift}}$, reaching the boundary $\{\tau_{k+1} \leq T\}$ can only happen if the martingale noise attains a negative excursion of size roughly $(\rho - \epsilon) - B_T^{\mathrm{SGD}}$. We enforce bounded increments through Assumption 3.3 by stopping at the first update-violation time, and we control the predictable quadratic variation using Eq. (9). Freedman's inequality then yields the exponential term appearing in $g_T^{\mathrm{SGD}}$ (Eq. (7)). Finally, a union bound over (i) update-violation, (ii) drift-bad, and (iii) martingale-deviation events completes the bound in Eq. (8).

Finally, Remark D.11 explains how the two inputs—bounded updates (Assumption 3.3) and re-entry control (Assumption 3.4)—interface with momentum/Adam-type recursions via Eqs. (10)–(11).

**Setup.** Recall the coordinate process $w_t := e_i^\top v_t$ and the stopping times $\sigma_k$ (outer-entry) and $\tau_{k+1}$ (boundary-hit) from Definition 3.2. Fix $k \geq 0$ and condition on $\mathcal{F}_{\sigma_k}$. Let $s_k := \mathrm{sign}(w_{\sigma_k}) \in \{\pm 1\}$ (well-defined on $\{\sigma_k \leq T\}$ since $|w_{\sigma_k}| \geq \rho$), and define the oriented coordinate

$$z_t := s_k\, w_t \;\; (t \geq \sigma_k).$$

On the excursion interval $\{\sigma_k \leq t < \tau_{k+1}\}$ we have $z_t \geq \epsilon$ and $z_{\sigma_k} \geq \rho$. Moreover, for the favorable event $\mathcal{E}_\Delta(\sigma_k)$ from Assumption 3.3, the increments satisfy

$$|z_{t+1} - z_t| = |w_{t+1} - w_t| \leq \Delta \;\; \forall t \in \{\sigma_k, \ldots, T-1\}.$$

*Remark.* Assumption 3.3 can be enforced, for example, by deterministic gradient clipping: if $\|g_t\| \leq G$ almost surely and the step-size schedule satisfies $\eta_t \leq \eta_{\max}$ for all $t$, then one can take $\Delta = \eta_{\max} G$ and $\delta_{\mathrm{upd}} = 0$. More generally, if such a bound holds with high probability uniformly over the horizon, then $\delta_{\mathrm{upd}}$ captures the corresponding failure probability.

**An expected descent assumption for scheduled SGD.** We use a standard smoothness-based expected descent inequality for the (population) objective $\mathcal{L}$ as below.

**Assumption D.5** (Smooth objective and scheduled SGD expected descent). *There exists a differentiable function $\mathcal{L} : \mathbb{R}^d \to \mathbb{R}$ and constants $L_{\mathrm{sm}} > 0$ and $\mathcal{L}_\star \in \mathbb{R}$ such that:*

(i) *($L_{\mathrm{sm}}$-smoothness) $\mathcal{L}$ has $L_{\mathrm{sm}}$-Lipschitz gradient.*

(ii) *(Lower bound) $\mathcal{L}(v) \geq \mathcal{L}_\star$ for all $v$.*

(iii) *(Scheduled SGD step) $v_{t+1} = v_t - \eta_t g_t$ where $(g_t)$ is adapted to $(\mathcal{F}_t)$ and*

$$\mathbb{E}[g_t \mid \mathcal{F}_t] = \nabla\mathcal{L}(v_t), \;\; \mathbb{E}[\|g_t - \nabla\mathcal{L}(v_t)\|^2 \mid \mathcal{F}_t] \leq \xi^2 \;\; \text{for some } \xi \geq 0.$$

(iv) *(Step-size schedule) $(\eta_t)_{t \geq 0}$ is deterministic (or predictable) with $0 \leq \eta_t \leq 1/L_{\mathrm{sm}}$ for all $t$.*

(v) *(Bounded loss on the horizon) there exists $\mathcal{L}_{\max} \in \mathbb{R}$ such that $\mathcal{L}(v_t) \leq \mathcal{L}_{\max}$ for all $t \leq T$.*

*We also define $\Delta\mathcal{L}_{\max} := \mathcal{L}_{\max} - \mathcal{L}_\star$.*

**Lemma D.6** (One-step expected descent scheduled SGD). *Assume Assumption D.5. Then for all $t \geq 0$,*

$$\mathbb{E}[\mathcal{L}(v_{t+1}) \mid \mathcal{F}_t] \leq \mathcal{L}(v_t) - \frac{\eta_t}{2}\|\nabla\mathcal{L}(v_t)\|^2 + \frac{L_{\mathrm{sm}}\eta_t^2}{2}\,\xi^2. \tag{6}$$

*Proof.* By $L_{\mathrm{sm}}$-smoothness, for any $u$ we have $\mathcal{L}(u) \leq \mathcal{L}(v_t) + \langle \nabla\mathcal{L}(v_t), u - v_t \rangle + \frac{L_{\mathrm{sm}}}{2}\|u - v_t\|^2$. Plugging $u = v_{t+1} = v_t - \eta_t g_t$ gives

$$\mathcal{L}(v_{t+1}) \leq \mathcal{L}(v_t) - \eta_t\langle \nabla\mathcal{L}(v_t), g_t \rangle + \frac{L_{\mathrm{sm}}\eta_t^2}{2}\|g_t\|^2.$$

Taking $\mathbb{E}[\cdot \mid \mathcal{F}_t]$, using unbiasedness $\mathbb{E}[g_t \mid \mathcal{F}_t] = \nabla \mathcal{L}(v_t)$, and $\mathbb{E}[\|g_t\|^2 \mid \mathcal{F}_t] \leq \|\nabla \mathcal{L}(v_t)\|^2 + \xi^2$, we obtain

$$\mathbb{E}[\mathcal{L}(v_{t+1}) \mid \mathcal{F}_t] \leq \mathcal{L}(v_t) - \eta_t \|\nabla \mathcal{L}(v_t)\|^2 + \frac{L_{\mathrm{sm}} \eta_t^2}{2} \big( \|\nabla \mathcal{L}(v_t)\|^2 + \xi^2 \big).$$

Finally, since $\eta_t \leq 1/L_{\mathrm{sm}}$ we have $1 - \frac{L_{\mathrm{sm}} \eta_t}{2} \geq \frac{1}{2}$, yielding Lemma D.6. $\qquad \square$

**Lemma D.7** (Conditional weighted squared-gradient bound). *Assume Assumption D.5. Fix any stopping time $\theta \leq T$ and any integer $n \geq 1$ with $\theta + n \leq T$. Then*

$$\mathbb{E}\left[ \sum_{t=\theta}^{\theta+n-1} \eta_t \|\nabla \mathcal{L}(v_t)\|^2 \,\middle|\, \mathcal{F}_\theta \right] \leq 2\, \Delta \mathcal{L}_{\max} + L_{\mathrm{sm}} \xi^2 \sum_{t=\theta}^{\theta+n-1} \eta_t^2.$$

*Proof.* Apply Eq. (6) and take conditional expectations given $\mathcal{F}_\theta$. Summing from $t = \theta$ to $\theta + n - 1$ and telescoping yields

$$\mathbb{E}[\mathcal{L}(v_{\theta+n}) \mid \mathcal{F}_\theta] \leq \mathcal{L}(v_\theta) - \frac{1}{2} \mathbb{E}\left[ \sum_{t=\theta}^{\theta+n-1} \eta_t \|\nabla \mathcal{L}(v_t)\|^2 \,\middle|\, \mathcal{F}_\theta \right] + \frac{L_{\mathrm{sm}} \xi^2}{2} \sum_{t=\theta}^{\theta+n-1} \eta_t^2.$$

Using $\mathcal{L}(v_{\theta+n}) \geq \mathcal{L}_\star$ and $\mathcal{L}(v_\theta) \leq \mathcal{L}_{\max}$ (since $\theta \leq T$) gives Lemma D.7. $\qquad \square$

**Lemma D.8** (Cumulative inward drift control in expectation). *Assume Assumption D.5. Fix any $k \geq 0$ and define, for $t \geq \sigma_k$,*
$$d_t := \mathbb{E}[z_{t+1} - z_t \mid \mathcal{F}_t].$$

*Then for any integer $n \geq 1$ with $\sigma_k + n \leq T$,*

$$\mathbb{E}\left[ - \sum_{t=\sigma_k}^{\sigma_k+n-1} d_t \,\middle|\, \mathcal{F}_{\sigma_k} \right] \leq \bar{B}_n,$$

*where (writing $\Lambda_{k,n} := \sum_{t=\sigma_k}^{\sigma_k+n-1} \eta_t$ and $\Lambda_{k,n}^{(2)} := \sum_{t=\sigma_k}^{\sigma_k+n-1} \eta_t^2$)*

$$\bar{B}_n := \sqrt{\Lambda_{k,n} S_{k,n}}, \quad S_{k,n} := 2\, \Delta \mathcal{L}_{\max} + \xi^2 \Lambda_{k,n} + L_{\mathrm{sm}} \xi^2 \Lambda_{k,n}^{(2)}.$$

*In particular, letting $\Lambda_T := \sum_{t=0}^{T-1} \eta_t$ and $\Lambda_T^{(2)} := \sum_{t=0}^{T-1} \eta_t^2$, we have $\bar{B}_n \leq \bar{B}_T$ for all $n \leq T - \sigma_k$, where*

$$\bar{B}_T := \sqrt{\Lambda_T \Big( 2\, \Delta \mathcal{L}_{\max} + \xi^2 \Lambda_T + L_{\mathrm{sm}} \xi^2 \Lambda_T^{(2)} \Big)}.$$

*Proof.* By the scheduled SGD update, $w_{t+1} - w_t = -\eta_t\, e_i^\top g_t$, hence $z_{t+1} - z_t = s_k(w_{t+1} - w_t) = -\eta_t\, s_k\, e_i^\top g_t$. Therefore

$$-d_t = \eta_t\, \mathbb{E}\big[ s_k\, e_i^\top g_t \mid \mathcal{F}_t \big] \leq \eta_t\, \mathbb{E}[\|g_t\| \mid \mathcal{F}_t] \leq \eta_t\, \sqrt{\mathbb{E}[\|g_t\|^2 \mid \mathcal{F}_t]},$$

where we used $|s_k| = 1$, $|e_i^\top x| \leq \|x\|$, and Jensen.

Summing from $t = \sigma_k$ to $\sigma_k + n - 1$ and applying Cauchy–Schwarz conditionally on $\mathcal{F}_{\sigma_k}$,

$$\mathbb{E}\left[ - \sum_{t=\sigma_k}^{\sigma_k+n-1} d_t \,\middle|\, \mathcal{F}_{\sigma_k} \right] \leq \sqrt{\Lambda_{k,n}} \sqrt{\mathbb{E}\left[ \sum_{t=\sigma_k}^{\sigma_k+n-1} \eta_t\, \mathbb{E}[\|g_t\|^2 \mid \mathcal{F}_t] \,\middle|\, \mathcal{F}_{\sigma_k} \right]}.$$

Moreover, using $\mathbb{E}[\|g_t\|^2 \mid \mathcal{F}_t] \leq \|\nabla \mathcal{L}(v_t)\|^2 + \xi^2$ and Lemma D.7 with $\theta = \sigma_k$ yields

$$\mathbb{E}\left[ \sum_{t=\sigma_k}^{\sigma_k+n-1} \eta_t\, \mathbb{E}[\|g_t\|^2 \mid \mathcal{F}_t] \,\middle|\, \mathcal{F}_{\sigma_k} \right] \leq \Big( 2\, \Delta \mathcal{L}_{\max} + L_{\mathrm{sm}} \xi^2 \Lambda_{k,n}^{(2)} \Big) + \xi^2 \Lambda_{k,n} = S_{k,n}.$$

Combining proves Lemma D.8. The final inequality $\bar{B}_n \leq \bar{B}_T$ follows from $\Lambda_{k,n} \leq \Lambda_T$ and $\Lambda_{k,n}^{(2)} \leq \Lambda_T^{(2)}$. $\qquad \square$

**Lemma D.9** (Conditional bound for the drift-budget proxy $\tilde{U}$). *Assume Assumption D.5. Fix any stopping time $\theta \leq T$ and any integer $n \geq 1$ with $\theta + n \leq T$. Define*

$$\tilde{u}_t := \eta_t \, \mathbb{E}\big[\|g_t\| \mid \mathcal{F}_t\big], \quad \tilde{U}_{\theta,n} := \sum_{t=\theta}^{\theta+n-1} \tilde{u}_t.$$

*Let*

$$\Lambda_{\theta,n} := \sum_{t=\theta}^{\theta+n-1} \eta_t, \quad \Lambda_{\theta,n}^{(2)} := \sum_{t=\theta}^{\theta+n-1} \eta_t^2, \quad \Delta\mathcal{L}_{\max} := \mathcal{L}_{\max} - \mathcal{L}_\star.$$

*Then*

$$\mathbb{E}\big[\tilde{U}_{\theta,n} \mid \mathcal{F}_\theta\big] \leq \bar{B}_{\theta,n} := \sqrt{\Lambda_{\theta,n} \, S_{\theta,n}},$$

*where*

$$S_{\theta,n} := 2\Delta\mathcal{L}_{\max} + \xi^2 \Lambda_{\theta,n} + L_{\mathrm{sm}} \, \xi^2 \, \Lambda_{\theta,n}^{(2)}.$$

*In particular, since $\eta_t \geq 0$ for all $t$ (Assumption D.5(iv)), for any stopping time $\theta \leq T$ and any $n \geq 1$ with $\theta + n \leq T$, the random partial sums satisfy almost surely*

$$\Lambda_{\theta,n} \leq \Lambda_T \quad and \quad \Lambda_{\theta,n}^{(2)} \leq \Lambda_T^{(2)},$$

*and hence $S_{\theta,n} \leq S_T$ and $\bar{B}_{\theta,n} \leq \bar{B}_T$ almost surely, where*

$$S_T := 2\Delta\mathcal{L}_{\max} + \xi^2 \Lambda_T + L_{\mathrm{sm}}\xi^2\Lambda_T^{(2)}, \quad \bar{B}_T := \sqrt{\Lambda_T S_T}.$$

*Consequently,*

$$\mathbb{E}\big[\tilde{U}_{\theta,n} \mid \mathcal{F}_\theta\big] \leq \bar{B}_T.$$

*Proof.* By the tower property,

$$\mathbb{E}\big[\tilde{U}_{\theta,n} \mid \mathcal{F}_\theta\big] = \sum_{t=\theta}^{\theta+n-1} \eta_t \, \mathbb{E}\big[\|g_t\| \mid \mathcal{F}_\theta\big].$$

Using Jensen's inequality $\mathbb{E}[\|g_t\| \mid \mathcal{F}_\theta] \leq \sqrt{\mathbb{E}[\|g_t\|^2 \mid \mathcal{F}_\theta]}$ and conditional Cauchy–Schwarz,

$$\mathbb{E}\big[\tilde{U}_{\theta,n} \mid \mathcal{F}_\theta\big] \leq \sqrt{\Lambda_{\theta,n}}\sqrt{\sum_{t=\theta}^{\theta+n-1} \eta_t \, \mathbb{E}\big[\|g_t\|^2 \mid \mathcal{F}_\theta\big]}.$$

Moreover, $\mathbb{E}[\|g_t\|^2 \mid \mathcal{F}_t] \leq \|\nabla\mathcal{L}(v_t)\|^2 + \xi^2$ by Assumption D.5(iii), hence

$$\mathbb{E}\big[\|g_t\|^2 \mid \mathcal{F}_\theta\big] = \mathbb{E}\big[\mathbb{E}\big[\|g_t\|^2 \mid \mathcal{F}_t\big]\big|\mathcal{F}_\theta\big] \leq \mathbb{E}\big[\|\nabla\mathcal{L}(v_t)\|^2 \mid \mathcal{F}_\theta\big] + \xi^2.$$

Therefore,

$$\sum_{t=\theta}^{\theta+n-1} \eta_t \, \mathbb{E}\big[\|g_t\|^2 \mid \mathcal{F}_\theta\big] \leq \mathbb{E}\left[\sum_{t=\theta}^{\theta+n-1} \eta_t \|\nabla\mathcal{L}(v_t)\|^2 \Big| \mathcal{F}_\theta\right] + \xi^2\Lambda_{\theta,n}.$$

Applying Lemma D.7 gives

$$\mathbb{E}\left[\sum_{t=\theta}^{\theta+n-1} \eta_t \|\nabla\mathcal{L}(v_t)\|^2 \Big| \mathcal{F}_\theta\right] \leq 2\Delta\mathcal{L}_{\max} + L_{\mathrm{sm}}\xi^2\Lambda_{\theta,n}^{(2)}.$$

Combining the last three displays yields

$$\mathbb{E}\big[\tilde{U}_{\theta,n} \mid \mathcal{F}_\theta\big] \leq \sqrt{\Lambda_{\theta,n}\Big(2\Delta\mathcal{L}_{\max} + \xi^2\Lambda_{\theta,n} + L_{\mathrm{sm}}\xi^2\Lambda_{\theta,n}^{(2)}\Big)} = \bar{B}_{\theta,n}.$$

It remains to justify the uniform bound by $\bar{B}_T$. Since $\eta_t \geq 0$ for all $t$, for each outcome $\omega$ we have

$$\Lambda_{\theta,n}(\omega) = \sum_{t=0}^{T-1} \eta_t \, \mathbf{1}\{\theta(\omega) \leq t \leq \theta(\omega) + n - 1\} \leq \sum_{t=0}^{T-1} \eta_t = \Lambda_T,$$

and similarly,

$$\Lambda_{\theta,n}^{(2)}(\omega) = \sum_{t=0}^{T-1} \eta_t^2 \, \mathbf{1}\{\theta(\omega) \leq t \leq \theta(\omega) + n - 1\} \leq \sum_{t=0}^{T-1} \eta_t^2 = \Lambda_T^{(2)}.$$

Therefore $S_{\theta,n}(\omega) \leq S_T$ and hence

$$\bar{B}_{\theta,n}(\omega) = \sqrt{\Lambda_{\theta,n}(\omega)\, S_{\theta,n}(\omega)} \leq \sqrt{\Lambda_T S_T} = \bar{B}_T \quad \text{for all } \omega,$$

which proves $\mathbb{E}[\tilde{U}_{\theta,n} \mid \mathcal{F}_\theta] \leq \bar{B}_T$. $\qquad \qquad \square$

**Proposition D.10** (Re-entry bound in SGD). *Assume Assumptions 3.3 and D.5. Recall $\Delta\mathcal{L}_{\max} := \mathcal{L}_{\max} - \mathcal{L}_\star$ and define the cumulative schedule quantities*

$$\Lambda_T := \sum_{t=0}^{T-1} \eta_t, \ \ \Lambda_T^{(2)} := \sum_{t=0}^{T-1} \eta_t^2.$$

*Fix $\delta \in (0,1)$ and define*

$$\bar{B}_T := \sqrt{\Lambda_T \left(2\,\Delta\mathcal{L}_{\max} + \xi^2 \Lambda_T + L_{\mathrm{sm}}\xi^2 \Lambda_T^{(2)}\right)},$$

$$B_T^{\mathrm{SGD}} := \frac{1}{\delta}\, \bar{B}_T = \frac{1}{\delta}\sqrt{\Lambda_T \left(2\,\Delta\mathcal{L}_{\max} + \xi^2 \Lambda_T + L_{\mathrm{sm}}\xi^2 \Lambda_T^{(2)}\right)},$$

$$V_T^{\mathrm{SGD}} := \xi^2 \sum_{t=0}^{T-1} \eta_t^2 = \xi^2 \Lambda_T^{(2)}, \tag{7}$$

$$a_T := \left((\rho - \epsilon) - B_T^{\mathrm{SGD}}\right)_+,$$

$$g_T^{\mathrm{SGD}} := \delta_{\mathrm{upd}} + \delta + \exp\left(-\frac{a_T^2}{2\left(V_T^{\mathrm{SGD}} + \frac{2\Delta}{3} a_T\right)}\right).$$

*Then for every $k \geq 0$,*

$$\mathbb{P}[\tau_{k+1} \leq T \mid \mathcal{F}_{\sigma_k}] \leq g_T^{\mathrm{SGD}} \quad \text{on } \{\sigma_k \leq T\}. \tag{8}$$

*In particular, if $g_T^{\mathrm{SGD}} < 1$ (e.g., $\delta_{\mathrm{upd}}$ and $\delta$ are small and $(\rho - \epsilon) > B_T^{\mathrm{SGD}}$), then Assumption 3.4 holds.*

*Proof.* Fix $k$ and condition on $\mathcal{F}_{\sigma_k}$. If $\sigma_k = T$, then $\tau_{k+1} > T$ by definition and the trivial result

$$\mathbb{P}[\tau_{k+1} \leq T \mid \mathcal{F}_{\sigma_k}] = 0 \leq g_T^{\mathrm{SGD}},$$

holds. Hence assume $\sigma_k \leq T - 1$.

**Shifted filtration and shifted stopping times.** Define the shifted filtration

$$\mathcal{H}_m := \mathcal{F}_{\sigma_k + m} \ \ (m \geq 0).$$

We use $\mathcal{H}_m$ to avoid confusion with the global notation $G_t := \mathcal{F}_{\sigma_k + t}$. Define shifted stopping times (w.r.t. $(\mathcal{H}_m)_{m \geq 0}$)

$$\tau' := \tau_{k+1} - \sigma_k, \ \ \nu' := \nu - \sigma_k,$$

where $\nu$ is defined below. Note that $\tau' \in \{1, 2, \dots\} \cup \{\infty\}$ and $\nu' \in \{0, 1, 2, \dots\} \cup \{\infty\}$.

Define the (random but $\mathcal{F}_{\sigma_k}$-measurable) horizon

$$T^{(k)} := T - \sigma_k.$$

Under the conditioning on $\mathcal{F}_{\sigma_k}$, $T^{(k)}$ is deterministic.

**First update-violation time.** Define the *first update-violation time*

$$\nu := \inf\{t \geq \sigma_k : |w_{t+1} - w_t| > \Delta\},$$

with the convention $\inf \emptyset = \infty$. Note that $\{\nu \leq T - 1\} = \mathcal{E}_\Delta(\sigma_k)^c$. By Assumption 3.3 applied at the stopping time $\theta = \sigma_k$,

$$\mathbb{P}[\nu \leq T - 1 \mid \mathcal{F}_{\sigma_k}] \leq \delta_{\text{upd}}.$$

**Martingale differences and conditional variance bound.** Define the predictable drift increments $d_t$ as in Lemma D.8 and the martingale differences

$$X_t := (z_{t+1} - z_t) - d_t.$$

Using $z_{t+1} - z_t = -\eta_t s_k e_i^\top g_t$ and $d_t = \mathbb{E}[z_{t+1} - z_t \mid \mathcal{F}_t] = -\eta_t s_k e_i^\top \nabla\mathcal{L}(v_t)$, we have

$$X_t = -\eta_t s_k e_i^\top \big(g_t - \nabla\mathcal{L}(v_t)\big),$$

and therefore

$$\mathrm{Var}(X_t \mid \mathcal{F}_t) \leq \mathbb{E}[X_t^2 \mid \mathcal{F}_t] \leq \eta_t^2 \,\mathbb{E}[\|g_t - \nabla\mathcal{L}(v_t)\|^2 \mid \mathcal{F}_t] \leq \eta_t^2 \xi^2. \tag{9}$$

**Drift-good event.** Let $n := \min\{T - \sigma_k, \tau_{k+1} - \sigma_k\}$ and define $\tilde{u}_t := \eta_t \mathbb{E}[\|g_t\| \mid \mathcal{F}_t]$ and $\tilde{U}_m := \sum_{t=\sigma_k}^{\sigma_k+m-1} \tilde{u}_t$ (with $\tilde{U}_0 := 0$). Consider the drift-good event

$$G_{\text{drift}} := \{\tilde{U}_n \leq B_T^{\text{SGD}}\}.$$

Since $\tilde{U}_n \leq \tilde{U}_{T-\sigma_k}$ and $\tilde{U}_{T-\sigma_k} \geq 0$, Markov's inequality yields

$$\mathbb{P}(G_{\text{drift}}^c \mid \mathcal{F}_{\sigma_k}) = \mathbb{P}(\tilde{U}_n > B_T^{\text{SGD}} \mid \mathcal{F}_{\sigma_k}) \leq \frac{\mathbb{E}[\tilde{U}_{T-\sigma_k} \mid \mathcal{F}_{\sigma_k}]}{B_T^{\text{SGD}}}.$$

By Lemma D.9 (applied with $\theta = \sigma_k$ and horizon $T - \sigma_k$),

$$\mathbb{E}[\tilde{U}_{T-\sigma_k} \mid \mathcal{F}_{\sigma_k}] \leq \bar{B}_T.$$

Therefore, using $B_T^{\text{SGD}} = \bar{B}_T/\delta$, we conclude

$$\mathbb{P}(G_{\text{drift}}^c \mid \mathcal{F}_{\sigma_k}) \leq \delta.$$

On $G_{\text{drift}}$, for any $m \leq n$ we have $\sum_{t=\sigma_k}^{\sigma_k+m-1} d_t \geq -\tilde{U}_m \geq -\tilde{U}_n \geq -B_T^{\text{SGD}}$. Thus,

$$z_{\sigma_k+m} = z_{\sigma_k} + \sum_{t=\sigma_k}^{\sigma_k+m-1} d_t + \sum_{t=\sigma_k}^{\sigma_k+m-1} X_t \geq \rho - B_T^{\text{SGD}} + \sum_{t=\sigma_k}^{\sigma_k+m-1} X_t,$$

where we used $z_{\sigma_k} \geq \rho$. Consequently, on $\{\min_{0 \leq m \leq n} z_{\sigma_k+m} \leq \epsilon\} \cap G_{\text{drift}}$ we must have

$$\min_{0 \leq m \leq n} \sum_{t=\sigma_k}^{\sigma_k+m-1} X_t \leq -(\rho - \epsilon) + B_T^{\text{SGD}}.$$

**Stopped martingale and Freedman under shifted filtration.** Let

$$x := \big((\rho - \epsilon) - B_T^{\text{SGD}}\big)_+,$$

so $x = a_T$ in Proposition D.10. Define shifted increments $Y_m := X_{\sigma_k+m}$ for $m \geq 0$. Then $(Y_m)_{m \geq 0}$ is a martingale difference sequence w.r.t. $(\mathcal{H}_m)_{m \geq 0}$.

Define the *stopped* increments and stopped martingale

$$\bar{Y}_m := Y_m \mathbf{1}\{m < \nu'\}, \quad \bar{M}_m := \sum_{j=0}^{m-1} \bar{Y}_j \ (m \geq 0), \quad \bar{M}_0 := 0.$$

Then $(\bar{M}_m)_{m \geq 0}$ is a martingale w.r.t. $(\mathcal{H}_m)$.

*Increment bound (holds a.s. without conditioning on $\{\nu > T - 1\}$):* Since $\nu$ is a stopping time, $\{m < \nu'\} \in \mathcal{H}_m$. On $\{m < \nu'\}$, we have $|w_{\sigma_k+m+1} - w_{\sigma_k+m}| \leq \Delta$, hence $|z_{\sigma_k+m+1} - z_{\sigma_k+m}| \leq \Delta$ and thus

$$|d_{\sigma_k+m}| = \left|\mathbb{E}[z_{\sigma_k+m+1} - z_{\sigma_k+m} \mid \mathcal{H}_m]\right| \leq \mathbb{E}[|z_{\sigma_k+m+1} - z_{\sigma_k+m}| \mid \mathcal{H}_m] \leq \Delta.$$

Therefore on $\{m < \nu'\}$, $|Y_m| = |X_{\sigma_k+m}| \leq |z_{\sigma_k+m+1} - z_{\sigma_k+m}| + |d_{\sigma_k+m}| \leq 2\Delta$. Since $\bar{Y}_m = 0$ on $\{m \geq \nu'\}$, we obtain the uniform a.s. bound

$$|\bar{Y}_m| \leq 2\Delta \text{ for all } m \geq 0 \text{ a.s.}$$

*Conditional variance bound:* Using Eq. (9) and $\mathrm{Var}(\bar{Y}_m \mid \mathcal{H}_m) \leq \mathrm{Var}(Y_m \mid \mathcal{H}_m)$, we have

$$\mathrm{Var}(\bar{Y}_m \mid \mathcal{H}_m) \leq \xi^2 \eta_{\sigma_k+m}^2.$$

Define the predictable quadratic variation up to $m$:

$$\bar{V}_m := \sum_{j=0}^{m-1} \mathrm{Var}(\bar{Y}_j \mid \mathcal{H}_j).$$

Then for all $m \leq T^{(k)}$,

$$\bar{V}_m \leq \xi^2 \sum_{t=\sigma_k}^{\sigma_k+m-1} \eta_t^2 \leq \xi^2 \sum_{t=0}^{T-1} \eta_t^2 = V_T^{\mathrm{SGD}}.$$

From the drift-good reduction above, on $\mathsf{G}_{\mathrm{drift}}$ and $\{\tau_{k+1} \leq T\}$ we have

$$\min_{0 \leq m \leq n} \sum_{t=\sigma_k}^{\sigma_k+m-1} X_t \leq -x.$$

If additionally $\{\nu > T - 1\}$ holds, then $\nu' > T^{(k)}$ so $m < \nu'$ for all $m \leq T^{(k)}$, hence $\bar{M}_m = \sum_{j=0}^{m-1} Y_j = \sum_{t=\sigma_k}^{\sigma_k+m-1} X_t$ for all $m \leq T^{(k)}$. Therefore,

$$\{\tau_{k+1} \leq T\} \cap \{\nu > T - 1\} \cap \mathsf{G}_{\mathrm{drift}} \subseteq \left\{\min_{0 \leq m \leq T^{(k)}} \bar{M}_m \leq -x\right\}.$$

*Freedman's inequality (maximal form) applied to $(\bar{M}_m)$:* Equivalently, apply Freedman's inequality to the martingale $(-\bar{M}_m)$ to bound the lower-tail event. With increment bound $2\Delta$ and variance bound $V_T^{\mathrm{SGD}}$, we obtain

$$\mathbb{P}\left[\min_{0 \leq m \leq T^{(k)}} \bar{M}_m \leq -x \,\middle|\, \mathcal{F}_{\sigma_k}\right] \leq \exp\left(-\frac{x^2}{2\left(V_T^{\mathrm{SGD}} + \frac{2\Delta}{3}x\right)}\right).$$

Consequently,

$$\mathbb{P}[\tau_{k+1} \leq T, \, \nu > T - 1, \, \mathsf{G}_{\mathrm{drift}} \mid \mathcal{F}_{\sigma_k}] \leq \exp\left(-\frac{x^2}{2\left(V_T^{\mathrm{SGD}} + \frac{2\Delta}{3}x\right)}\right).$$

**Union bound completion.** Finally,

$$\mathbb{P}[\tau_{k+1} \leq T \mid \mathcal{F}_{\sigma_k}] \leq \mathbb{P}[\nu \leq T - 1 \mid \mathcal{F}_{\sigma_k}] + \mathbb{P}[\mathsf{G}_{\mathrm{drift}}^c \mid \mathcal{F}_{\sigma_k}] + \mathbb{P}[\tau_{k+1} \leq T, \, \nu > T - 1, \, \mathsf{G}_{\mathrm{drift}} \mid \mathcal{F}_{\sigma_k}]$$

$$\leq \delta_{\mathrm{upd}} + \delta + \exp\left(-\frac{x^2}{2\left(V_T^{\mathrm{SGD}} + \frac{2\Delta}{3}x\right)}\right),$$

which is exactly Proposition D.10. $\qquad\square$

**Remark D.11** (How the proofs interface with momentum / AdamW). *Theorem 3.6 is invoked through only two ingredients: (i) the bounded-update condition (Assumption 3.3), and (ii) the Re-entry control (Assumption 3.4). In contrast, Proposition D.10 is merely a* scheduled-SGD sufficient condition *for verifying (ii). Therefore, any optimizer for which one can verify (i)–(ii) yields the same geometric-tail conclusion. As a generic template, consider a preconditioned momentum recursion*

$$p_{t+1} = \beta_t^{\mathrm{mom}} p_t - \eta_t D_t g_t, \quad v_{t+1} = v_t + p_{t+1}, \tag{10}$$

*with $p_0 = 0$ and a diagonal $\mathcal{F}_t$-measurable preconditioner $D_t$. Since $v_{t+1} - v_t = p_{t+1}$, Assumption 3.3 holds on any good event on which*

$$\|p_{t+1}\|_\infty \leq \Delta \ \text{ for all } t \leq T - 1, \tag{11}$$

*(with failure probability given by the complement of the good event). A convenient sufficient condition for Eq. (11) is, for example, the conjunction of $\|\eta_t D_t g_t\|_\infty \leq (1 - \beta)\Delta$ for all $t \leq T - 1$ and $\sup_{t \leq T-1} |\beta_t^{\mathrm{mom}}| \leq \beta < 1$ (so that Eq. (11) follows by a simple induction starting from $p_0 = 0$). For Adam/AdamW, one may instantiate Eq. (10) with $D_t = \mathrm{diag}((\sqrt{\hat{s}_t} + \epsilon_{\mathrm{adam}})^{-1})$ and $g_t = \hat{m}_t$. Thus it suffices (along the trajectory / on a good event) to control $\|\hat{m}_t\|_\infty$, to ensure a uniform lower bound $\sqrt{\hat{s}_t} + \epsilon_{\mathrm{adam}} \geq c > 0$, and to verify Eq. (11) for the resulting effective step $p_{t+1}$. For AdamW, the decoupled weight-decay term can be absorbed into the same bounded-update event (e.g., by including it in the definition of $p_{t+1}$ and bounding its $l_\infty$-magnitude).*

### D.4. Practical Implication of Sign Lock-In Theory

In this section, we translate our theoretical results into design principles for modern training pipelines. The key observation is that, by Theorem 3.6, the statistics of effective sign flips are characterized by the initial-hit factor $h_T$ and the re-entry ratio $g_T$. For the initial-hit factor $h_T$, Appendix D.2 and Proposition D.4 provide an initialization-dominated upper bound, showing that lock-in becomes stronger as the probability of reaching the boundary band decreases. For the re-entry ratio $g_T$, satisfying Assumption 3.4 is essential, and Proposition D.10 justifies this assumption by giving a sufficient condition that upper bounds $g_T$ in terms of SGD and schedule-dependent cumulative quantities. Below, through these two controlling factors, we systematically summarize how learning-rate schedules, batch size, width, and scale-invariant mechanisms shift lock-in in either direction.

- Appendix D.4.1 compares learning-rate schedules by instantiating the $g_T$ upper bound from Proposition D.10 with the corresponding schedule-dependent cumulative quantities, and summarizes which schedules tend to yield stronger lock-in.

- Appendix D.4.2 explains how increasing the minibatch size and model size suppresses stochastic gradient noise, thereby reducing re-entry behavior and decreasing $g_T$, which strengthens lock-in.

- Appendix D.4.3 summarizes how scale-invariant mechanisms stabilize effective updates and thereby support the conditions in Assumption 3.3 and Assumption 3.4, resulting in stronger lock-in.

- Appendix D.4.4 presents auxiliary consequences of sign dynamics, including an initial-hit bound that does not require an outer start assumption and the resulting front-loaded nature of sign changes.

- Appendix D.4.5 provides numerical validation of these practical insights through the learning-rate schedule, batch-size, and width sweeps, together with fitted trends $(\hat{h}, \hat{g})$.

D.4.1. ORDERING OF LEARNING-RATE SCHEDULES UNDER FIXED COMPUTING RESOURCE

We compare four learning-rate schedules under the same training horizon $T$ and the same peak step size $\eta_{\mathrm{max}}$: a constant learning rate, warmup with cosine decay, warmup with exponential decay, and warmup with inverse decay. Our goal is to order the resulting re-entry bounds using the schedule-aware sufficient condition of Proposition D.10 (Appendix D.3), which verifies Assumption 3.4 and hence yields the geometric-tail conclusion of Theorem 3.6.

**Reminder (where $g_T$ enters).** Assumption 3.4 postulates the existence of a re-entry bound $g_T$ such that

$$\mathbb{P}[\tau_{k+1} \leq T \mid \mathcal{F}_{\sigma_k}] \leq g_T.$$

Theorem 3.6 then converts this bound into a geometric-tail estimate for both $\mathbb{P}(\tau_k \leq T)$ and $\mathbb{P}(K_T^{\mathrm{eff}}(\rho) \geq k)$. Appendix D.3 provides a schedule-aware sufficient condition: Proposition D.10 upper-bounds the re-entry probability by an explicit quantity $g_T^{\mathrm{SGD}}$, which depends on the learning-rate schedule only through the cumulative quantities

$$\Lambda_T = \sum_{t<T} \eta_t, \quad \Lambda_T^{(2)} = \sum_{t<T} \eta_t^2.$$

**Schedule-specific notation.** To distinguish schedules, we write

$$g_{T,\mathrm{const}}, \quad g_{T,\mathrm{cos}}, \quad g_{T,\mathrm{exp}}, \quad g_{T,\mathrm{inv}}$$

for the re-entry bounds associated with a constant learning rate, warmup with cosine decay, warmup with exponential decay, and warmup with inverse decay, respectively. When referring to the sufficient bound in Proposition D.10, we analogously write $g_{T,\mathrm{const}}^{\mathrm{SGD}}$, $g_{T,\mathrm{cos}}^{\mathrm{SGD}}$, $g_{T,\mathrm{exp}}^{\mathrm{SGD}}$, and $g_{T,\mathrm{inv}}^{\mathrm{SGD}}$.

**Lemma D.12** (Monotonicity of the SGD re-entry bound). *Fix the constants $\rho, \epsilon, \Delta, \delta, \delta_{\mathrm{upd}}, L_{\mathrm{sm}}, \xi, \Delta\mathcal{L}_{\mathrm{max}}$ as in Proposition D.10. For a schedule $\{\eta_t\}_{t=0}^{T-1}$, define*

$$\Lambda_T^{(1)} := \sum_{t=0}^{T-1} \eta_t, \quad \Lambda_T^{(2)} := \sum_{t=0}^{T-1} \eta_t^2.$$

*If two schedules satisfy*

$$\Lambda_T^{(1)} \leq \widetilde{\Lambda}_T^{(1)}, \quad \Lambda_T^{(2)} \leq \widetilde{\Lambda}_T^{(2)},$$

*then their schedule-specific bounds satisfy*

$$g_T^{\mathrm{SGD}} \leq \widetilde{g}_T^{\mathrm{SGD}}.$$

*Moreover, if at least one inequality is strict and the effective margin $a_T = \big((\rho - \epsilon) - \widetilde{B}_T^{\mathrm{SGD}}\big)_+$ for the larger schedule is strictly positive, then the inequality is strict.*

*Proof.* By Proposition D.10,

$$g_T^{\mathrm{SGD}} = \delta_{\mathrm{upd}} + \delta + \exp\left(-\frac{a_T^2}{2\left(V_T^{\mathrm{SGD}} + \frac{2\Delta}{3} a_T\right)}\right),$$

where

$$V_T^{\mathrm{SGD}} = \xi^2 \Lambda_T^{(2)}, \quad B_T^{\mathrm{SGD}} = \frac{1}{\delta}\sqrt{\Lambda_T^{(1)}\left(2\Delta\mathcal{L}_{\mathrm{max}} + \xi^2 \Lambda_T^{(1)} + L_{\mathrm{sm}}\xi^2 \Lambda_T^{(2)}\right)}.$$

Both $V_T^{\mathrm{SGD}}$ and $B_T^{\mathrm{SGD}}$ are nondecreasing in $(\Lambda_T^{(1)}, \Lambda_T^{(2)})$, hence $a_T = \big((\rho - \epsilon) - B_T^{\mathrm{SGD}}\big)_+$ is nonincreasing in those quantities. Therefore increasing either $\Lambda_T^{(1)}$ or $\Lambda_T^{(2)}$ can only weaken the exponent (make it less negative), so $g_T^{\mathrm{SGD}}$ is nondecreasing in $(\Lambda_T^{(1)}, \Lambda_T^{(2)})$. This proves the first claim. The strictness statement follows when at least one inequality is strict and the larger schedule has $a_T > 0$, so the exponential term changes strictly. $\square$

**Proposition D.13** (Learning-rate dependence of the re-entry ratio). *Assume the hypotheses of Proposition D.10 and fix the constants $\rho, \epsilon, \Delta, \delta, \delta_{\mathrm{upd}}, L_{\mathrm{sm}}, \xi, \Delta\mathcal{L}_{\mathrm{max}}$ as in Lemma D.12. Let $\{\eta_t\}_{t=0}^{T-1}$ and $\{\widetilde{\eta}_t\}_{t=0}^{T-1}$ be two learning-rate schedules, and define*

$$\Lambda_T^{(1)} := \sum_{t=0}^{T-1} \eta_t, \quad \Lambda_T^{(2)} := \sum_{t=0}^{T-1} \eta_t^2, \quad \widetilde{\Lambda}_T^{(1)} := \sum_{t=0}^{T-1} \widetilde{\eta}_t, \quad \widetilde{\Lambda}_T^{(2)} := \sum_{t=0}^{T-1} \widetilde{\eta}_t^2.$$

*If*

$$\Lambda_T^{(1)} \leq \widetilde{\Lambda}_T^{(1)}, \quad \Lambda_T^{(2)} \leq \widetilde{\Lambda}_T^{(2)},$$

*then the corresponding schedule-aware re-entry bounds satisfy*

$$g_T^{\mathrm{SGD}} \leq \widetilde{g}_T^{\mathrm{SGD}}.$$

*Moreover, if at least one of the two inequalities above is strict and the effective margin associated with the larger schedule is nontrivial,*

$$\widetilde{a}_T := \big((\rho - \epsilon) - \widetilde{B}_T^{\mathrm{SGD}}\big)_+ > 0,$$

*then the inequality is strict, i.e.,*

$$g_T^{\mathrm{SGD}} < \widetilde{g}_T^{\mathrm{SGD}}.$$

*Furthermore, suppose that the learning-rate schedules are related by a multiplicative rescaling $\widetilde{\eta}_t = c\,\eta_t$ with a constant $c > 1$. Then the cumulative quantities satisfy $\widetilde{\Lambda}_T^{(1)} = c\,\Lambda_T^{(1)}$ and $\widetilde{\Lambda}_T^{(2)} = c^2\,\Lambda_T^{(2)}$, and consequently the corresponding re-entry bounds obey*

$$\widetilde{g}_T^{\mathrm{SGD}} \ge g_T^{\mathrm{SGD}}.$$

*If, in addition, the effective margin for the rescaled schedule satisfies $\widetilde{a}_T > 0$, the inequality is strict.*

*Proof.* The claim follows directly from Lemma D.12. By Proposition D.10, the explicit bound $g_T^{\mathrm{SGD}}$ depends on the learning-rate schedule only through the cumulative quantities $\Lambda_T^{(1)}$ and $\Lambda_T^{(2)}$, and is nondecreasing in each of them. The strictness statement follows from the strict monotonicity part of Lemma D.12 whenever the effective margin is positive. $\square$

**Lemma D.14** (Constant learning rate is maximal). *Fix $T$ and $\eta_{\max} > 0$. Among all schedules satisfying $0 \le \eta_t \le \eta_{\max}$ for all $t$, the constant learning rate $\eta_t \equiv \eta_{\max}$ maximizes both $\Lambda_T^{(1)}$ and $\Lambda_T^{(2)}$.*

*Proof.* For any schedule with $0 \le \eta_t \le \eta_{\max}$,

$$\Lambda_T^{(1)} = \sum_{t=0}^{T-1} \eta_t \le \sum_{t=0}^{T-1} \eta_{\max} = T\eta_{\max}, \quad \Lambda_T^{(2)} = \sum_{t=0}^{T-1} \eta_t^2 \le \sum_{t=0}^{T-1} \eta_{\max}^2 = T\eta_{\max}^2,$$

with equality for the constant schedule. $\square$

**Lemma D.15** (Cosine decay yields linear squared-step accumulation (exact form)). *Let $T_{\mathrm{wu}}$ denote the warmup length and $N := T - T_{\mathrm{wu}}$. Assume $N \ge 2$. For warmup with cosine decay,*

$$\eta_{T_{\mathrm{wu}}+k} = \eta_{\max} \frac{1 + \cos(\pi k/N)}{2}, \quad k = 0, \ldots, N-1,$$

*we have the exact identity*

$$\sum_{k=0}^{N-1} \eta_{T_{\mathrm{wu}}+k}^2 = \frac{3}{8}\eta_{\max}^2 N + \frac{1}{2}\eta_{\max}^2.$$

*Proof.* Let $\vartheta_k := \pi k/N$ and write

$$\eta_{T_{\mathrm{wu}}+k}^2 = \eta_{\max}^2 \left(\frac{1 + \cos\vartheta_k}{2}\right)^2.$$

Using $\cos^2 x = \frac{1+\cos(2x)}{2}$,

$$\left(\frac{1 + \cos x}{2}\right)^2 = \frac{3}{8} + \frac{1}{2}\cos x + \frac{1}{8}\cos(2x).$$

Hence

$$\sum_{k=0}^{N-1} \eta_{T_{\mathrm{wu}}+k}^2 = \eta_{\max}^2 \left(\frac{3}{8}N + \frac{1}{2}\sum_{k=0}^{N-1} \cos\vartheta_k + \frac{1}{8}\sum_{k=0}^{N-1} \cos(2\vartheta_k)\right).$$

We evaluate the trigonometric sums exactly. First, (using $N \ge 2$ so that $e^{i2\pi/N} \ne 1$),

$$\sum_{k=0}^{N-1} \cos\left(\frac{2\pi k}{N}\right) = \Re \sum_{k=0}^{N-1} e^{i2\pi k/N} = \Re\left(\frac{1 - (e^{i2\pi/N})^N}{1 - e^{i2\pi/N}}\right) = 0.$$

Second, using the standard closed form $\sum_{k=0}^{N-1} \cos(k\theta) = \frac{\sin(N\theta/2)\cos((N-1)\theta/2)}{\sin(\theta/2)}$ with $\theta = \pi/N$, we obtain

$$\sum_{k=0}^{N-1} \cos\left(\frac{\pi k}{N}\right) = \frac{\sin(\pi/2)\cos((N-1)\pi/(2N))}{\sin(\pi/(2N))} = \frac{\cos(\pi/2 - \pi/(2N))}{\sin(\pi/(2N))} = 1.$$

Substituting gives

$$\sum_{k=0}^{N-1} \eta_{T_{\mathrm{wu}}+k}^2 = \eta_{\max}^2 \left(\frac{3}{8}N + \frac{1}{2} \cdot 1 + \frac{1}{8} \cdot 0\right) = \frac{3}{8}\eta_{\max}^2 N + \frac{1}{2}\eta_{\max}^2.$$

$\square$

**Lemma D.16** (Exponential versus cosine decay). *With the same $T_{\mathrm{wu}}$ and $N := T - T_{\mathrm{wu}}$, consider warmup with exponential decay $\eta_{T_{\mathrm{wu}}+k} = \eta_{\max}\gamma^k$ ($0 < \gamma < 1$). If both*

$$\frac{N+1}{2} \leq \frac{1-\gamma^N}{1-\gamma} \quad and \quad \frac{3}{8}N + \frac{1}{2} \leq \frac{1-\gamma^{2N}}{1-\gamma^2},$$

*then*

$$g_{T,\cos}^{\mathrm{SGD}} \leq g_{T,\exp}^{\mathrm{SGD}}.$$

*Proof.* Since the warmup phase is common, it suffices to compare the post-warmup tails. Define the tail cumulative sums

$$\Lambda_{\cos}^{(1)} := \sum_{k=0}^{N-1} \eta_{\max}\frac{1+\cos(\pi k/N)}{2}, \quad \Lambda_{\cos}^{(2)} := \sum_{k=0}^{N-1} \left(\eta_{\max}\frac{1+\cos(\pi k/N)}{2}\right)^2,$$

and similarly for the exponential schedule,

$$\Lambda_{\exp}^{(1)} := \sum_{k=0}^{N-1} \eta_{\max}\gamma^k, \quad \Lambda_{\exp}^{(2)} := \sum_{k=0}^{N-1} \eta_{\max}^2\gamma^{2k}.$$

We compute these exactly. For cosine, using $\sum_{k=0}^{N-1}\cos(\pi k/N) = 1$ (as in Lemma D.15),

$$\Lambda_{\cos}^{(1)} = \eta_{\max} \cdot \frac{1}{2}\left(N + \sum_{k=0}^{N-1}\cos(\pi k/N)\right) = \eta_{\max} \cdot \frac{N+1}{2}.$$

For cosine squared sum, Lemma D.15 gives

$$\Lambda_{\cos}^{(2)} = \frac{3}{8}\eta_{\max}^2 N + \frac{1}{2}\eta_{\max}^2.$$

For exponential,

$$\Lambda_{\exp}^{(1)} = \eta_{\max}\sum_{k=0}^{N-1}\gamma^k = \eta_{\max}\frac{1-\gamma^N}{1-\gamma}, \quad \Lambda_{\exp}^{(2)} = \eta_{\max}^2\sum_{k=0}^{N-1}\gamma^{2k} = \eta_{\max}^2\frac{1-\gamma^{2N}}{1-\gamma^2}.$$

Thus the two stated conditions are exactly $\Lambda_{\cos}^{(1)} \leq \Lambda_{\exp}^{(1)}$ and $\Lambda_{\cos}^{(2)} \leq \Lambda_{\exp}^{(2)}$. Adding the identical warmup contributions preserves these inequalities for $\Lambda_T^{(1)}$ and $\Lambda_T^{(2)}$ over the full horizon. Therefore Lemma D.12 implies $g_{T,\cos}^{\mathrm{SGD}} \leq g_{T,\exp}^{\mathrm{SGD}}$. $\square$

**Lemma D.17** (Inverse decay is asymptotically minimal). *With the same $T_{\mathrm{wu}}$ and $N$, consider warmup with inverse decay $\eta_{T_{\mathrm{wu}}+k} = \eta_{\max}(1+k)^{-p}$ with $p \geq \frac{1}{2}$. Then*

$$\sum_{k=0}^{N-1} \eta_{T_{\mathrm{wu}}+k}^2 = \Theta(\eta_{\max}^2 \log N) \quad for\ p = \tfrac{1}{2}, \quad O(\eta_{\max}^2) \quad for\ p > \tfrac{1}{2}.$$

*In particular, for sufficiently large $N$, the strict inequality below holds whenever the corresponding bound is nontrivial, e.g., whenever the effective margin*

$$a_{T,\cos} := \left((\rho - \epsilon) - B_{T,\cos}^{\mathrm{SGD}}\right)_+$$

*is strictly positive under the conditions of Proposition D.10, so that the strictness clause in Lemma D.12 applies,*

$$g_{T,\mathrm{inv}}^{\mathrm{SGD}} < g_{T,\cos}^{\mathrm{SGD}}.$$

*Proof.* We first prove the squared-step accumulation. Write

$$\sum_{k=0}^{N-1} \eta_{T_{\mathrm{wu}}+k}^2 = \eta_{\max}^2 \sum_{k=0}^{N-1} (1+k)^{-2p} = \eta_{\max}^2 \sum_{j=1}^{N} j^{-2p}.$$

If $p = \frac{1}{2}$, then $2p = 1$ and the harmonic sum yields $\sum_{j=1}^{N} j^{-1} = \Theta(\log N)$. If $p > \frac{1}{2}$, then $2p > 1$ so $\sum_{j=1}^{\infty} j^{-2p} < \infty$ and hence $\sum_{j=1}^{N} j^{-2p} = O(1)$. This proves the first display.

Next, we compare the inverse schedule to cosine to obtain a strict bound on $g_T^{\mathrm{SGD}}$ for large $N$. As in Lemma D.16, warmup is common, so it suffices to compare tail sums. For cosine, Lemma D.15 gives

$$\Lambda_{\cos}^{(2)} = \frac{3}{8} \eta_{\max}^2 N + \frac{1}{2} \eta_{\max}^2 = \Theta(\eta_{\max}^2 N).$$

For inverse, the first part shows

$$\Lambda_{\mathrm{inv}}^{(2)} = \eta_{\max}^2 \sum_{j=1}^{N} j^{-2p} = \Theta(\eta_{\max}^2 \log N) \ (p = \tfrac{1}{2}), \quad O(\eta_{\max}^2) \ (p > \tfrac{1}{2}),$$

so $\Lambda_{\mathrm{inv}}^{(2)} < \Lambda_{\cos}^{(2)}$ for all sufficiently large $N$.

We also compare the first-moment tail sums:

$$\Lambda_{\cos}^{(1)} = \eta_{\max} \frac{N+1}{2} = \Theta(\eta_{\max} N),$$

whereas for inverse decay,

$$\Lambda_{\mathrm{inv}}^{(1)} = \eta_{\max} \sum_{j=1}^{N} j^{-p} = \begin{cases} \Theta(\eta_{\max}\sqrt{N}), & p = \frac{1}{2}, \\ O(\eta_{\max} N^{1-p}), & \frac{1}{2} < p < 1, \\ O(\eta_{\max} \log N), & p = 1, \\ O(\eta_{\max}), & p > 1, \end{cases}$$

which is $o(\eta_{\max} N)$ for every $p \geq \frac{1}{2}$. Hence $\Lambda_{\mathrm{inv}}^{(1)} < \Lambda_{\cos}^{(1)}$ for sufficiently large $N$.

Adding the identical warmup parts preserves strict inequalities for the full-horizon $\Lambda_T^{(1)}$ and $\Lambda_T^{(2)}$. Applying Lemma D.12 (with strictness, once $a_T > 0$) yields $g_{T,\mathrm{inv}}^{\mathrm{SGD}} < g_{T,\cos}^{\mathrm{SGD}}$ for sufficiently large $N$. □

**Proposition D.18** (Ordering of schedule-aware re-entry guarantees). *Assume the hypotheses of Proposition D.10. Fix $T$ and $\eta_{\max}$ and consider the four schedules (constant, warmup+cosine, warmup+exponential, warmup+inverse) with a common warmup length $T_{\mathrm{wu}}$. Then the schedule-specific re-entry bounds satisfy*

$$g_{T,\mathrm{inv}}^{\mathrm{SGD}} < g_{T,\cos}^{\mathrm{SGD}} \ \lesssim \ g_{T,\exp}^{\mathrm{SGD}} < g_{T,\mathrm{const}}^{\mathrm{SGD}},$$

*where the middle relation holds under the conditions of Lemma D.16. Moreover, each strict inequality "$<$" above is to be understood under the strictness regime of Lemma D.12: it holds whenever the effective margin*

$$a_{T,\mathrm{sch}} := \left((\rho - \epsilon) - B_{T,\mathrm{sch}}^{\mathrm{SGD}}\right)_+$$

*for the larger schedule in the corresponding comparison (as defined via Proposition D.10) is strictly positive. Consequently, Assumption 3.4 holds with $g_T = g_{T,\mathrm{sch}}^{\mathrm{SGD}}$ for each schedule $\mathrm{sch} \in \{\mathrm{const}, \cos, \exp, \mathrm{inv}\}$, and the geometric-tail conclusions of Theorem 3.6 apply with the corresponding bounds.*

*Proof.* We compare the bounds $g_{T,\mathrm{sch}}^{\mathrm{SGD}}$ through the cumulative quantities $\Lambda_T^{(1)}$ and $\Lambda_T^{(2)}$ that enter the explicit expression in Proposition D.10. The required monotonicity is provided by Lemma D.12.

**Constant schedule is maximal.** By Lemma D.14, among all schedules satisfying the common peak constraint $0 \leq \eta_t \leq \eta_{\max}$, the constant schedule maximizes both $\Lambda_T^{(1)}$ and $\Lambda_T^{(2)}$. Therefore, Lemma D.12 yields

$$g_{T,\text{sch}}^{\text{SGD}} \leq g_{T,\text{const}}^{\text{SGD}} \quad \text{for any schedule sch.}$$

Moreover, whenever at least one of the two cumulative inequalities is strict and the strictness condition in Lemma D.12 holds for the larger schedule (in particular, $a_{T,\text{const}} > 0$), we obtain the strict inequality $g_{T,\text{sch}}^{\text{SGD}} < g_{T,\text{const}}^{\text{SGD}}$.

**Inverse decay is asymptotically minimal relative to cosine decay.** By Lemma D.17, for sufficiently large $N := T - T_{\text{wu}}$, the inverse-decay schedule yields strictly smaller cumulative quantities $\Lambda_T^{(1)}$ and $\Lambda_T^{(2)}$ than the cosine-decay schedule. Hence, applying Lemma D.12 and its strictness clause (e.g., under $a_{T,\text{cos}} > 0$), we conclude

$$g_{T,\text{inv}}^{\text{SGD}} < g_{T,\text{cos}}^{\text{SGD}}.$$

**Cosine versus exponential decay.** Under the conditions of Lemma D.16, we have $\Lambda_{T,\text{cos}}^{(1)} \leq \Lambda_{T,\text{exp}}^{(1)}$ and $\Lambda_{T,\text{cos}}^{(2)} \leq \Lambda_{T,\text{exp}}^{(2)}$. Therefore Lemma D.12 gives

$$g_{T,\text{cos}}^{\text{SGD}} \leq g_{T,\text{exp}}^{\text{SGD}},$$

which is recorded in the statement via the conventional notation "$\lesssim$".

**Transfer to Assumption 3.4 and geometric tails.** For each schedule sch, Proposition D.10 provides an explicit bound $g_{T,\text{sch}}^{\text{SGD}}$ such that for all $k \geq 0$,

$$\mathbb{P}(\tau_{k+1} \leq T \,|\, \mathcal{F}_{\sigma_k}) \leq g_{T,\text{sch}}^{\text{SGD}} \quad \text{almost surely on } \{\sigma_k \leq T\}.$$

Hence Assumption 3.4 holds with $g_T = g_{T,\text{sch}}^{\text{SGD}}$, and the geometric-tail conclusions follow by invoking Theorem 3.6. □

### D.4.2. BATCH SIZE AND MODEL SIZE DEPENDENCY

**Proposition D.19** (Larger minibatch yields a smaller SGD re-entry bound). *Assume the hypotheses of Proposition D.10. Suppose the conditional noise proxy satisfies $\xi^2 = \xi^2(\mathsf{B})$ for minibatch size $\mathsf{B}$, where $\xi^2(\mathsf{B})$ is nonincreasing in $\mathsf{B}$. Let $g_T^{\text{SGD}}(\mathsf{B})$ denote the bound in Eq. (7) computed with $\xi^2 = \xi^2(\mathsf{B})$. Then for any $\tilde{\mathsf{B}} \geq \mathsf{B}$,*

$$g_T^{\text{SGD}}(\tilde{\mathsf{B}}) \leq g_T^{\text{SGD}}(\mathsf{B}).$$

*Consequently, if Assumption 3.4 holds with $g_T = g_T^{\text{SGD}}(\mathsf{B})$, Theorem 3.6 implies*

$$\mathbb{P}\big(K_T^{\text{eff}}(\rho) \geq k\big) \leq h_T \big(g_T^{\text{SGD}}(\mathsf{B})\big)^{k-1} + \delta_{\text{upd}} \quad (k \geq 1).$$

*Proof.* In Eq. (7), both $V_T^{\text{SGD}} = \xi^2 \Lambda_T^{(2)}$ and $B_T^{\text{SGD}} = \delta^{-1}\bar{B}_T$ are nondecreasing in $\xi^2$, hence $a_T = ((\rho - \epsilon) - B_T^{\text{SGD}})_+$ is nonincreasing in $\xi^2$. Therefore the exponential term in Eq. (7) is nondecreasing in $\xi^2$, so $g_T^{\text{SGD}}$ is nondecreasing in $\xi^2$. Since $\xi^2(\mathsf{B})$ is nonincreasing in $\mathsf{B}$, the claim follows. □

**Proposition D.20** (Batch–schedule exchange condition for preserving the re-entry bound). *Assume the hypotheses of Proposition D.10. Consider two training protocols indexed by $j \in \{1, 2\}$ with (possibly different) minibatch sizes $\mathsf{B}_j$, noise proxies $\xi_j^2$, horizons $T_j$, and schedules $\{\eta_{j,t}\}_{t=0}^{T_j-1}$. For each protocol, define*

$$\Lambda_{T_j}^{(j)} := \sum_{t=0}^{T_j-1} \eta_{j,t}, \quad (\Lambda_{T_j}^{(2)})^{(j)} := \sum_{t=0}^{T_j-1} \eta_{j,t}^2,$$

*and define $\bar{B}_{T_j}^{(j)}$ and $V_{T_j}^{(j)}$ by*

$$\bar{B}_{T_j}^{(j)} := \sqrt{\Lambda_{T_j}^{(j)}\Big(2\Delta\mathcal{L}_{\max} + \xi_j^2 \Lambda_{T_j}^{(j)} + L_{\text{sm}}\xi_j^2 (\Lambda_{T_j}^{(2)})^{(j)}\Big)}, \quad V_{T_j}^{(j)} := \xi_j^2 (\Lambda_{T_j}^{(2)})^{(j)}.$$

Let $g_{T_j}^{\text{SGD},(j)}$ denote the Proposition D.10 bound computed from $(\bar{B}_{T_j}^{(j)}, V_{T_j}^{(j)})$ (equivalently from Eq. (7)). If

$$\bar{B}_{T_2}^{(2)} \leq \bar{B}_{T_1}^{(1)} \quad and \quad V_{T_2}^{(2)} \leq V_{T_1}^{(1)},$$

then

$$g_{T_2}^{\text{SGD},(2)} \leq g_{T_1}^{\text{SGD},(1)}.$$

In particular, when $\mathsf{B}_2 \geq \mathsf{B}_1$ reduces the noise proxy (e.g., $\xi_2^2 \leq \xi_1^2$), one may trade a larger squared-step accumulation $(\Lambda_{T_2}^{(2)})^{(2)}$ against the smaller $\xi_2^2$ as long as the two budget inequalities above hold.

*Proof.* From Eq. (7), $B_T^{\text{SGD}} = \delta^{-1}\bar{B}_T$ and $V_T^{\text{SGD}} = V_T$. If $\bar{B}$ decreases then $B_T^{\text{SGD}}$ decreases and hence $a_T = ((\rho - \epsilon) - B_T^{\text{SGD}})_+$ increases. If $V$ decreases as well, the exponent $-\frac{a_T^2}{2(V + \frac{2\Delta}{3}a_T)}$ becomes no larger, so the exponential term decreases. Therefore $g_T^{\text{SGD}} = \delta_{\text{upd}} + \delta + \exp(\cdots)$ decreases, proving the claim. $\square$

**Proposition D.21** (Similarity width expansion yields smaller initial hit). *Fix a finite horizon $T$, an outer threshold $\rho > 0$, and a base boundary radius $\epsilon_0 \in (0, \rho)$ as in Definition 3.1. Consider a family of scalar coordinate processes $\{(w_t^{(\mathsf{m})})_{t \geq 0}\}_{\mathsf{m} \in \mathbb{N}}$ indexed by a width parameter $\mathsf{m}$. Assume that for every $\mathsf{m}$, Assumption 3.3 holds for $(w_t^{(\mathsf{m})})$ with update bound $\Delta = \Delta(\mathsf{m}) > 0$ and the same failure probability $\delta_{\text{upd}}$. Define the boundary radius and the initial-hit radius by*

$$\epsilon(\mathsf{m}) := \max\{\epsilon_0, \Delta(\mathsf{m})\}, \quad b_T(\mathsf{m}) := \epsilon(\mathsf{m}) + T\Delta(\mathsf{m}).$$

*Assume that for any $\tilde{\mathsf{m}} \geq \mathsf{m}$,*

$$\Delta(\tilde{\mathsf{m}}) \leq \Delta(\mathsf{m}), \tag{12}$$

$$\mathbb{P}\Big[|w_0^{(\tilde{\mathsf{m}})}| \leq x\Big] \leq \mathbb{P}\Big[|w_0^{(\mathsf{m})}| \leq x\Big] \quad for \ all \ x \geq 0. \tag{13}$$

*Let*

$$\bar{h}_T(\mathsf{m}) := \mathbb{P}\Big[|w_0^{(\mathsf{m})}| \leq b_T(\mathsf{m})\Big] + \delta_{\text{upd}}.$$

*Then $h_T^{(\mathsf{m})} := \mathbb{P}[\tau_1^{(\mathsf{m})} \leq T] \leq \bar{h}_T(\mathsf{m})$ (by Proposition D.4 applied to $(w_t^{(\mathsf{m})})$), and moreover $\bar{h}_T(\tilde{\mathsf{m}}) \leq \bar{h}_T(\mathsf{m})$.*

*Proof.* The upper bound $h_T^{(\mathsf{m})} \leq \bar{h}_T(\mathsf{m})$ was already stated above, so it remains to prove the monotonicity claim. Fix $\tilde{\mathsf{m}} \geq \mathsf{m}$. From Eq. (12) and $\epsilon(\mathsf{m}) = \max\{\epsilon_0, \Delta(\mathsf{m})\}$, we have $\epsilon(\tilde{\mathsf{m}}) \leq \epsilon(\mathsf{m})$ and hence

$$b_T(\tilde{\mathsf{m}}) = \epsilon(\tilde{\mathsf{m}}) + T\Delta(\tilde{\mathsf{m}}) \leq \epsilon(\mathsf{m}) + T\Delta(\mathsf{m}) = b_T(\mathsf{m}).$$

Apply Proposition D.4 to the process $(w_t^{(\tilde{\mathsf{m}})})$ (with its constants $\epsilon(\tilde{\mathsf{m}}), \Delta(\tilde{\mathsf{m}}), \delta_{\text{upd}}$):

$$h_T^{(\tilde{\mathsf{m}})} = \mathbb{P}(\tau_1^{(\tilde{\mathsf{m}})} \leq T) \leq \mathbb{P}\Big(|w_0^{(\tilde{\mathsf{m}})}| \leq b_T(\tilde{\mathsf{m}})\Big) + \delta_{\text{upd}} = \bar{h}_T(\tilde{\mathsf{m}}).$$

To compare $\bar{h}_T(\tilde{\mathsf{m}})$ and $\bar{h}_T(\mathsf{m})$, use the monotonicity in the threshold and the stochastic ordering Eq. (13):

$$\mathbb{P}\Big[|w_0^{(\tilde{\mathsf{m}})}| \leq b_T(\tilde{\mathsf{m}})\Big] \leq \mathbb{P}\Big[|w_0^{(\tilde{\mathsf{m}})}| \leq b_T(\mathsf{m})\Big] \leq \mathbb{P}\Big[|w_0^{(\mathsf{m})}| \leq b_T(\mathsf{m})\Big].$$

Adding $\delta_{\text{upd}}$ to both sides yields $\bar{h}_T(\tilde{\mathsf{m}}) \leq \bar{h}_T(\mathsf{m})$. $\square$

### D.4.3. SCALE-INVARIANCE ENHANCEMENT OF SIGN LOCK-IN

In architectures with BatchNorm and ReLU, the loss typically exhibits an (approximate) positive scale invariance along certain parameter blocks: rescaling a block by a positive factor can be compensated elsewhere without changing the network function, a consequence of BN-induced scale invariance together with the positive homogeneity of ReLU. In such settings, it is natural to analyze excursions in a fixed gauge, i.e., in a normalized coordinate that factors out the redundant scale.

Concretely, let $u_t \in \mathbb{R}^m$ be a parameter block containing the tracked coordinate $w_t = e_i^\top v_t$ as one of its entries, and define its block scale $r_t := \|u_t\|_2$ as well as the normalized coordinate $\widetilde{w}_t := w_t/r_t$. Intuitively, if the block scale is bounded away from zero throughout an excursion, then the normalized coordinate has smaller effective one-step increments. This reduces the overshoot-driven part of the Freedman exponent in Proposition D.10 by replacing the raw increment bound $\Delta$ with a smaller normalized bound $\widetilde{\Delta}$. We formalize the required scale control as an additional setup event, and then show how the re-entry bound tightens in this scale-invariant case.

**Additional setup (scale control).** Fix a stopping time $\theta \leq T$ (in our application, $\theta = \sigma_k$). We introduce a block-update good event $\mathcal{E}_{\mathrm{blk}}(\theta)$ on which the block $u_t$ evolves in a controlled manner: there exist constants $\Delta_{\mathrm{blk}} > 0$ and $\delta_{\mathrm{blk}} \in [0, 1)$ such that

$$\|u_{t+1} - u_t\|_2 \leq \Delta_{\mathrm{blk}} \text{ for all } t \in \{\theta, \ldots, T-1\},$$

and $\mathbb{P}(\mathcal{E}_{\mathrm{blk}}(\theta)^c \mid \mathcal{F}_\theta) \leq \delta_{\mathrm{blk}}$. This setup can be enforced, for instance, by deterministic block-wise clipping of the update (or any mechanism yielding a uniform bound on $\|u_{t+1} - u_t\|_2$). The next lemma shows that under this additional setup the block scale stays bounded away from zero over a finite horizon, and that the normalized coordinate $\widetilde{w}_t = w_t/r_t$ admits a smaller increment bound.

**Lemma D.22** (Block-scale lower bound and normalized increment bound). *Let $u_t \in \mathbb{R}^m$ be a parameter block that contains the tracked coordinate $w_t = e_i^\top v_t$ as one of its entries, and define the block scale*

$$r_t := \|u_t\|_2.$$

*Fix a stopping time $\theta \leq T$ and assume that there exist constants $\Delta_{\mathrm{blk}} > 0$ and $\delta_{\mathrm{blk}} \in [0, 1)$ such that on the good event $\mathcal{E}_{\mathrm{blk}}(\theta)$ we have the uniform block update bound*

$$\|u_{t+1} - u_t\|_2 \leq \Delta_{\mathrm{blk}} \text{ for all } t \in \{\theta, \ldots, T-1\},$$

*and $\mathbb{P}(\mathcal{E}_{\mathrm{blk}}(\theta)^c \mid \mathcal{F}_\theta) \leq \delta_{\mathrm{blk}}$.*

*Then on $\mathcal{E}_{\mathrm{blk}}(\theta)$, for all $t \in \{\theta, \ldots, T\}$,*

$$r_\theta - (t - \theta)\Delta_{\mathrm{blk}} \leq r_t \leq r_\theta + (t - \theta)\Delta_{\mathrm{blk}}.$$

*In particular, if $r_\theta > T\Delta_{\mathrm{blk}}$ then $r_t \geq r_{\min} := r_\theta - T\Delta_{\mathrm{blk}} > 0$ for all $t \leq T$.*

*Moreover, define the normalized coordinate $\widetilde{w}_t := w_t/r_t$. On $\mathcal{E}_{\mathrm{blk}}(\theta) \cap \{r_t \geq r_{\min} \ \forall t \leq T\}$, we have*

$$|\widetilde{w}_{t+1} - \widetilde{w}_t| \leq \frac{2\Delta_{\mathrm{blk}}}{r_{\min}} \text{ for all } t \in \{\theta, \ldots, T-1\}.$$

*Proof.* On $\mathcal{E}_{\mathrm{blk}}(\theta)$, the triangle inequality yields $|r_{t+1} - r_t| = |\|u_{t+1}\|_2 - \|u_t\|_2| \leq \|u_{t+1} - u_t\|_2 \leq \Delta_{\mathrm{blk}}$, so telescoping gives the two-sided bound on $(r_t)$.

For the normalized increment, write

$$\widetilde{w}_{t+1} - \widetilde{w}_t = \frac{w_{t+1}}{r_{t+1}} - \frac{w_t}{r_t} = \frac{w_{t+1} - w_t}{r_{t+1}} + w_t \left( \frac{1}{r_{t+1}} - \frac{1}{r_t} \right).$$

On $\mathcal{E}_{\mathrm{blk}}(\theta)$ we have $|w_{t+1} - w_t| \leq \|u_{t+1} - u_t\|_2 \leq \Delta_{\mathrm{blk}}$ and $|r_{t+1} - r_t| \leq \|u_{t+1} - u_t\|_2 \leq \Delta_{\mathrm{blk}}$. Moreover, since $|w_t| \leq \|u_t\|_2 = r_t$, we have $|\widetilde{w}_t| = |w_t|/r_t \leq 1$. Therefore, on $\mathcal{E}_{\mathrm{blk}}(\theta) \cap \{r_s \geq r_{\min} \ \forall s \leq T\}$ (so that $r_{t+1} \geq r_{\min}$),

$$\begin{aligned}
|\widetilde{w}_{t+1} - \widetilde{w}_t| &\leq \frac{|w_{t+1} - w_t|}{r_{t+1}} + |w_t| \left| \frac{1}{r_{t+1}} - \frac{1}{r_t} \right| \\
&= \frac{|w_{t+1} - w_t|}{r_{t+1}} + \left| \frac{w_t}{r_t} \right| \frac{|r_{t+1} - r_t|}{r_{t+1}} \\
&\leq \frac{\Delta_{\mathrm{blk}}}{r_{\min}} + 1 \cdot \frac{\Delta_{\mathrm{blk}}}{r_{\min}} = \frac{2\Delta_{\mathrm{blk}}}{r_{\min}}.
\end{aligned}$$

This proves the normalized increment bound with $\widetilde{\Delta} := 2\Delta_{\mathrm{blk}}/r_{\min}$. $\qquad \square$

**Proposition D.23** (Scale control yields a tighter increment term). *Assume the setting of Proposition D.10. In addition, suppose there exists a parameter block $u_t$ containing the tracked coordinate $w_t$ such that (motivated by BN-induced scale invariance together with ReLU positive homogeneity) its scale stays bounded away from zero on the excursion with high probability: for $\theta = \sigma_k$, Lemma D.22 holds with constants $\Delta_{\mathrm{blk}}$, $\delta_{\mathrm{blk}}$, and $r_{\min} := r_{\sigma_k} - T\Delta_{\mathrm{blk}} > 0$.*

*Define the normalized coordinate $\widetilde{w}_t := w_t/r_t$ and the corresponding oriented process $\widetilde{z}_t := s_k \widetilde{w}_t$. On the event $\mathcal{E}_{\mathrm{blk}}(\sigma_k) \cap \{r_t \geq r_{\min} \ \forall t \leq T\}$, Lemma D.22 implies the normalized increment bound $|\widetilde{w}_{t+1} - \widetilde{w}_t| \leq \widetilde{\Delta}$ with*

$$\widetilde{\Delta} := \frac{2\Delta_{\mathrm{blk}}}{r_{\min}}.$$

*Consequently, the proof of Proposition D.10 yields the same form of re-entry bound Eq. (8) but with the increment term $\Delta$ replaced by $\widetilde{\Delta}$, at an additional failure cost $\delta_{\mathrm{blk}}$:*

$$\mathbb{P}[\tau_{k+1} \leq T \mid \mathcal{F}_{\sigma_k}] \leq g_T^{\mathrm{Iv}} \quad \text{on } \{\sigma_k \leq T\},$$

*where*

$$g_T^{\mathrm{Iv}} := (\delta_{\mathrm{upd}} + \delta_{\mathrm{blk}}) + \delta + \exp\left(-\frac{a_T^2}{2\left(V_T^{\mathrm{SGD}} + \frac{2\widetilde{\Delta}}{3} a_T\right)}\right),$$

*and $a_T$ and $V_T^{\mathrm{SGD}}$ are exactly as defined in Eq. (7).*

*Proof.* Repeat the proof of Proposition D.10, but intersect the "good" event $\mathcal{E}_\Delta(\sigma_k)$ with $\mathcal{E}_{\mathrm{blk}}(\sigma_k) \cap \{r_t \geq r_{\min} \ \forall t \leq T\}$. By Lemma D.22, on this intersection we have the normalized increment bound $|\widetilde{z}_{t+1} - \widetilde{z}_t| \leq \widetilde{\Delta}$ (and hence $|X_t| \leq 2\widetilde{\Delta}$ in the Freedman step), while the drift and variance parts are handled exactly as in Proposition D.10. The additional failure probability contributes $\delta_{\mathrm{blk}}$ by a union bound. $\qquad\square$

### D.4.4. SIGN DYNAMICS

This section records auxiliary consequences for sign dynamics implied by the sign lock-in theory. In particular, we characterize the temporal distribution of effective sign changes and show that sign flips are inherently front-loaded, occurring predominantly in the early phase of training. The resulting stabilization of signs explains why later optimization mainly affects magnitudes rather than signs.

**Lemma D.24** (Initial-hit bound without assuming an outer start). *Assume the hypotheses of Proposition D.10. Let $h_T := \mathbb{P}(\tau_1 \leq T)$ be the initial-hit factor, where $\sigma_0, \tau_1$ are defined in Definition 3.2. For each learning-rate schedule $\mathrm{sch} \in \{\mathrm{const}, \mathrm{cos}, \exp, \mathrm{inv}\}$, let $g_{T,\mathrm{sch}}^{\mathrm{SGD}}$ denote the Proposition D.10 bound computed with that schedule.*

*Then, without assuming $\sigma_0 = 0$,*

$$h_T \leq g_{T,\mathrm{sch}}^{\mathrm{SGD}}. \tag{14}$$

*Consequently, the schedule ordering in Proposition D.18 carries over to the initial-hit upper bounds.*

*Proof.* By Definition 3.2, $\tau_1 := \inf\{t > \sigma_0 : |w_t| \leq \epsilon\}$, hence $\{\tau_1 \leq T\} \subseteq \{\sigma_0 \leq T\}$. Therefore, by the tower property,

$$h_T = \mathbb{P}[\tau_1 \leq T] = \mathbb{E}[\mathbb{P}[\tau_1 \leq T \mid \mathcal{F}_{\sigma_0}] \mathbf{1}\{\sigma_0 \leq T\}].$$

Applying Proposition D.10 with $k = 0$ yields $\mathbb{P}[\tau_1 \leq T \mid \mathcal{F}_{\sigma_0}] \leq g_{T,\mathrm{sch}}^{\mathrm{SGD}}$ on $\{\sigma_0 \leq T\}$. Thus,

$$h_T \leq \mathbb{E}\big[g_{T,\mathrm{sch}}^{\mathrm{SGD}} \mathbf{1}\{\sigma_0 \leq T\}\big] \leq g_{T,\mathrm{sch}}^{\mathrm{SGD}},$$

which proves Eq. (14). The final statement follows by combining with Proposition D.18. $\qquad\square$

**Lemma D.25** (Early initiation of sign flips). *Fix a finite horizon $T$ and consider the stopping times $\tau_k$ and $\sigma_k$ defined in Section 3. Under Theorem 3.6's hypotheses, any effective outer-to-outer sign flip up to time $T$ must be preceded by an early boundary interaction: specifically, for any $k \geq 1$,*

$$\{\sigma_k \leq T, \ \mathrm{sign}(w_{\sigma_k}) \neq \mathrm{sign}(w_{\sigma_{k-1}})\} \cap \mathcal{E}_\Delta \subseteq \{\tau_1 \leq \tau_k \leq T\}.$$

*Moreover, the probability that a new effective sign-flip sequence is initiated late in training is exponentially suppressed: for all $k \geq 1$,*

$$\mathbb{P}[\tau_k \leq T] \leq h_T \, g_T^{k-1},$$

*where $h_T := \mathbb{P}[\tau_1 \leq T]$ is the initial-hit factor and $g_T \in (0, 1)$ is the Re-entry bound from Assumption 3.4.*

*Proof.* We separate the claim into two parts.

**A sign change at $\sigma_k$ implies an early boundary interaction.** Fix $k \geq 1$ and consider the event

$$\{\sigma_k \leq T, \ \text{sign}(w_{\sigma_k}) \neq \text{sign}(w_{\sigma_{k-1}})\} \cap \mathcal{E}_\Delta.$$

By Definition 3.2, an *effective outer-to-outer sign flip* is counted only when the trajectory (i) starts from an outer state at time $\sigma_{k-1}$, (ii) reaches the boundary band at time $\tau_k$, and then (iii) exits to the opposite outer side at time $\sigma_k$. Hence, on the above event, the boundary-hit time $\tau_k$ must exist and satisfy

$$\tau_k \leq \sigma_k \leq T.$$

Also, since $(\tau_j)_j$ is nondecreasing by construction of successive excursions, the first boundary-hit time satisfies $\tau_1 \leq \tau_k$. Combining these relations gives

$$\tau_1 \leq \tau_k \leq T,$$

which proves

$$\{\sigma_k \leq T, \ \text{sign}(w_{\sigma_k}) \neq \text{sign}(w_{\sigma_{k-1}})\} \cap \mathcal{E}_\Delta \subseteq \{\tau_1 \leq \tau_k \leq T\}.$$

**Geometric suppression of late initiations.** Theorem 3.6 states that under the same hypotheses, for every $k \geq 1$,

$$\mathbb{P}[\tau_k \leq T] \leq h_T g_T^{k-1},$$

where $h_T = \mathbb{P}(\tau_1 \leq T)$ and $g_T \in (0,1)$ is the re-entry factor. This is exactly the second displayed inequality in the lemma.

Therefore both statements hold. $\qquad\square$

**Remark D.26** (Front-loaded structure of sign changes). *Lemma D.25 shows that sign flips are front-loaded in the training trajectory. Importantly, this statement does not rely on specifying a concrete "early-time" window or on particular learning-rate schedules. Rather, it follows from the stopping-time structure and the geometric decay of boundary-hit probabilities: a parameter that flips its sign at any point during training must have already interacted with the boundary neighborhood at an earlier stage. Conversely, parameters that do not approach the boundary early are unlikely to exhibit sign flips later, regardless of the remaining training duration.*

### D.4.5. NUMERICAL VALIDATION OF PRACTICAL INSIGHTS

We follow the CharLM setup in Appendix C.4, and only change the learning-rate schedule. Data: Tiny Shakespeare with a contiguous 90/10 train/validation split; next-character prediction on random contiguous blocks with sequence length $L = 64$ and batch size $B = 64$. Model: TinyCharLM (causal Transformer) with $d_{\text{model}} = 128$, $n_{\text{layers}} = 2$, $n_{\text{heads}} = 4$, $d_{\text{ff}} = 256$, context length 256, and no dropout. Optimization: AdamW with initialization scale $\sigma_{\text{init}} = 0.02$; all runs use three seeds (0/1/2) and the same training horizon $T$ and peak step size $\eta_{\text{max}}$.

Schedules: constant learning rate $\eta_t = \eta_{\text{max}}$; and warmup of length $T_{\text{wu}}$ followed by (i) cosine decay, (ii) exponential decay with factor $\gamma$, and (iii) inverse decay with exponent $p$ (as defined in this section). We compute $K := K_T^{\text{eff}}(\rho)$ from tracked weights, fit the zero-inflated geometric model to obtain $\hat{h}$ and $\hat{g}$, and report mean±std across seeds.

We next validate the batchsize dependency. We fix the optimization setup above and only change the training minibatch size $B \in \{1, 2, 4, 8, 16, 32, 64, 128, 256, 512, 1024\}$, keeping the same training horizon $T$ and peak step size $\eta_{\text{max}}$. Figure 14 shows that the estimated re-entry parameter $\hat{g}$ decreases monotonically as $B$ increases (mean±std across seeds), supporting the batchsize dependency in Proposition D.19.

Finally, we empirically validate the width dependence predicted by Corollary D.21. Using the CharLM setup described in Appendix C.4, we sweep the model width $d_{\text{model}} \in \{32, 64, 128, 256, 512\}$ while keeping all other training hyperparameters fixed, including the initialization scale $\sigma_{\text{init}}$. For each configuration, we track the effective flip count $K_{\text{eff}, T}(\rho)$ and fit the zero-inflated geometric model (Appendix C.4.2) to estimate the lock-in parameters $(\hat{h}, \hat{g})$. Figure 15 shows that both the initial-hit factor $\hat{h}$ and the re-entry ratio $\hat{g}$ decrease monotonically as the width increases. The reduction in $\hat{h}$ indicates that fewer weights ever reach the near-zero boundary, consistent with Proposition D.21.

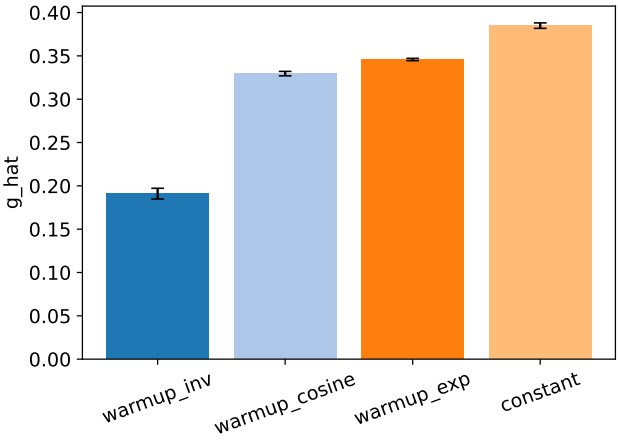

Figure 12. Estimated $\hat{g}$ (mean±std) across schedules.

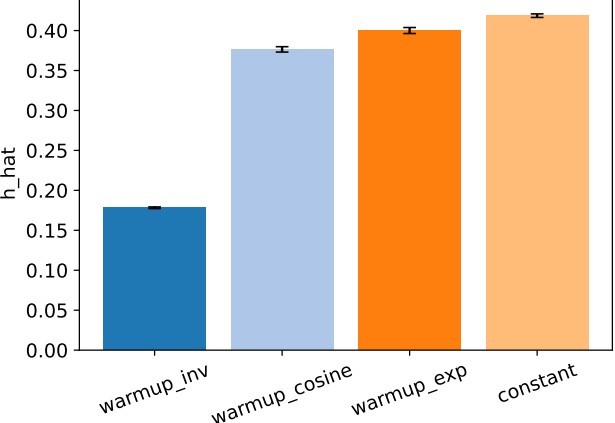

Figure 13. Estimated $\hat{h}$ (mean±std) across schedules.

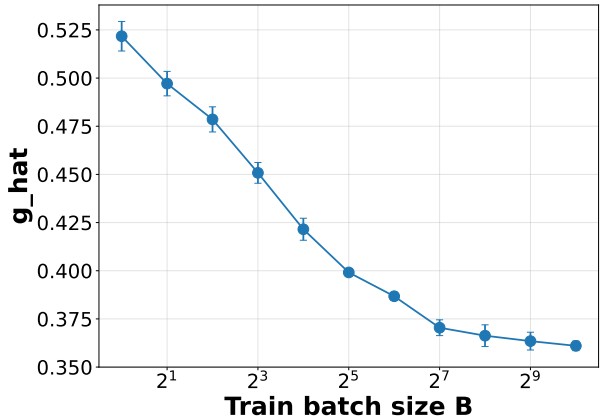

Figure 14. Estimated $\hat{g}$ (mean±std across seeds) vs. train batch size $B$.

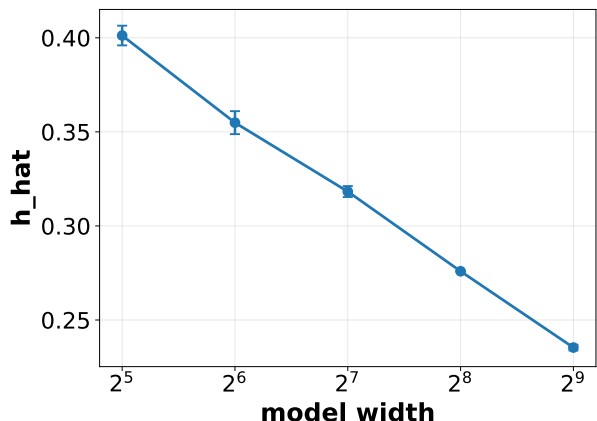

Figure 15. Estimated initial-hit factor $\hat{h}$ vs. model width $d_{\mathrm{model}}$.

### D.5. Theory for Sign Lock-In Enhancement

Having established the deterministic and stochastic foundations of sign lock-in, we now turn to theory-driven mechanisms that actively enhance this effect. The results in Appendix D.2 show that the initial-hit factor $h_T$ is largely dominated by initialization under bounded updates, while Appendix D.3 demonstrates how stochastic optimization dynamics control the re-entry ratio $g_T$. This Appendix builds on these insights to formalize practical interventions that reduce boundary visits and re-entry, thereby strengthening sign lock-in beyond its naturally emerging regime.

**Proposition D.27** (Gap initialization suppresses the initial-hit factor). *Assume the setting of Proposition D.4. Under the gap initialization described in Section 4, let $Z \sim \mathcal{N}(0, \sigma_{\mathrm{init}}^2)$ and set $|w_0| = |Z|$ conditioned on $|Z| \geq a_{\mathrm{init}}$. Then*

$$h_T \leq \mathbb{P}\big[|Z| \leq b_T \mid |Z| \geq a_{\mathrm{init}}\big] + \delta_{\mathrm{upd}} = \frac{\mathbb{P}[a_{\mathrm{init}} \leq |Z| \leq b_T]}{\mathbb{P}[|Z| \geq a_{\mathrm{init}}]} + \delta_{\mathrm{upd}}.$$

*Here $b_T := \epsilon + T\Delta$. In particular, if $b_T < a_{\mathrm{init}}$, then $h_T \leq \delta_{\mathrm{upd}}$.*

*Proof.* By Proposition D.4, $h_T \leq \mathbb{P}[|w_0| \leq b_T] + \delta_{\mathrm{upd}}$ with $b_T = \epsilon + T\Delta$. Under gap initialization, $|w_0|$ is distributed as $|Z|$ conditioned on $|Z| \geq a_{\mathrm{init}}$. Therefore,

$$h_T = \mathbb{P}[\tau_1 \leq T] \leq \mathbb{P}[|w_0| \leq b_T] + \delta_{\mathrm{upd}} = \mathbb{P}\big[|Z| \leq b_T \mid |Z| \geq a_{\mathrm{init}}\big] + \delta_{\mathrm{upd}},$$

which proves Proposition D.27. If $b_T < a_{\mathrm{init}}$, the conditional probability term vanishes, and hence $h_T \leq \delta_{\mathrm{upd}}$. □

**Proposition D.28** (Outer-drift implies an explicit Re-entry bound). *Assume Assumption 3.3 and suppose additionally that there exists a constant $\mu > 0$ such that*

$$\mathbb{E}\big[\,|w_{t+1}| - |w_t|\,\big|\,\mathcal{F}_t\big] \geq \mu \ \text{almost surely on } \{|w_t| > \epsilon\}. \tag{15}$$

*Then for every $k \geq 0$, on the event $\{\sigma_k \leq T\}$,*

$$\mathbb{P}\big[\tau_{k+1} \leq T \,\big|\, \mathcal{F}_{\sigma_k}\big] \leq \delta_{\mathrm{upd}} + \exp\Big(-\frac{2\mu(\rho - \epsilon)}{\Delta^2}\Big). \tag{16}$$

*Consequently, Assumption 3.4 holds with the (time-uniform) choice*

$$g_T^{\mathrm{OD}} := \delta_{\mathrm{upd}} + \exp\Big(-\frac{2\mu(\rho - \epsilon)}{\Delta^2}\Big),$$

*whenever $g_T^{\mathrm{OD}} < 1$.*

**Interpretation for the log-barrier.** *In the presence of the log-barrier term in Section 4, a sufficient condition for Eq. (15) is that the regularizer-induced outward push dominates any (inward) bias from the task update near the boundary. One convenient way to express this is via a lower bound $\mu \approx \gamma\lambda/(\rho + \epsilon_{\mathrm{lb}}) - \beta$ (notation as in the main text), whenever the barrier is active throughout the band $(\epsilon, \rho]$.*

*Proof.* Fix $k \geq 0$ and work on $\{\sigma_k \leq T\}$. If $\sigma_k = T$ then $\tau_{k+1} > T$ by definition, so Eq. (16) is trivial. Hence assume $\sigma_k \leq T - 1$.

Define the *first update-violation time*

$$\eta := \inf\{t \geq \sigma_k : \ |w_{t+1} - w_t| > \Delta\},$$

with $\inf \emptyset = \infty$. By Assumption 3.3 applied at $\theta = \sigma_k$, $\mathbb{P}(\eta \leq T - 1 \mid \mathcal{F}_{\sigma_k}) \leq \delta_{\mathrm{upd}}$.

Define the shifted filtration $\mathcal{G}_t := \mathcal{F}_{\sigma_k + t}$ and the shifted process $r'_t := |w_{\sigma_k + t}|$ for $t \geq 0$. Let

$$\tau' := \inf\{t \geq 1 : \ r'_t \leq \epsilon\}, \quad \eta' := \inf\{t \geq 0 : \ \sigma_k + t \geq \eta\}.$$

Then $\tau_{k+1} = \sigma_k + \tau'$ and $\{\eta > T - 1\}$ is equivalent to $\{\eta' > T - \sigma_k - 1\}$.

On the event $\{t < \eta'\}$ we have $|w_{\sigma_k + t + 1} - w_{\sigma_k + t}| \leq \Delta$ and since $x \mapsto |x|$ is 1-Lipschitz,

$$|r'_{t+1} - r'_t| \leq |w_{\sigma_k + t + 1} - w_{\sigma_k + t}| \leq \Delta. \tag{17}$$

Moreover, by Eq. (15), for all $t < \tau'$ we have $\mathbb{E}[r'_{t+1} - r'_t \mid \mathcal{G}_t] \geq \mu$.

Let $\kappa := 2\mu/\Delta^2$ and define

$$M_t := \exp\big(-\kappa\, r'_{t \wedge \tau' \wedge \eta'}\big).$$

We claim that $(M_t)_{t \geq 0}$ is a supermartingale with respect to $(\mathcal{G}_t)$. Indeed, for $t < \tau' \wedge \eta'$, set $X_t := -(r'_{t+1} - r'_t)$. Then $X_t \in [-\Delta, \Delta]$ by Eq. (17) and $\mathbb{E}[X_t \mid \mathcal{G}_t] \leq -\mu$. Hoeffding's lemma gives

$$\mathbb{E}\big[e^{\kappa X_t} \mid \mathcal{G}_t\big] \leq \exp\Big(\kappa\,\mathbb{E}[X_t \mid \mathcal{G}_t] + \frac{\kappa^2\Delta^2}{2}\Big) \leq \exp\Big(-\kappa\mu + \frac{\kappa^2\Delta^2}{2}\Big) = 1,$$

where the last equality uses $\kappa = 2\mu/\Delta^2$. Therefore $\mathbb{E}[M_{t+1} \mid \mathcal{G}_t] \leq M_t$ for $t < \tau' \wedge \eta'$, and for $t \geq \tau' \wedge \eta'$ we have $M_{t+1} = M_t$, so $(M_t)$ is a supermartingale.

Fix an integer $n \geq 1$. By optional stopping applied to the bounded stopping time $(\tau' \wedge \eta') \wedge n$,

$$\mathbb{E}[M_{(\tau' \wedge \eta') \wedge n} \mid \mathcal{G}_0] \leq M_0 = \exp(-\kappa r'_0).$$

On the event $\{\tau' \leq n, \ \eta' > n\}$ we have $r'_{\tau'} \leq \epsilon$, hence $M_{(\tau' \wedge \eta') \wedge n} = M_{\tau'} = \exp(-\kappa r'_{\tau'}) \geq \exp(-\kappa\epsilon)$. Thus,

$$\exp(-\kappa\epsilon)\,\mathbb{P}(\tau' \leq n, \ \eta' > n \mid \mathcal{G}_0) \leq \mathbb{E}[M_{(\tau' \wedge \eta') \wedge n} \mid \mathcal{G}_0] \leq \exp(-\kappa r'_0),$$

which gives

$$\mathbb{P}(\tau' \leq n,\ \eta' > n \mid \mathcal{F}_{\sigma_k}) \leq \exp\big(-\kappa(r_0' - \epsilon)\big) \leq \exp\big(-\kappa(\rho - \epsilon)\big) = \exp\Big(-\frac{2\mu(\rho - \epsilon)}{\Delta^2}\Big).$$

Finally, taking $n = T - \sigma_k$ and using

$$\mathbb{P}(\tau_{k+1} \leq T \mid \mathcal{F}_{\sigma_k}) \leq \mathbb{P}(\eta \leq T - 1 \mid \mathcal{F}_{\sigma_k}) + \mathbb{P}(\tau' \leq T - \sigma_k,\ \eta' > T - \sigma_k - 1 \mid \mathcal{F}_{\sigma_k})$$

yields the Proposition D.28. $\qquad\square$

**Proposition D.29** (Flip-histogram bound under Gap initialization and outer-drift)**.** *Assume the hypotheses of Proposition D.27 (Gap initialization) and Proposition D.28, and keep the definitions of $K_T^{\mathrm{eff}}(\rho)$, $\tau_k$, $\sigma_k$ from Section 3. Then, for all $k \geq 1$,*

$$\mathbb{P}\big[K_T^{\mathrm{eff}}(\rho) \geq k\big] \leq h_T^{\mathrm{gap}}\,(g_T^{\mathrm{OD}})^{k-1} + \delta_{\mathrm{upd}},\quad g_T^{\mathrm{OD}} = \delta_{\mathrm{upd}} + \exp\Big(-\frac{2\mu(\rho - \epsilon)}{\Delta^2}\Big). \tag{18}$$

*In particular, since $K_T^{\mathrm{eff}}(\rho) \leq T$, the expected effective flip count admits the bound*

$$\mathbb{E}\big[K_T^{\mathrm{eff}}(\rho)\big] = \sum_{k=1}^{T} \mathbb{P}\big[K_T^{\mathrm{eff}}(\rho) \geq k\big] \leq \frac{h_T^{\mathrm{gap}}}{1 - g_T^{\mathrm{OD}}} + T\,\delta_{\mathrm{upd}},$$

*whenever $g_T^{\mathrm{OD}} < 1$.*

*Proof.* Corollary D.28 verifies Assumption 3.4 with $g_T = g_T^{\mathrm{OD}}$. Applying Theorem 3.6 then gives $\mathbb{P}[\tau_k \leq T] \leq h_T(g_T^{\mathrm{OD}})^{k-1}$. Replacing $h_T$ by the Gap-initialization probability $h_T^{\mathrm{gap}}$ (Corollary D.27) and using Eq. (4) in Theorem 3.6 yields Eq. (18).

Finally, since $K_T^{\mathrm{eff}}(\rho) \leq T$, we can sum the tail bound up to $T$:

$$\mathbb{E}[K_T^{\mathrm{eff}}(\rho)] = \sum_{k=1}^{T} \mathbb{P}[K_T^{\mathrm{eff}}(\rho) \geq k] \leq \sum_{k=1}^{T} \Big(h_T^{\mathrm{gap}}(g_T^{\mathrm{OD}})^{k-1} + \delta_{\mathrm{upd}}\Big) \leq \frac{h_T^{\mathrm{gap}}}{1 - g_T^{\mathrm{OD}}} + T\,\delta_{\mathrm{upd}}.$$

$\qquad\square$

## D.6. Sign Floating Mode

Although sign lock-in is generally expected to occur, one can artificially establish conditions to break this lock-in. Here, we demonstrate an example of this approach. This sign floating mode is induced by an auxiliary regularizer with an extraordinarily large weight, and should be viewed as an artificial regime rather than the typical behavior of standard deep-network training.

### D.6.1. THEORY FOR SIGN FLOATING MODE

We keep the same stopping-time framework as in Section 3: outer-entry times $(\sigma_k)_{k \geq 0}$, boundary-hit times $(\tau_k)_{k \geq 1}$, and the effective outer-to-outer flip count $K_T^{\mathrm{eff}}(\rho)$. In sign floating mode, boundary re-entries are frequent and, crucially, the sign upon exiting back to the outer region is *well-mixed* (approximately random). This yields a binomial law for the histogram of effective flips.

**Assumption D.30** (Floating regime: excursion-count concentration up to $T$)**.** *There exist an integer $\bar{n}_T \geq 0$ and parameters $\Delta_T \geq 0$ and $\delta_T \in [0, 1)$ such that*

$$\mathbb{P}\big(|N_T(\rho) - \bar{n}_T| \leq \Delta_T\big) \ \geq\ 1 - \delta_T,$$

*where the completed outer-entry count is $N_T(\rho) := \max\{k \geq 0 : \sigma_k \leq T\}$. In the induced floating regime, $\bar{n}_T$ is typically moderate to large.*

**Assumption D.31** (Sign mixing at outer re-entry)**.** *Define the outer-entry sign sequence*

$$Z_k := \mathrm{sign}(w_{\sigma_k}) \in \{\pm 1\},\quad k \geq 0,$$

*(on $\{\sigma_k \leq T\}$ this is well-defined since $|w_{\sigma_k}| \geq \rho > 0$). Assume that, for every $k \geq 1$,*

$$\mathbb{P}[Z_k = +1 \mid \mathcal{F}_{\tau_k}] = \mathbb{P}[Z_k = -1 \mid \mathcal{F}_{\tau_k}] = \tfrac{1}{2},$$

*and moreover, conditionally on the completed outer-entry count*

$$N_T(\rho) := \max\{k \geq 0 : \sigma_k \leq T\},$$

*the random variables $(Z_1, Z_2, \ldots, Z_{N_T(\rho)})$ are independent. (No assumption is imposed on $Z_0$.)*

**Remark D.32.** *Assumption D.30 posits that the completed outer-entry count $N_T(\rho)$ is effectively fixed (or tightly concentrated) up to time $T$, so boundary excursions are not governed by the "rare re-entry" regime behind Theorem 3.6. However, having many excursions alone does not determine the histogram shape of $K_T^{\mathrm{eff}}(\rho)$. The binomial law in Theorem D.33 is driven by Assumption D.31, which enforces (approximately) symmetric sign re-randomization upon each outer re-entry; Assumption D.30 controls how the resulting conditional binomial laws mix through $N_T(\rho)$ (see the remark below).*

**Theorem D.33** (Binomial histogram of effective flips in sign floating mode)**.** *Assume Assumption D.31. Conditionally on $N_T(\rho) = n$, the effective flip count satisfies*

$$K_T^{\mathrm{eff}}(\rho) \mid \{N_T(\rho) = n\} \sim \begin{cases} \delta_0, & n = 0, \\ \mathrm{Binomial}\left(n, \tfrac{1}{2}\right), & n \geq 1. \end{cases}$$

*Proof.* Fix $n \geq 1$ and condition on $\{N_T(\rho) = n\}$. On this event,

$$K_T^{\mathrm{eff}}(\rho) = \sum_{k=1}^{n} \mathbf{1}\{Z_k \neq Z_{k-1}\}.$$

Define flip indicators $I_k := \mathbf{1}\{Z_k \neq Z_{k-1}\} \in \{0, 1\}$ for $k = 1, \ldots, n$. Let $B_k := (1 + Z_k)/2 \in \{0, 1\}$. Under Assumption D.31, $(B_1, \ldots, B_n)$ are i.i.d. Bernoulli$(1/2)$, and we impose no constraint on $B_0$.

Condition on $B_0 = b_0 \in \{0, 1\}$. Then $(B_1, \ldots, B_n)$ is uniform on $\{0, 1\}^n$. Moreover,

$$I_k = \mathbf{1}\{Z_k \neq Z_{k-1}\} = B_k \oplus B_{k-1},$$

where $\oplus$ denotes XOR on $\{0, 1\}$. Consider the mapping

$$\Phi_{b_0} : (B_1, \ldots, B_n) \mapsto (I_1, \ldots, I_n) \text{ with fixed } B_0 = b_0.$$

This map is a bijection: given $(I_1, \ldots, I_n)$, we recover recursively

$$B_k = b_0 \oplus I_1 \oplus \cdots \oplus I_k.$$

Hence, conditional on $B_0 = b_0$, the vector $(I_1, \ldots, I_n)$ is uniform on $\{0, 1\}^n$, so $I_1, \ldots, I_n$ are i.i.d. Bernoulli$(1/2)$. Since the conditional law does not depend on $b_0$, the same holds given $N_T(\rho) = n$. Therefore, $K_T^{\mathrm{eff}}(\rho) = \sum_{k=1}^{n} I_k \sim$ Binomial$(n, 1/2)$. $\square$

**Remark D.34** (Unconditional law is a mixture)**.** *Unconditionally, $K_T^{\mathrm{eff}}(\rho)$ is a mixture over the random excursion count:*

$$\mathbb{P}\big[K_T^{\mathrm{eff}}(\rho) = k\big] = \sum_{n \geq k} \mathbb{P}[N_T(\rho) = n] \binom{n}{k} \left(\tfrac{1}{2}\right)^n.$$

*Thus, the empirical flip histogram is binomial-shaped once the layer-wise excursion count $n$ is effectively fixed (or tightly concentrated).*

### D.6.2. INDUCTION OF SIGN FLOATING

**Inward Drift.** To induce sign floating, we assume the *opposite* radial drift to Proposition D.28: the dynamics are biased *toward* the sign boundary on the band $(\epsilon, \rho]$. Concretely, we postulate the existence of $\mu_{\mathrm{ID}} > 0$ such that

$$\mathbb{E}\big[\,|w_{t+1}| - |w_t| \,\big|\, \mathcal{F}_t\big] \leq -\mu_{\mathrm{ID}} \text{ almost surely on } \{\epsilon < |w_t| \leq \rho\}. \tag{19}$$

**One concrete enhancement method: local $l_1$ attraction.**  A simple way to realize Eq. (19) is to add a *local $l_1$* penalty that is active only near the boundary:

$$R_{\mathrm{ID}}(W^{(l)}; \rho_f) := \frac{1}{mn} \sum_{i,j} \min\left\{|W_{ij}^{(l)}|, \ \rho_f\right\}, \ \ l \in \mathcal{M}_{\mathrm{float}},$$

and optimize

$$\mathcal{L}_{\mathrm{float}}(\theta) = \mathcal{L}_{\mathrm{task}}(\theta) + \lambda_{\mathrm{float}} \sum_{l \in \mathcal{M}_{\mathrm{float}}} R_{\mathrm{ID}}(W^{(l)}; \rho_f), \ \ \lambda_{\mathrm{float}} > 0.$$

For a single coordinate update (e.g., SGD with step size $\gamma$), the added term contributes approximately $-\gamma\lambda_{\mathrm{float}}\operatorname{sign}(w_t)$ when $0 < |w_t| < \rho_f$, which decreases $|w_t|$ in expectation and thus promotes repeated boundary visits, enabling sign floating.

### D.6.3. EXPERIMENTAL VALIDATION

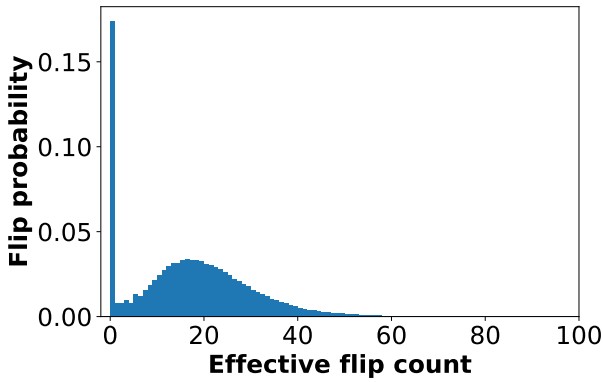

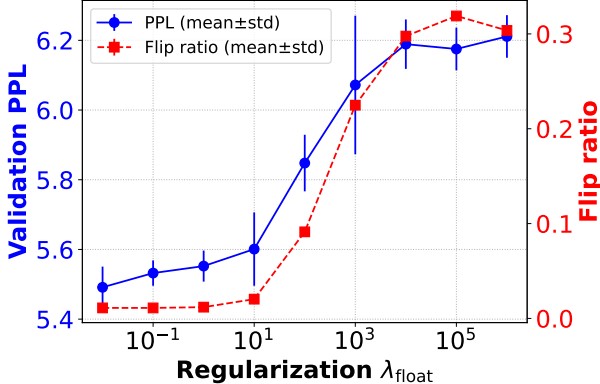

*(a)* Histogram of effective flip counts $K_{\mathrm{eff},T}(\rho)$ in a representative floating-mode run ($\lambda_{\mathrm{float}} = 10^4$).

*(b)* Sweep over $\lambda_{\mathrm{float}}$: validation PPL and flip ratio at training end (mean±std over 3 seeds).

*Figure 16.* **Empirical induction of sign floating mode via local $l_1$ attraction** (Appendix D.6).

**Setup.**  We empirically induce the sign floating mode by adding an inward-drift regularizer that attracts weights toward the sign boundary. Concretely, we train the same Tiny Shakespeare character-level Transformer (CharLM) setting as in Appendix C.4 (sequence length 64, batch size 64; AdamW optimizer used), but we optimize the following objective:

$$\mathcal{L}_{\mathrm{float}}(\theta) = \mathcal{L}_{\mathrm{task}}(\theta) + \lambda_{\mathrm{float}} \sum_{l \in \mathcal{M}_{\mathrm{float}}} R_{\mathrm{ID}}\left(W^{(l)}; \rho_f\right),$$

where $\mathcal{M}_{\mathrm{float}}$ denotes the set of matrix-shaped parameters (2D tensors), and the local $l_1$ attraction is

$$R_{\mathrm{ID}}(W; \rho_f) := \frac{1}{mn} \sum_{i,j} \min\left\{|W_{ij}|, \rho_f\right\}.$$

In all runs, we set $\rho_f = 10^{-2}$ and train for $T = 2000$ steps. We sweep

$$\lambda_{\mathrm{float}} \in \{10^{-2}, 10^{-1}, 1.0, 10, 10^2, 10^3, 10^4, 10^5, 10^6\},$$

and report mean±std over 3 random seeds.

**Result.**  Figure 16 summarizes the behavior. As $\lambda_{\mathrm{float}}$ increases, the step-wise flip ratio increases sharply, indicating frequent sign changes consistent with sign floating. At the same time, the validation PPL degrades moderately, reflecting the cost of forcing weights to repeatedly approach the sign boundary. Moreover, the histogram of $K_{\mathrm{eff},T}(\rho)$ becomes broad, rather than geometrically decaying as in the lock-in regime, qualitatively matching the binomial-shaped flip histogram predicted in Theorem D.33 when outer re-entries are frequent and the re-entry sign is well-mixed. We note that inducing sign floating required an extraordinarily large $\lambda_{\mathrm{float}}$, such as $10^4$, suggesting that such strong attraction toward the sign boundary is hard to realize under standard training protocols.

# E. Additional Experiments

This appendix provides additional experiments and extended empirical results that complement the main text. Specifically:

- **Appendix E.1** summarizes additional empirical results on scale, ImageNet compression, AdamW update magnitudes, and CNN sign lock-in.

- **Appendix E.2** reports additional randomness tests and supplementary analyses.

- **Appendix E.3** provides additional vision-task experiments and sign lock-in validations.

- **Appendix E.4** documents the (approximately) zero-cost sign template method and the full bit-budget accounting for sub-bit compression.

## E.1. Additional Empirical Results

This appendix collects additional empirical results on scale, expressivity, optimizer assumptions, and the practical role of sign-template compression.

### E.1.1. TEMPLATE EXPRESSIVITY ACROSS MODEL SCALES

To test whether a fixed rank-2 sign template limits expressivity at larger scales, we trained Transformer language models from 0.3M to 512M parameters. Table 2 reports the best validation perplexity. The gap between vanilla training and gap+regularization decreases with model size and becomes negligible.

*Table 2.* **Best validation PPL on Transformer next-token prediction.**

| #Params | Vanilla | Gap+Reg | Degradation PPL |
|---------|---------|---------|-----------------|
| 0.3M    | 5.22    | 5.61    | +0.39           |
| 1.9M    | 4.55    | 4.69    | +0.14           |
| 4.8M    | 4.67    | 4.64    | -0.03           |
| 10.8M   | 4.67    | 4.63    | -0.04           |
| 25.4M   | 4.70    | 4.69    | -0.01           |
| 59.3M   | 4.82    | 4.78    | -0.04           |
| 99.5M   | 4.87    | 4.78    | -0.09           |
| 512M    | 4.93    | 4.94    | +0.01           |

Table 3 reports best-vs-final validation perplexity. At larger scales, vanilla training overfits strongly, while the low-rank sign-template training with gap+regularization substantially reduces this degradation.

*Table 3.* **Overfitting: best vs. final PPL on Transformer next-token prediction.**

| #Params | Vanilla Best | Vanilla Final | Vanilla Overfit | Gap+Reg Best | Gap+Reg Final | Gap+Reg Overfit |
|---------|--------------|---------------|-----------------|--------------|---------------|-----------------|
| 0.3M    | 5.22         | 5.22          | +0.00           | 5.61         | 5.56          | -0.05           |
| 1.9M    | 4.55         | 4.69          | +0.14           | 4.69         | 4.70          | +0.01           |
| 4.8M    | 4.67         | 5.96          | +1.29           | 4.64         | 4.67          | +0.03           |
| 10.8M   | 4.67         | 9.42          | +4.75           | 4.63         | 4.94          | +0.31           |
| 25.4M   | 4.70         | 12.58         | +7.88           | 4.69         | 5.76          | +1.07           |
| 59.3M   | 4.82         | 15.36         | +10.54          | 4.78         | 5.79          | +1.01           |
| 99.5M   | 4.87         | 18.09         | +13.22          | 4.78         | 5.79          | +1.01           |
| 512M    | 4.93         | 7.92          | +2.99           | 4.94         | 5.00          | +0.06           |

### E.1.2. BILLION-SCALE SIGN LOCK-IN EVIDENCE

We computed the effective sign-flip histogram on a 1B-parameter Transformer trained on 300B tokens. Table 4 shows that the empirical distribution is well captured by a zero-inflated geometric fit.

*Table 4.* **Pythia-1B trained on 300B tokens: effective flip-count histogram and geometric-tail fit.**

| $K_{\text{eff}}$ | Count | Empirical prob. | Fitted prob. | Abs. error |
|---|---|---|---|---|
| 0 | 168032 | 0.840160 | 0.840160 | 0.000000 |
| 1 | 25405 | 0.127025 | 0.126960 | 0.000065 |
| 2 | 5170 | 0.025850 | 0.026116 | 0.000266 |
| 3 | 1120 | 0.005600 | 0.005372 | 0.000228 |
| 4 | 230 | 0.001150 | 0.001105 | 0.000045 |
| 5 | 37 | 0.000185 | 0.000227 | 0.000042 |
| 6 | 5 | 0.000025 | 0.000047 | 0.000022 |
| 7 | 1 | 0.000005 | 0.000010 | 0.000005 |

### E.1.3. IMAGENET-SCALE END-TO-END COMPRESSION

To validate the compression consequence on a larger benchmark, we applied the sign-template method to ImageNet training with ResNet and evaluated SVD compression with NormalFloat factor quantization. Table 5 reports Top-5 accuracy. The sign-template method is substantially stronger near and below the one-bit regime.

*Table 5.* **ImageNet Top-5 accuracy (%) of ResNet under compression via SVD with NormalFloat factor quantization.**

| bpw | Sign Template (ours) | Vanilla SVD (baseline) | $\Delta$ |
|---|---|---|---|
| 0.50 | 5.3 | 0.5 | +4.8 |
| 0.75 | 23.1 | 0.5 | +22.6 |
| 1.00 | 42.0 | 0.5 | +41.5 |
| 1.50 | 65.6 | 12.1 | +53.5 |
| 2.00 | 74.7 | 65.1 | +9.6 |
| 2.50 | 78.1 | 76.3 | +1.8 |
| 3.00 | 79.8 | 80.9 | -1.1 |
| 4.00 | 81.3 | 83.8 | -2.5 |

### E.1.4. ADAMW UPDATE MAGNITUDES

All main training experiments used AdamW. To empirically check the high-probability bounded-update condition, we measured elementwise update magnitudes over training phases using outer threshold $\rho = 10^{-3}$. Table 6 shows that the maximum observed update remains far below $2\rho = 2 \times 10^{-3}$ in all phases, and no step exceeds this threshold.

### E.1.5. COMPONENT-WISE LOW-RANK ERROR

To clarify why direct low-rank approximation of $W$ is difficult, Table 7 reports component-wise rank approximation error at $r/d = 0.03125$. Across models, $E_r(W)$ is much closer to $E_r(\text{sign}(W))$ than to $E_r(|W|)$, supporting the claim that sign structure is the dominant obstacle.

### E.1.6. CNN EFFECTIVE FLIP HISTOGRAM

Finally, Table 8 reports the effective sign-flip histogram in a CNN experiment, showing that sign lock-in is also observed outside Transformer language models.

## E.2. Additional Randomness Test

### E.2.1. ALGORITHMIC COMPRESSIBILITY OF SIGN BITS VIA GENERAL-PURPOSE COMPRESSORS

In the sub-bit regime, the sign pattern becomes a first-class storage target: storing signs naively costs one bit per weight and can dominate the effective bit budget when magnitudes are aggressively compressed. While our earlier analyses examine randomness of sign matrices through spectral statistics, here we add a complementary check based on *algorithmic compressibility*. If the learned sign pattern contains exploitable regularities, then off-the-shelf lossless compressors should

*Table 6.* **Phase-wise empirical AdamW update magnitudes.**

| Phase | Statistic | Value |
|---|---|---|
| Early | median update | $9.999 \times 10^{-5}$ |
| Early | 90th-percentile update | $1.191 \times 10^{-4}$ |
| Early | 99th-percentile update | $1.540 \times 10^{-4}$ |
| Early | 99.9th-percentile update | $1.865 \times 10^{-4}$ |
| Early | maximum update | $2.346 \times 10^{-4}$ |
| Early | # steps with max update $> 2\rho$ | 0 |
| Intermediate | median update | $2.364 \times 10^{-5}$ |
| Intermediate | 90th-percentile update | $6.021 \times 10^{-5}$ |
| Intermediate | 99th-percentile update | $1.122 \times 10^{-4}$ |
| Intermediate | 99.9th-percentile update | $1.718 \times 10^{-4}$ |
| Intermediate | maximum update | $2.354 \times 10^{-4}$ |
| Intermediate | # steps with max update $> 2\rho$ | 0 |
| Late | median update | $1.452 \times 10^{-5}$ |
| Late | 90th-percentile update | $3.893 \times 10^{-5}$ |
| Late | 99th-percentile update | $6.902 \times 10^{-5}$ |
| Late | 99.9th-percentile update | $9.574 \times 10^{-5}$ |
| Late | maximum update | $1.943 \times 10^{-4}$ |
| Late | # steps with max update $> 2\rho$ | 0 |

*Table 7.* **Low-rank approximation error by component at $r/d = 0.03125$.**

| Model | $E_r(\mathrm{sign}(W))$ | $E_r(|W|)$ | $E_r(W)$ |
|---|---|---|---|
| MLP | 0.927 | 0.573 | 0.895 |
| ResNet | 0.943 | 0.593 | 0.921 |
| TinyLlama | 0.934 | 0.571 | 0.908 |

*Table 8.* **Effective sign-flip histogram in the CNN experiment.**

| Flip count | Observed count | Fitted count |
|---|---|---|
| 0 | 39161 | 39161 |
| 1 | 2059 | 1929 |
| 2 | 703 | 961 |
| 3 | 574 | 479 |
| 4 | 263 | 239 |
| 5 | 142 | 119 |
| 6 | 57 | 59 |
| 7 | 26 | 30 |
| 8 | 15 | 15 |
| 9 | 6 | 7 |

achieve nontrivial savings when applied directly to the sign bitstream.

**Protocol.** For each selected weight matrix, we extract the elementwise sign pattern $\mathrm{sign}(W)$ using the same convention as the rest of the paper. We bit-pack the binary signs into a byte stream and measure the *compression ratio* defined as compressed size divided by the raw packed size (smaller means more compressible). To avoid conclusions driven by a single coding scheme, we evaluate a diverse set of widely used lossless compressors. Concretely, we always include raw DEFLATE, zlib, gzip, bzip2, and LZMA; when available, we also report Zstandard, Brotli, LZ4, and Snappy. For the DEFLATE family, we report raw DEFLATE to reduce constant header/metadata effects (DEFLATE/zlib/gzip follow the

*Table 9.* Compressibility of sign bits across compression algorithms (ratio = compressed / raw).

| model | brotli(q11) | brotli(q5) | bz2(l9) | gzip(l9) | lzma(p6) | raw_deflate(l9) | snappy | zlib(l9) | zstd(l10) | zstd(l3) |
|---|---|---|---|---|---|---|---|---|---|---|
| Rademacher baseline (2048x2048) | 1.000 | 1.000 | 1.005 | 1.000 | 1.000 | 1.000 | 1.000 | 1.000 | 1.000 | 1.000 |
| 2D Ising baseline (beta=0.6) | 0.001 | 0.002 | 0.004 | 0.025 | 0.004 | 0.025 | 0.067 | 0.025 | 0.002 | 0.003 |
| Low-rank sign baseline (rank=2) | 0.047 | 0.054 | 0.062 | 0.184 | 0.043 | 0.184 | 0.417 | 0.184 | 0.053 | 0.083 |
| MLP-Mixer-B16 | 0.978 | 0.983 | 0.998 | 0.985 | 0.984 | 0.985 | 1.000 | 0.985 | 0.983 | 0.983 |
| ResNet18 | 0.957 | 0.958 | 0.984 | 0.964 | 0.966 | 0.964 | 1.000 | 0.964 | 0.960 | 0.959 |
| TinyLlama-1.1B-Chat | 0.979 | 0.984 | 0.999 | 0.989 | 0.973 | 0.989 | 1.000 | 0.989 | 0.991 | 0.992 |

corresponding RFC specifications) (Deutsch, 1996a; Deutsch & Gailly, 1996; Deutsch, 1996b). Brotli and Zstandard follow their RFC specifications (Alakuijala & Szabadka, 2016; Collet & Kucherawy, 2021). bzip2 is a widely used implementation based on the Burrows–Wheeler transform (Burrows & Wheeler, 1994; Seward, 2019). LZMA is based on public SDK documentation (Pavlov, 2026). LZ4 and Snappy follow the descriptions of their public reference implementations (Collet, 2011; Google, 2011).

**Baselines.** To calibrate the results, we apply the same procedure to three reference sources and include them in the same table: (i) an i.i.d. Rademacher sign stream as a maximally unstructured reference; (ii) a structured *two-dimensional Ising* sign field at a fixed inverse-temperature setting, which introduces short-range correlations (Ising, 1925; Onsager, 1944); and (iii) a rank-2 low-rank sign template constructed by taking the sign of a low-rank factor product, which provides an explicit low-complexity sign pattern. Because (ii) and (iii) contain deliberate structure, they are expected to be more compressible than the i.i.d. reference under generic codecs.

**Results.** Table 9 reports compression ratios for learned sign bitstreams across compressors. Across architectures, learned sign bits compress nearly as poorly as the i.i.d. Rademacher reference under multiple codecs, whereas the structured baselines (2D Ising and the rank-2 template) are consistently more compressible. This consistency across compression families supports the interpretation that the observed "random-like" behavior of learned sign patterns is not an artifact of a particular spectral statistic: from an algorithmic coding perspective, the learned sign bitstreams expose little redundancy that generic lossless compressors can exploit. Taken together with our spectral evidence, these results reinforce the picture that sign patterns can dominate bit-cost in the sub-bit regime while remaining difficult to compress by generic means.

### E.3. Additional Vision Task Experiments

#### E.3.1. EXPERIMENTAL SETUP

**Dataset.** We use MNIST in the standard Keras format, normalized by the usual mean and standard deviation (mean 0.1307, std 0.3081), and we add a single channel dimension. We form a deterministic train/validation split by shuffling the original 60k training examples with a fixed seed and taking 10k examples for validation (50k remain for training). The test set contains 10k examples.

**Model.** We train a simple MLP with two hidden layers: *Flatten* $\rightarrow$ Linear($784 \rightarrow 256$) $\rightarrow$ ReLU $\rightarrow$ Linear($256 \rightarrow 256$) $\rightarrow$ ReLU $\rightarrow$ Linear($256 \rightarrow 10$).

**Optimization.** We optimize the cross-entropy loss using AdamW with batch size 128 (training) and 512 (evaluation), we train for $T = 2000$ steps per run. We report validation error $\text{Err}_{\text{val}} = 1 - \text{Acc}_{\text{val}}$ and also track sign-related statistics described below.

**Initialization and Gap+reg method.** The baseline initialization samples each linear weight entry i.i.d. from $\mathcal{N}(0, \sigma_{\text{init}}^2)$ with $\sigma_{\text{init}} = 0.02$, and sets biases to zero. For the Gap+reg runs, we additionally apply a *near-zero rejection* (gap) rule to matrix parameters: we resample entries until $|w| \geq a$ (gap threshold $a$). We also add an *outer-drift (log-barrier) regularizer* with weight $\lambda$ as in Appendix C.4.

**Sign tracking and effective flips.** This setting is the same as Appendix C.4.

**Zero-inflated geometric model.** This setting is the same as Appendix C.4.

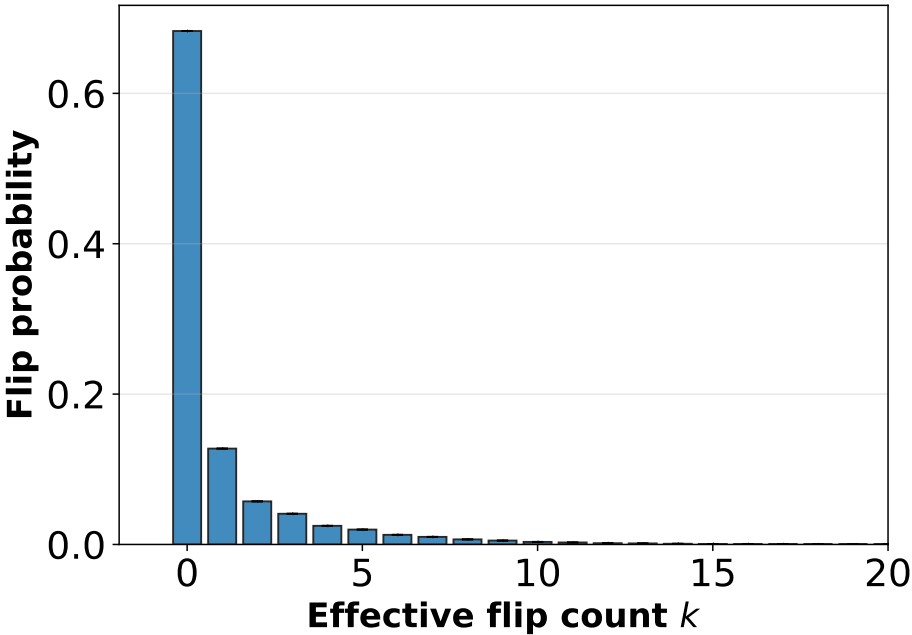

*Figure 17.* **Empirical distribution of the effective flip count $K$.** Most weights exhibit $k = 0$, with a rapidly decaying tail for $k \geq 1$.

**Mean flip rate.** This setting is the same as Appendix C.4.

### E.3.2. EFFECTIVE FLIP DISTRIBUTION AND GEOMETRIC TAIL

Figure 17 shows a representative histogram of the effective flip count $K$ under baseline training. A large fraction of parameters never experience an effective flip ($K = 0$), with rapidly diminishing probability mass for larger $K$. This heavy mass at zero is consistent with the sign lock-in picture: once a parameter commits to an outer-region sign, it rarely returns to the near-zero boundary band and re-emerges with the opposite sign.

To test the geometric-tail prediction, Figure 18 plots the tail probability $\mathbb{P}[K \geq k]$ on a log scale for multiple learning rates. Across learning rates, the tail is approximately linear in $k$ on the semi-log plot, indicating an exponential/geometric decay. The dashed lines show simple geometric fits of the form $\mathbb{P}[K \geq k] \approx h \, g^{k-1}$. We observe that increasing the learning rate increases both the fitted prefactor $h$ and the ratio $g$ (e.g., $h$ and $g$ are smallest for lr $= 10^{-4}$ and largest for lr $= 10^{-3}$), which corresponds to (i) more parameters reaching the boundary at least once and (ii) a heavier tail once flips occur. This matches the intuition that larger updates make boundary re-entry events more common.

### E.3.3. ESTIMATED LOCK-IN PARAMETERS OVER GAP-INIT AND REGULARIZATION ON VISION TASK.

We sweep the gap threshold and regularizer weight over

$$a \in \{0.001, 0.005, 0.02, 0.03, 0.05\}, \qquad \lambda \in \{10^{-4}, 10^{-3}, 10^{-2}, 0.05, 0.1, 0.3, 0.5\}.$$

We then fit $(\hat{h}, \hat{g})$ from the sampled weights and report mean±std over 3 seeds.

Figure 19 supports the interpretation that the two mechanisms act in complementary ways: *(i) Gap initialization primarily suppresses $\hat{h}$,* because moving initial weights away from the boundary reduces the probability that a parameter ever visits the near-zero band during training; indeed, at $\lambda \approx 0$, $\hat{h}$ decreases sharply as $a$ increases. *(ii) Outer-drift regularization strongly reduces $\hat{g}$,* because the log-barrier discourages re-entry into the boundary band once a parameter is in the outer region; correspondingly, $\hat{g}$ drops monotonically as $\lambda$ grows. At sufficiently large $\lambda$, both $\hat{h}$ and $\hat{g}$ become small, implying a much lighter tail and fewer total effective sign flips.

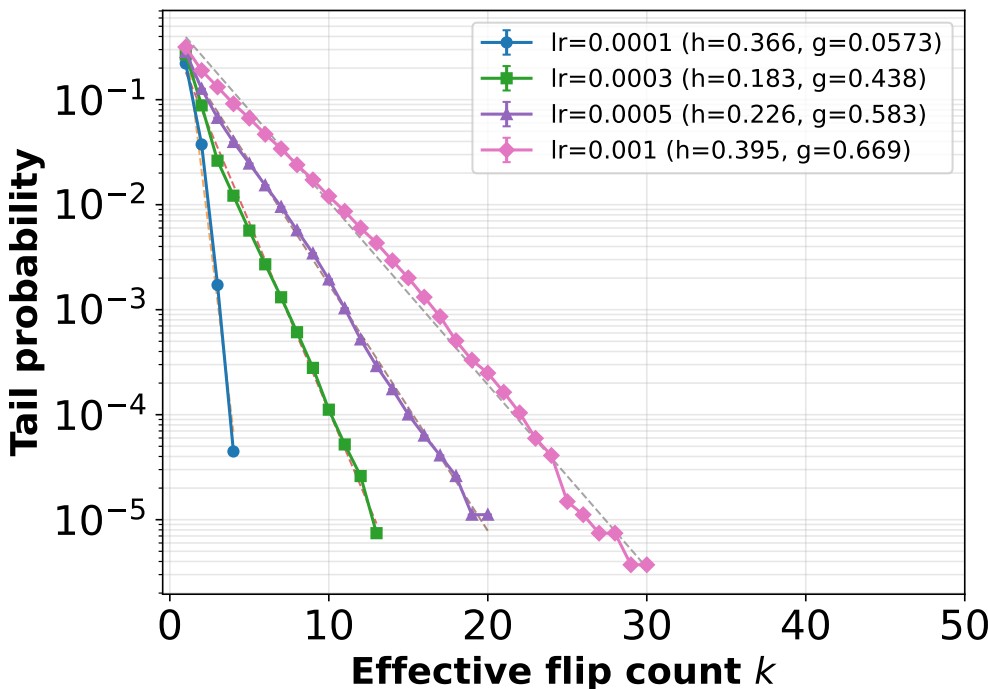

*Figure 18.* **Semi-log plot of tail probabilities** $\mathbb{P}[K \geq k]$ **for different learning rates.** Dashed lines show geometric fits $\mathbb{P}[K \geq k] \approx h\,g^{k-1}$, indicating an approximately geometric tail whose heaviness increases with learning rate.

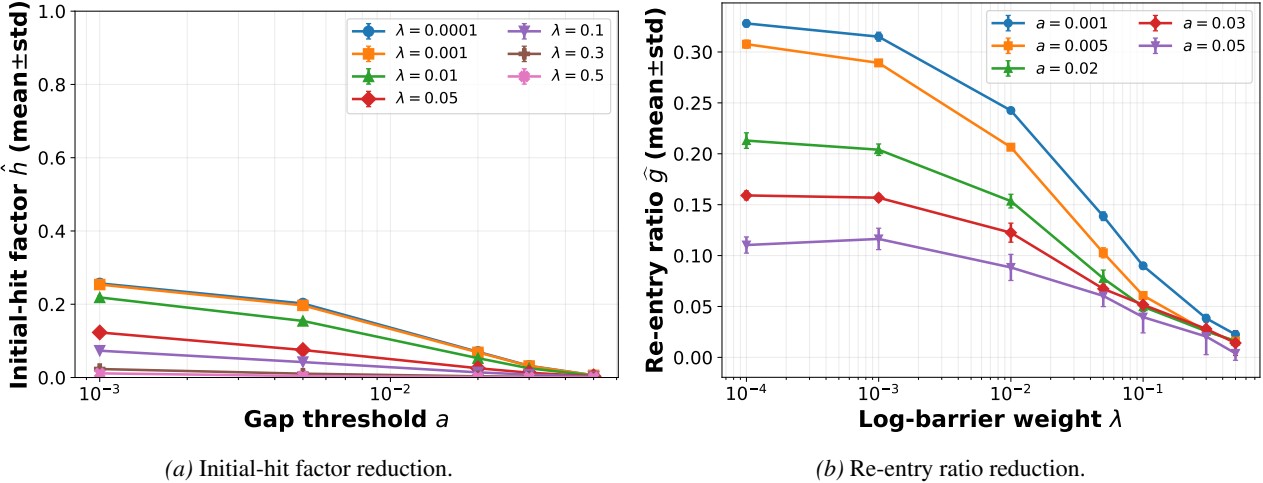

*(a)* Initial-hit factor reduction.
      *(b)* Re-entry ratio reduction.

*Figure 19.* **Estimated lock-in parameters over gap-init and regularization on vision task. Left:** $\hat{h}$ (initial-hit factor) as a function of the gap threshold $a_{\text{init}}$ for different log-barrier weights $\lambda$. **Right:** $\hat{g}$ (re-entry ratio) as a function of $\lambda$ (log scale) for different $a_{\text{init}}$. Points show mean±std over three seeds.

### E.3.4. TRADE-OFF BETWEEN MEAN FLIP RATE AND VALIDATION ERROR

Figure 20 plots validation error against the mean per-step flip rate (log-$x$) for the baseline model and for the Gap+reg sweep. Each curve corresponds to a fixed gap threshold $a$ and varying $\lambda$.

We observe a clear sign-stability/performance trade-off for some settings (notably small $a$): aggressive regularization can reduce flip rates substantially but may increase validation error, suggesting that overly constraining early dynamics can slow or misdirect optimization. However, the sweep also reveals favorable regimes (e.g., moderate/large $a$ with intermediate $\lambda$) where flip rates drop by orders of magnitude relative to the baseline while validation error remains comparable. Overall,

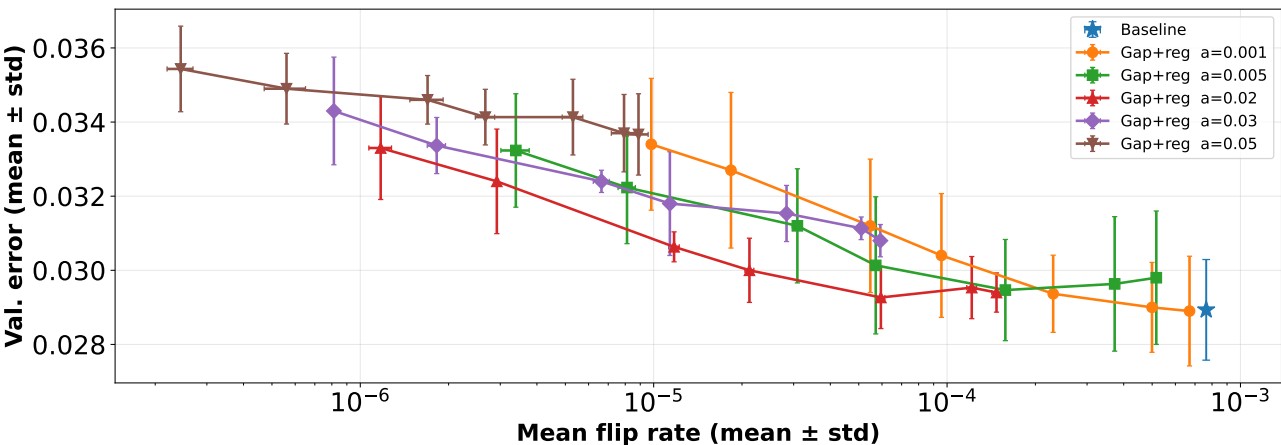

*Figure 20.* **Validation error vs. mean flip rate.** Baseline is shown as a single point, and Gap+reg curves connect runs with the same $a$ while varying $\lambda$. Error bars denote mean±std over seeds. The plot highlights both the trade-off regime (very low flip rate can hurt validation error for small $a$) and regimes where flip rates are drastically reduced with little or no degradation in validation error. Curves connect points with the same gap threshold $a_{\text{init}}$ and the log barrier weight decreases from left to right as $0.5, 0.3, 0.1, 0.05, 0.01, 0.001, 0.0001$.

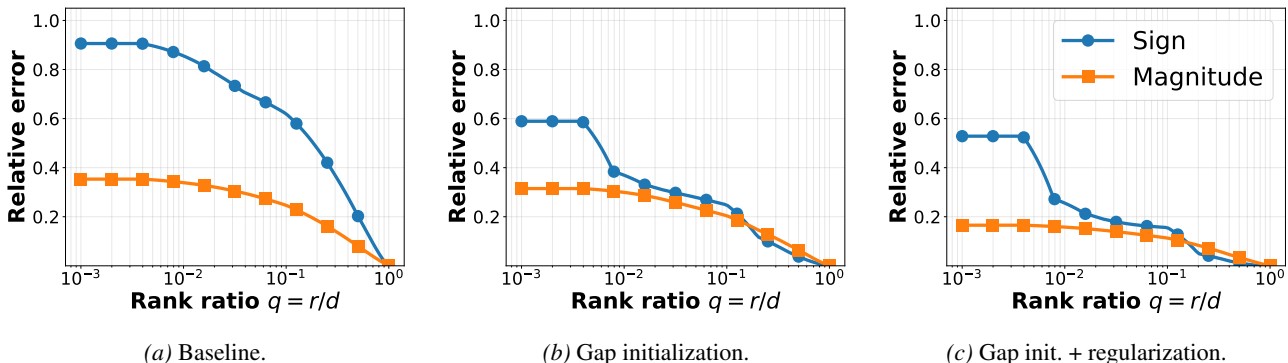

*(a)* Baseline.   *(b)* Gap initialization.   *(c)* Gap init. + regularization.

*Figure 21.* **Sign vs. magnitude low-rank compressibility.** Relative Frobenius error $E_r(M) = \|M - M_r\|_F / \|M\|_F$ as a function of rank ratio $q = r/d$ (log scale), for the sign matrix $S = \text{sign}(W)$ and magnitude matrix $A = |W|$, evaluated on the final trained weights under (a) baseline, (b) gap initialization only, and (c) gap initialization with outer-drift regularization.

these results indicate that sign stabilization can be tuned to reduce sign churn substantially without requiring a large accuracy penalty.

The resulting curves are shown in Figure 21a–21c. Consistent with the CharLM analysis in the main paper, the sign matrix is less compressible than the magnitude matrix in the baseline regime, while the sign becomes substantially more low-rank compressible under the enhanced lock-in settings. In particular, the regularized regime exhibits the strongest improvement for $\text{sign}(W)$ across the rank-ratio sweep, indicating that enforcing sign lock-in can mitigate the sign bottleneck in the low-rank compression view.

## E.4. Approximately Zero-Cost Sign Template Method for Sub-Bit Compression

We provide additional details on the template-based sub-bit compression method referenced in the main text. Let $W \in \mathbb{R}^{m \times n}$ be a weight matrix and decompose it as

$$W = S \odot A, \quad S \in \{\pm 1\}^{m \times n}, \quad A \in \mathbb{R}_{\geq 0}^{m \times n},$$

where $S = \text{sign}(W)$ and $A = |W|$. In a conventional representation, storing $S$ requires 1 bit per weight. In contrast, we restrict the sign matrix to a *known sign template* $\mathbf{T} \in \{\pm 1\}^{m \times n}$, which can be deterministically regenerated from minimal side information (e.g., a seed parameter). If we enforce $\text{sign}(W) = \mathbf{T}$ for the targeted matrices, the decoder can reconstruct $\mathbf{T}$ on the fly, and only the magnitudes $A$ need to be stored. As a result, the cost per-weight sign becomes approximately zero:

$$\text{bits}_{\text{sign}}(W) \approx 0. \tag{20}$$

With respect to the memory capacity of the sign matrix in the sign template method, the storage cost of the model becomes zero when the matrix is generated using an identical seed and the same random number generator. Consequently, the storage cost associated with the sign matrix of the model is fully attributable to the storage cost of the program, except the seed integer. When deploying on GPUs, the constraints differ from those in CPU-based settings. In particular, constructing the coding (sign) matrices on-the-fly during inference via a pseudorandom number generator is impractical. Instead, we generate a single global coding matrix from a fixed seed and obtain the coding matrix for each weight matrix by slicing the corresponding submatrix. Even in this scheme, $\text{bits}_{\text{sign}}(W)$ is less than $1/100$ because the single global coding matrix is reduced by the low-rank matrix factorization in Section E.4.1 and the reuse effect, which has a large contribution in LLM. In this Appendix, the bit cost of magnitude is about $1/2^4$ in lowest case and we ignore the cost of the sign template.

### E.4.1. SIGN TEMPLATE

For each target layer $l \in \mathcal{M}$ with weight shape $m \times n$, we use a re-generable sign template $\mathbf{T}^{(l)} \in \{\pm 1\}^{m \times n}$ and enforce the sign constraint.

**Low-rank sign template.** We generate low-rank real factors and then take the sign:

$$G \in \mathbb{R}^{m \times r}, \quad H \in \mathbb{R}^{n \times r}, \quad G_{ik} \overset{\text{i.i.d.}}{\sim} P_{\text{init}}, \quad H_{jk} \overset{\text{i.i.d.}}{\sim} P_{\text{init}}, \quad \mathbf{T}^{(l)} = \text{sign}(GH^\top), \tag{21}$$

where $r \ll \min(m, n)$ controls the intrinsic degrees of freedom.

### E.4.2. ADDITIONAL SIGN LOCK-IN ENHANCEMENT MECHANISM

In practice, we combine the template approach with the two interventions introduced in the main text, such as gap initialization and outer-drift regularization, to suppress early visits to the sign boundary.

**Template-aware gap initialization.** Denoting the near-zero-rejection sample matrix by $Z^{(l)}$, we initialize

$$W_0^{(l)} = \mathbf{T}^{(l)} \odot |Z^{(l)}|.$$

**Hard projection.** After each optimizer update, we perform an element-wise hard projection

$$W^{(l)} \leftarrow \Pi_{\text{hard}}^{(l)}(W^{(l)}), \quad \Pi_{\text{hard}}^{(l)}(W)_{ij} = \mathbf{T}_{ij}^{(l)} \cdot |W_{ij}|, \quad l \in \mathcal{M}, \tag{22}$$

which enforces $\text{sign}(W^{(l)}) = \mathbf{T}^{(l)}$ precisely while preserving magnitudes. This method eliminates remaining sign flips that could not be corrected by gap initialization and outer-drift regularization.

**Remark E.1** (Combining Gap init and outer-drift reduces hard-projection activations). *Proposition D.29 shows that under Gap initialization and outer-drift regularization, the boundary-hit times satisfy a geometric tail bound: $\mathbb{P}[\tau_k \leq T] \leq h_T^{\text{gap}}(g_T^{\text{OD}})^{k-1}$. In our implementation, the hard projection (Eq. (22)) is nontrivial only when the raw update crosses the sign boundary, i.e., when an entry attempts to move to the opposite sign side. On the good event $\mathcal{E}_\Delta$ from Assumption 3.3 (hence $|w_{t+1} - w_t| \leq \Delta$ up to time $T$) and the choice $\epsilon = \max\{\epsilon_0, \Delta\}$ (Definition 3.1), such a sign-crossing event can occur only after the trajectory approaches the boundary neighborhood $\{|w_t| \leq \epsilon\}$ closely (indeed, crossing 0 in one step forces $|w_t| \leq \Delta \leq \epsilon$). Therefore, reducing boundary visits (smaller $h_T^{\text{gap}}$) and suppressing re-entries (smaller $g_T^{\text{OD}}$) also reduces the frequency of nontrivial hard-projection activations. Empirically, we observe that the combination of these three components—Gap initialization, outer-drift regularization, and hard projection—yields substantially fewer projection activations than using hard projection alone.*

### E.4.3. MAGNITUDE QUANTIZATION USING SVD

After fixing signs, we only store $A^{(l)} = |W^{(l)}|$ for each $l \in \mathcal{M}$. We apply truncated SVD:

$$A^{(l)} \approx A_r^{(l)} := U_r^{(l)} \Sigma_r^{(l)} (V_r^{(l)})^\top,$$

and quantize the factors with a $b$-bit symmetric uniform quantizer

$$Q_b(x; \alpha) := \alpha \cdot \text{clip}\big(\lfloor x/\alpha \rceil, -2^{b-1}, 2^{b-1} - 1\big). \tag{23}$$

Define

$$\widehat{A}^{(l)} := Q_{b_U}\big(U_r^{(l)}; \alpha_U^{(l)}\big) \, Q_{b_\Sigma}\big(\Sigma_r^{(l)}; \alpha_\Sigma^{(l)}\big) \, Q_{b_V}\big(V_r^{(l)}; \alpha_V^{(l)}\big)^\top,$$

so the reconstructed weight is

$$\widehat{W}^{(l)} = T^{(l)} \odot \widehat{A}^{(l)}.$$

The amortized magnitude bit-cost per weight for an $m \times n$ matrix is

$$\text{bits}_{\text{amp}}(W^{(l)}) \approx \frac{b_U \, mr + b_V \, nr + b_\Sigma \, r}{mn}, \tag{24}$$

while the sign bit-cost remains approximately zero for $l \in \mathcal{M}$.

### E.4.4. EXPERIMENTAL VALIDATION

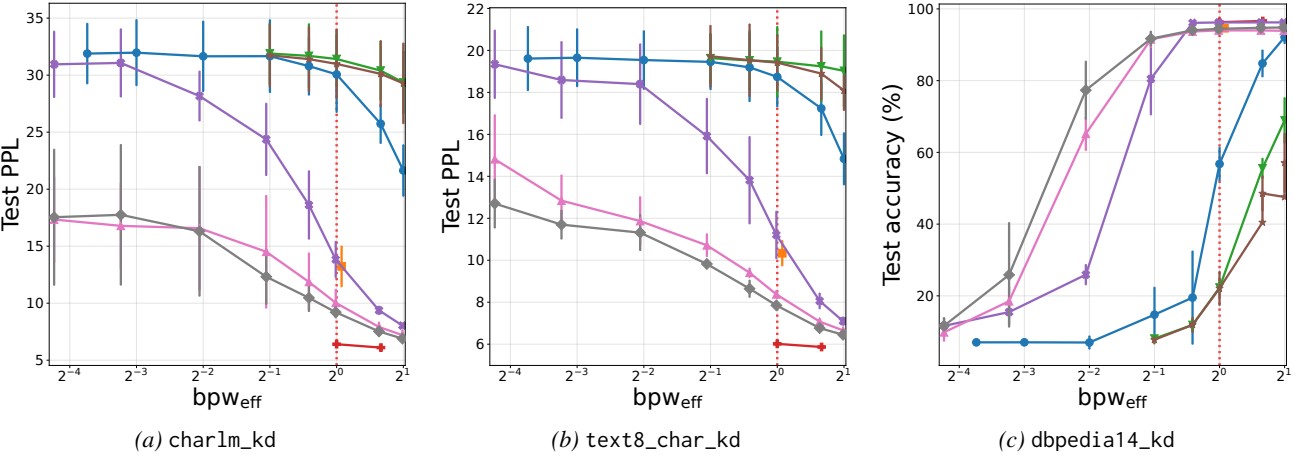

*(a)* `charlm_kd`      *(b)* `text8_char_kd`      *(c)* `dbpedia14_kd`

*Figure 22.* **Performance vs. effective bits per weight** ($\text{bpw}_{\text{eff}}$) **under knowledge distillation.** Markers indicate the mean over three seeds and error bars show one standard deviation. Lower is better for perplexity (CharLM-KD, Text8-Char-KD), while higher is better for accuracy (DBPedia14-KD). These panels complement the main benchmark results in Figure 8.

We next validate the practical effectiveness of the proposed *zero-cost sign template* approach. Recall that, when we constrain the sign of each targeted weight matrix to a deterministically re-generable template, the sign storage becomes zero in Eq. (20), and only the nonnegative magnitudes need to be stored and compressed. In our implementation, we compress magnitudes via truncated SVD and store quantized factors. We focus on the Transformer for language tasks due to emerging needs for the model compression. Figure 8 reports the main benchmark tasks, while Figure 22 reports additional knowledge-distillation variants. The details of this validation are reported in Section E.4.5.

**Protocol and metrics.** Figures 8 and 22 report task performance as a function of the *effective* bits-per-weight $\text{bpw}_{\text{eff}}$ on six benchmarks: CharLM and Text8-Char (test perplexity; lower is better), and DBPedia14 (test accuracy; higher is better), including their KD variants. Each marker indicates the mean over three random seeds and error bars show one standard deviation. All methods are compared at approximately matched $\text{bpw}_{\text{eff}}$ under the accounting rules in Appendix E.4.5.

**Main result: magnitude-only SVD with zero-cost signs is consistently strong in the sub-bit regime.** Across all benchmarks, our template-based method (SVD $|W_{\text{lockin}}|$) substantially improves over applying the same SVD budget directly to raw weights (SVD $W$ baseline) applied to the baseline weight matrix in the extreme low-bit region $\text{bpw}_{\text{eff}} < 1$. This behavior matches the empirical motivation in Section 1: sign patterns are random-like and difficult to compress, while magnitudes are more compressible and become the natural target once signs are made free. Concretely, on DBPedia14 at $\text{bpw}_{\text{eff}} \approx 0.24$, SVD $|W_{\text{lockin}}|$ achieves high accuracy while SVD $W$ baseline collapses; similarly, on Text8-Char and CharLM at the same budget, SVD $|W_{\text{lockin}}|$ yields clearly lower perplexity than SVD $W$ baseline. The gains persist on KD tasks (Figure 22), indicating that the advantage is not specific to a single training objective.

**Effect of simple preconditioning (naive vs. z-score).** We report two variants of magnitude SVD: (i) a naive pipeline that factorizes $|W|$ as-is, and (ii) a lightly preconditioned variant that normalizes magnitudes before SVD (z-score). The detail of this method is described in Appendix E.4.6. The z-score variant consistently dominates the naive variant across budgets and tasks, suggesting that even simple normalization improves the stability of low-rank factor quantization at very small $\text{bpw}_{\text{eff}}$.

**Comparison with existing extreme-compression baselines.** We additionally include representative existing approaches: HashedNets (weight sharing), OneBit, and unstructured pruning baselines (magnitude pruning and WANDA). Overall, the template-based magnitude SVD is the most reliable performer in the *sub-bit* region. Notably, when pruning is evaluated with a realistic sparse-storage model (CSR; Appendix E.4.5), the indexing overhead dominates at very small densities, leading to weak performance at matched $\text{bpw}_{\text{eff}}$. HashedNets improves as the budget increases but remains unreliable at the smallest budgets.

### E.4.5. EXPERIMENTAL DETAILS OF ZERO-COST SIGN TEMPLATE METHOD

This appendix describes the experimental setup, bit-budget accounting, and baseline implementations used to produce the results in Figures 8 and 22.

**Benchmarks.** We evaluate on six tasks:

- **CharLM / Text8-Char**: character-level language modeling; we report *test perplexity* (PPL).

- **DBPedia14**: 14-way text classification; we report *test accuracy*.

- **KD variants** (_kd): student models trained with knowledge distillation (KD) using a teacher trained on the corresponding base task.

Across all tasks, we report mean±std over three seeds (0/1/2).

**Models and Targeted Weights.** Our zero-template method is applied to a fixed set of targeted weight matrices (linear layers) in each model; all other parameters are maintained in full precision. In the experiments presented in Figures 8 and 22, the number of targeted parameter tensors is as follows:

- CharLM / Text8-Char: 14 targeted tensors.

- DBPedia14: 28 targeted tensors.

**Sign template.** For each targeted layer $l$ with shape $m \times n$, we define a re-generable sign template $\mathbf{T}^{(l)} \in \{\pm 1\}^{m \times n}$. Unless stated otherwise, we use $P_{\text{init}} = \text{Unif}[-1, 1]$ to sample the i.i.d. entries of the low-rank factors $(G, H)$ in Eq. (21) with a fixed global seed.

**Gap initialization and regularization.** We use a simple gap initialization that avoids near-zero magnitudes by enforcing an absolute minimum magnitude at initialization. In the reported runs, the gap threshold is chosen from

$$a_{\text{init}} = \{0.0, 0.01, 0.02, 0.03\}$$

and the outer-drift (log-barrier) regularization weight is set from

$$\lambda = \{0.0, 10^{-4}, 3.0 \times 10^{-4}, 10^{-3}\}$$

within 0.02 PPL drop or 2 % accuracy drop. This small performance degradation is consistent across tasks, so we could choose parameters other than $a_{\mathrm{init}} = 0$ and $\lambda = 0$ for two proposed methods. The two proposed methods are evaluated using the resulting sign-fixed weights. For the baseline, we use the resulting weight of $a_{\mathrm{init}} = 0$ and $\lambda = 0$.

**Hard projection.** During template-constrained training, after each optimizer update we apply the hard projection (Eq. (22)) to enforce $\mathrm{sign}(W^{(l)}) = \mathbf{T}^{(l)}$ exactly for all targeted layers. This guarantees that the sign component incurs zero storage cost at compression time.

### E.4.6. Preconditioning Variants for Magnitude SVD.

For SVD $|W_{\mathrm{lockin}}|$-naive, we apply truncated SVD directly to the magnitude matrix $A := |W_{\mathrm{lockin}}| \in \mathbb{R}_{\geq 0}^{m \times n}$. For SVD $|W_{\mathrm{lockin}}|$-zscore, we normalize $A$ column-wise before SVD and invert the normalization after reconstruction. Let $A := |W_{\mathrm{lockin}}| \in \mathbb{R}_{\geq 0}^{m \times n}$. Define the per-column mean and (population) standard deviation by

$$\zeta_j := \frac{1}{m} \sum_{i=1}^{m} A_{ij}, \quad \omega_j := \sqrt{\frac{1}{m} \sum_{i=1}^{m} (A_{ij} - \zeta_j)^2}, \quad j = 1, \ldots, n.$$

Let $\zeta := (\zeta_1, \ldots, \zeta_n)^\top \in \mathbb{R}^n$, $\omega := (\omega_1, \ldots, \omega_n)^\top \in \mathbb{R}_{\geq 0}^n$, and let $\mathbf{1}_m \in \mathbb{R}^m$ and $\mathbf{1}_n \in \mathbb{R}^n$ denote all-ones vectors. Introduce a small numerical stabilizer $\epsilon_{\mathrm{zs}} > 0$ and define

$$D_{\mathrm{zs}} := \mathrm{diag}(\omega + \epsilon_{\mathrm{zs}} \mathbf{1}_n) \in \mathbb{R}^{n \times n}.$$

The column-wise z-score normalized matrix is

$$A^{\mathrm{zs}} := (A - \mathbf{1}_m \zeta^\top) D_{\mathrm{zs}}^{-1} \in \mathbb{R}^{m \times n}.$$

We compute a rank-$r$ truncated SVD

$$A^{\mathrm{zs}} \approx U_r \Sigma_r V_r^\top,$$

fold the singular values into the left factor $\widetilde{U}_r := U_r \Sigma_r$, apply the main-text symmetric uniform $b$-bit quantizer $Q_b(\cdot)$ to the factors, and reconstruct

$$A_b^{\mathrm{zs}} := Q_b(\widetilde{U}_r) Q_b(V_r^\top).$$

Finally, we invert the normalization and enforce nonnegativity (element-wise):

$$A_b := A_b^{\mathrm{zs}} D_{\mathrm{zs}} + \mathbf{1}_m \zeta^\top, \quad A_b \leftarrow \max(A_b, 0).$$

We then form the sign-fixed weight using the fixed sign template $\mathbf{T}$:

$$W_b := T \odot A_b.$$

**Bit-budget Grid.** We evaluate a sweep of target effective budgets

$$\mathrm{bpw}_{\mathrm{target}} \in \{0.075, 0.125, 0.25, 0.5, 0.75, 1.0, 1.58, 2.0\}.$$

Figures 8 and 22 visualize the range up to 2.0 bpw to emphasize the sub-bit regime.

**Bit accounting of SVD-based storage for dense matrices (ours and SVDW baseline).** Given a target matrix $M \in \mathbb{R}^{m \times n}$ and rank $r$, we store quantized SVD factors ($U_r \in \mathbb{R}^{m \times r}$, $\Sigma_r \in \mathbb{R}^{r \times r}$ as a diagonal vector, and $V_r \in \mathbb{R}^{n \times r}$) using symmetric uniform $b$-bit quantizers (Eq. (23)). Following Eq. (24), the amortized bit cost per original weight is approximated by

$$\mathrm{bpw}_{\mathrm{eff}} \approx \frac{b \cdot (mr + nr + r)}{mn},$$

with $b = 4$ in our experiments (i.e., 4-bit quantization for the stored factors). For the **zero-template** method, sign bits are not stored (Eq. (20)); for **SVDW baseline**, the same SVD method is applied directly to $W$.

**Bit accounting of CSR storage for unstructured sparsity (pruning and WANDA).** For pruning-based methods, we assume CSR storage for each pruned matrix:

$$\mathrm{bpw_{eff}} \approx \frac{\underbrace{\mathrm{nnz} \cdot b_{\mathrm{val}}}_{\text{stored values}} + \underbrace{\mathrm{nnz} \cdot b_{\mathrm{idx}} + (m+1) \cdot b_{\mathrm{ptr}}}_{\text{indices/row pointers}}}{mn},$$

where $\mathrm{nnz}$ is the number of nonzeros, $b_{\mathrm{val}} = 4$ bits for quantized nonzero values, and we use 32-bit indices/pointers ($b_{\mathrm{idx}} = b_{\mathrm{ptr}} = 32$). This accounting explains why pruning can be unfavorable at extreme densities: index overhead dominates. We also impose a practical start threshold and only run pruning methods when the implied keep ratio exceeds

$$\mathtt{prune\_start\_keep\_frac} = 0.005.$$

**Baselines.** We summarize the baselines shown in Figures 8 and 22. All results shown in these figures are computed using three seeds (0, 1, 2) and reported as mean±std. We log per-method $\mathrm{bpw_{eff}}$ along with its decomposition (e.g., sign/index overhead vs. value bits) to ensure bit-budget correctness for each storage model.

- **HashedNets**: weight sharing through hashing into a budget-dependent number of buckets; bucket values are stored with the same value-bitwidth used elsewhere (4-bit in our implementation). No task-specific fine-tuning is employed.

- **OneBit**: explicit 1-bit sign storage with a small overhead for scale information.

- **Pruning**: magnitude-based unstructured pruning with CSR accounting; prune-only (no recovery fine-tuning applied).

- **WANDA**: activation-aware pruning utilizing the standard WANDA score; prune-only; CSR accounting.

- **QAT (reference)**: we include 1-bit and ternary QAT as reference points. In our code, QAT employs a STE with a short fine-tuning schedule applied to the targeted tensors; KD tasks utilize the same teacher as the KD student training.

