# OpenReview forum: "Sign Lock-In: Randomly Initialized Weight Signs Persist and Bottleneck Sub-Bit Model Compression"
_ICML.cc/2026/Conference — ICML 2026 regular_

### Official Review · Reviewer_VBmQ · 2026-02-23

**Soundness:** 2
**Presentation:** 3
**Significance:** 2
**Originality:** 3
**Overall Recommendation:** 4
**Confidence:** 3

**Summary:**

The authors analyze the structure of the learnt signs of various model architectures statistically during training. Their aim is to study the compressibility of the signs in sub 1 bit regimes, where they claim, signs are the bottleneck for compression (via quantization).
To this end, they first provide empirical analysis suggesting the similarity of learnt signs with Rademacher noise and that signs do not significantly change during training. Following which they propose a theoretical framework to characterize sign flipping during training and show that it follows a geometric tail structure. The authors then propose a regularizer to discourage sign changes during training, which helps maintain sign structure during training and thus keeps them compressible (low rank). The authors provide experiments on a 12 layer transformer model to validate their assumptions.

**Compliance With Llm Reviewing Policy:**

Affirmed.

**Final Justification:**

The rebuttal provided by the authors makes the premise of the problem much more clear. In my preliminary review, the importance of focusing on signs was not very obvious, the additional experiments provided help to address this, hence, I am increasing my score.

**Key Questions For Authors:**

I have added my question in the points above.

**Limitations:**

While the authors propose an interesting approach to the 1 bit wall for sign compression, the motivation for requiring separate sign compression is unclear. Moreover, some of the arguments presented in the paper need additional clarifications.

**Strengths And Weaknesses:**

Strengths:
1. The proposed 1 bit wall for compressing signs is an interesting bottleneck identified by the authors and suggests looking at parameter structures more closely for extreme compression regimes.
2. The authors provide a theoretical framework to closely study the different sign changes during training.

Weaknesses and Questions:

While the problem the authors are trying to tackle is a relevant and timely one (gaining insights for sub 1 bit compression), I have several questions about the structure and approach of this paper,

1. Context: The authors setup the problem by claiming that sign compression poses a bottleneck below 1 bits, while magnitudes can be easily compressed. I believe this claim requires more context. For example, why can we not consider a low rank approximation of the entire weight matrix, which can then be compressed (quantized subsequently) without needing to separate the signs and magnitudes?

2. Few Sign Flips: The authors suggest that very few sign flips happen during training. However, based on [1], this does not seem to be true for CNNs where almost half the initial signs are different from the learnt ones. Is the claim only related to sign flips which start from outside the boundary and enter the boundary as defined by the authors, if yes then this needs to be clarified.

3. Assumption 3.3 and 3.4: The assumptions consider only the sign flips that happen from outside the boundary to within the boundary but ignore the sign flips of the small parameters in the boundary (which are likely to happen more frequently). Similarly, the assumption ignores the outer -> outer sign flips, which maybe rare within one step but given momentum could happen in the span of a few steps. Given this, the assumptions are likely a bit strong and do not consider the full picture (to my understanding).

4. Sign Structure: A key concern for me is that the authors claim that the signs do not flip significantly during training and hence keeping them stable makes the signs more compressible. On the other hand the authors also claim that the signs follow a Rademacher noise distribution throughout training. Then, why is it the case that the initial signs are more compressible than the learnt ones, if both follow similar distributions. And why is forcing the signs to be fixed helping compressibility?

5. Performance tradeoff: From Fig 5 and the last figure in the appendix, there is a clear tradeoff for preventing sign flips via the proposed gap initialization and loss function, which significantly worsens performance. In this case, is achieving the low rank structure for better compressibility worth it, given the loss of performance? I assume, that the performance will be worsened further in actual compressions scenarios, which the authors have not discussed. And this trade off will only get worse for larger models (>1B), where compression is more relevant.

6. The presented experiments are only on the relatively small Shakespeare dataset, it would be interesting to see if this behaviour holds for larger datasets/models consistently.

7. Minor: Fig 6 is missing.

[1] Gadhikar, Advait, Tom Jacobs, Chao Zhou, and Rebekka Burkholz. "Sign-In to the Lottery: Reparameterizing Sparse Training From Scratch." NeurIPS 2025.

---

> ### Author Rebuttal · Authors · 2026-03-30
>
> ## General comment
> Thank you for the review. Let us clarify the key points. First, the weight matrix $W$ is not naively amenable to low-rank approximation. While this fact is known, **our main contribution is the discovery that the root cause lies in the sign component, not the magnitude.** This is because sign flips are rare and randomly initialized signs persist throughout training, and random sign patterns represent a worst case for compression. Furthermore, **we turn this sign persistence property to our advantage: by initializing with a compression-friendly sign pattern and preserving it through training**, we propose a method that removes this obstacle (1-bit wall).
>
> ## W1 (Why not low-rank the whole weight matrix directly?).
>
> The main obstacle in this direct SVD is the sign structure itself. To make it clear, the table below extends Fig. 2(a) by additionally reporting $E_r(W)$ together with $E_r(\mathrm{sign}(W))$ and $E_r(|W|)$. Across all models, **$E_r(W)$ is much closer to $E_r(\mathrm{sign}(W))$ than to $E_r(|W|)$. This shows that the low-rank incompressibility of the full weight matrix is primarily driven by the sign component rather than the magnitude.** Thus, our contribution is to identify the true bottleneck behind direct low-rank compression in the sub-bit regime. Once our sign-template method makes the sign structure low-rank-friendly, it can benefit both decomposed pipelines and direct SVD-based compression of $W$.
>
> ***Table (VBmQ-1). Low-rank approximation error by component (Fig. 2(a) at $r/d = 0.03125$).***
>
> | Model | $E_r(\mathrm{sign}(W))$ | $E_r(\lvert W \rvert)$ | $E_r(W)$ |
> |:---|---:|---:|---:|
> | MLP | 0.927 | 0.573 | 0.895 |
> | ResNet | 0.943 | 0.593 | 0.921 |
> | TinyLlama | 0.934 | 0.571 | 0.908 |
>
> ## W2 (Sign flips in CNN).
>
> Thank you for pointing us to this important reference. The Sign-In paper studies a specific sparse-training setting where inducing additional sign flips is important, rather than providing a general counterexample to sign lock-in in CNNs. In fact, as we show below, **sign lock-in is also observed in our CNN experiments.** In this sense, the two perspectives are broadly aligned: under natural SGD-based training, the sign lock-in phenomenon emerges, so explicit intervention is needed to induce more sign flips.
>
> ***Table (VBmQ-2). Effective sign-flip histogram in the CNN experiment.***
>
> | Flip count | Observed count | Fitted count |
> |---:|---:|---:|
> | 0 | 39161 | 39161 |
> | 1 | 2059 | 1929 |
> | 2 | 703 | 961 |
> | 3 | 574 | 479 |
> | 4 | 263 | 239 |
> | 5 | 142 | 119 |
> | 6 | 57 | 59 |
> | 7 | 26 | 30 |
> | 8 | 15 | 15 |
> | 9 | 6 | 7 |
>
>
> ## W3 (Why focus on effective flips?).
>
> The role of $\rho$ is to exclude such oscillations in the theory; empirically, $\rho$ is small and less than 1 \% flips are excluded. About multi-step flips, we do not ignore multi-step outer-to-outer flips. on the other hand, one-step outer-to-outer jump requires an update of $2\rho$. As shown in **Table (khej-1) in our response to Reviewer khej**, the observed maximum update never exceeds it.
>
> ## W4 (If signs stay Rademacher-like, why does fixing them help?).
>
> We believe this concern comes from conflating two different cases. If the sign matrix is essentially random, then preserving it does not help compression very much, because a random sign pattern is itself not low-rank-friendly. For example, consider
>
> $$S_{\\mathrm{rand}}=\\begin{bmatrix}1 & 1 & -1 & 1\\\\-1 & 1 & 1 & -1\\\\1 & -1 & 1 & -1\\\\-1 & -1 & 1 & 1\\end{bmatrix}$$
>
> which is difficult to approximate well by a very low-rank matrix.
> By contrast, if training preserves a deliberately structured sign template, then the situation is different. For example, consider
>
> $$S_{\\mathrm{temp}}=\\begin{bmatrix}1 & 1 & -1 & -1\\\\1 & 1 & -1 & -1\\\\-1 & -1 & 1 & 1\\\\-1 & -1 & 1 & 1\\end{bmatrix}=\\begin{bmatrix}1\\\\1\\\\-1\\\\-1\\end{bmatrix}\\begin{bmatrix}1 & 1 & -1 & -1\\end{bmatrix}$$
>
> which is rank $1$. In this case, preserving the sign pattern is highly beneficial, because the sign component is already compression-friendly. We do not claim that preserving an arbitrary Rademacher-like sign pattern will help. **Rather, our claim is that once sign lock-in is used to preserve a structured low-rank-friendly template, sign persistence becomes an advantage rather than a bottleneck.**
>
> ## W5&W6 (Scale of experiments).
>
> As the experimental scale increases, our claims become more pronounced. **Table (zyUe-1) in our response to Reviewer zyUe** confirms that sign lock-in persists at the 1B-parameter scale, **Table (hAUL-1) in our response to Reviewer hAUL** demonstrates that the expressivity overhead of the sign template becomes negligible at scale, and **Table (zyUe-2) in our response to Reviewer zyUe** shows that the compression gain breaks through the 1-bit wall on ImageNet.
>
> ## W7 (Minor).
>
> "Fig. 7" is intended to be Fig. 6.
>
> We hope these responses address your concerns and kindly ask you to reconsider your score.

---

> > ### Author Rebuttal · Reviewer_VBmQ · 2026-04-03
> >
> > Thank you for the additional experiments, and for highlighting the importance of signs in the low-rank approximation. This bottleneck is much more clear to me now. Additionally, the idea of fixing a more amenable (low rank structure) rather than a Rademacher one should probably be bettter explained in the paper.
> >
> > With this, my main concerns are addressed, I would increase my score.

---

> > > ### Author Response · Authors · 2026-04-03
> > >
> > > ## Reply to Reviewer VBmQ
> > >
> > > Thank you for the constructive review. We promise to improve it in the camera-ready version.
> > >
> > > ## Final Remark
> > > Dear Area Chair and all Reviewers
> > >
> > > Please allow us to make one final remark here. We believe that we have resolved all of the reviewers' major concerns. Through this discussion, our paper has become stronger and more valuable. Thank you to all reviewers for their dedicated contributions. We will further polish the paper for the camera-ready version.
> > >
> > > We would like to close by emphasizing the main message of our paper: randomly initialized weight signs persist during training and become the bottleneck in sub-bit compression, whereas a compression-friendly weight sign initialization can turn this property to an advantage. We believe this is a meaningful contribution because it reveals what fundamentally limits performance in the sub-bit regime and offers a practical example of how to remove that limitation.
> > >
> > > We are encouraged that the reviewers appreciated both the discovery of the sign lock-in phenomenon and its theoretical grounding, which together form the core contribution of this work. Although the current sign-template method is designed for training from scratch, we believe the underlying insight extends more broadly. If the sign structure is indeed the primary obstacle, then this insight can guide future work on more lightweight methods and the design of pretrained models that are compression-friendly from the outset. We believe this perspective can be broadly valuable to both the model compression community and researchers training deep learning models.
> > >
> > > Thank you very much for your consideration.
> > >
> > > Best regards
> > > The Authors

---

### Official Review · Reviewer_khej · 2026-03-13

**Soundness:** 3
**Presentation:** 3
**Significance:** 3
**Originality:** 3
**Overall Recommendation:** 4
**Confidence:** 3

**Summary:**

This paper investigates the challenge of sub-bit model compression, where the storage cost per weight falls below one bit. The key bottleneck identified is the incompressibility of weight sign patterns, which resemble i.i.d. noise and resist low-rank compression. The authors propose a "sign lock-in" theory to explain the persistence of initialization signs during training, attributing it to rare boundary crossings under SGD noise. They introduce two practical interventions—gap initialization and outer-drift regularization—to suppress early sign flips and stabilize sign structures. The work bridges theoretical analysis with empirical validation across diverse architectures (Transformers, CNNs, MLPs) and demonstrates that controlling sign patterns is critical for surpassing the "one-bit wall" in sub-bit compression.

**Compliance With Llm Reviewing Policy:**

Affirmed.

**Key Questions For Authors:**

1. How do the proposed "gap initialization" and "outer-drift regularization" differ in design and effectiveness from prior low-bitwidth training methods like DoReFa-Net or Trained Ternary Quantization, which also aim to stabilize sign structures? Could the authors provide a quantitative comparison (e.g., sign stability metrics, final model accuracy) to highlight the unique advantages of their approach?
2. The paper’s theoretical analysis relies on assumptions of "bounded updates" and "rare re-entry." How do these assumptions hold when using adaptive optimizers like Adam, which often exhibit unbounded or dynamic update magnitudes? Have the authors tested their methods under such optimizers, and if so, what adjustments were required to maintain the validity of the sign lock-in theory?

**Limitations:**

yes

**Strengths And Weaknesses:**

Strengths:
1. The paper is well-structured, with clear sections on problem formulation, theory, methods, and experiments.
2. The paper provides a rigorous theoretical framework (sign lock-in theory) to explain sign persistence, supported by stopping-time analysis and empirical validation.
3. Experiments are well-designed, covering multiple model architectures (Transformers, CNNs, MLPs) and compression regimes. The proposed methods (gap initialization, outer-drift regularization) are evaluated against strong baselines, showing consistent improvements in sign compressibility.
Weaknesses:
1. The idea of stabilizing sign structures through initialization and regularization is conceptually similar to prior work on training low-bitwidth networks (e.g., DoReFa-Net, Trained Ternary Quantization). The paper should explicitly compare its contributions to these works.
2. The theoretical analysis assumes bounded updates and rare re-entry, which may not hold in all training scenarios (e.g., with adaptive optimizers like Adam). The paper could benefit from discussing these limitations.

---

> ### Author Rebuttal · Authors · 2026-03-30
>
> ## General comment
> Thank you for the positive assessment and for raising two important clarification points.
>
> ## W4&Q1 (Relation to DoReFa-Net / TTQ-style low-bit training).
>
> We agree that this distinction should be made much more explicit. Our goal is different from classic binary / ternary QAT. Methods such as DoReFa-Net and TTQ constrain learned weights/activations to low-bit sets, but they still store the resulting learned sign / ternary states explicitly (typically together with auxiliary scales). Our target is the near- or sub-1-bit average-storage regime, where even one explicit sign bit per weight becomes the bottleneck. The paper's claim is therefore not "another binary training recipe," but rather: (i) learned signs are the bottleneck because they remain random-like and hard to compress; (ii) this happens because signs are largely locked to initialization; and (iii) if we choose a compressible sign prior and preserve it, the saved bit budget can be reallocated to magnitudes. Our findings can also be added on top of such binary QAT methods and may provide ideas for further improving their performance. In particular, since sign lock-in may also arise in these binary methods, it could enable the design of more effective optimization algorithms. In this sense, our approach is not opposed to existing binary QAT methods, but rather complementary to them, with the potential to boost their performance.
>
> ## W5&Q2 (Assumptions on Adam-like optimizers).
>
> **Our experiments were conducted using AdamW, as described in Appendix C.** The main theorem is optimizer-agnostic: Theorem 3.7 only needs Assumptions 3.3–3.4, while Proposition 3.5 is presented as an SGD-style sufficient condition. As noted in Remark 3.6, the sign lock-in theorem relies solely on these two assumptions, so any optimizer that satisfies comparable bounded-update and re-entry conditions arrives at the same conclusion. Moreover, for Adam-like methods with a preconditioner, the same conclusion can be reached by developing an analogous argument. Specifically, Assumption 3.3 can be verified by controlling the preconditioned momentum updates (as detailed in Remark D.9), and by deriving the counterpart of Proposition 3.5, one can directly prove Assumption 3.4 and promote it to a proposition. Therefore, Adam-like optimizers fall fully within the scope of the sign lock-in theory, just as SGD does.
>
> Importantly, Assumption 3.3 is formulated as a high-probability bounded-update condition with failure probability $\delta_{\mathrm{upd}}$, not as an almost-sure bound. The analysis is thus designed to accommodate occasional atypical updates. **Table (khej-1)** provides direct empirical verification: across all training phases, the maximum observed elementwise update ($2.354 \times 10^{-4}$) remains well below $2\rho = 2 \times 10^{-3}$, and the number of steps exceeding $2\rho$ is zero in every phase. This confirms that the bounded-update assumption is not merely a theoretical convenience but an empirically verified property of AdamW training, with no events that need to be absorbed into $\delta_{\mathrm{upd}}$.
>
> ***Table (khej-1). Phase-wise empirical AdamW update magnitudes.*** *We use the outer threshold $\rho = 10^{-3}$ (Appendix C.4.2 and C.5), split training into early (steps $0$–$199$), intermediate ($200$–$999$), and late ($1000$–$1999$) phases, and report the max-over-steps of each statistic. The last row counts steps whose elementwise max update exceeds $2\rho = 2\times 10^{-3}$.*
>
> | Phase | Statistic | Value |
> |:---|:---|---:|
> | Early | max over steps of median update | $9.999 \times 10^{-5}$ |
> | Early | max over steps of 90th-percentile update | $1.191 \times 10^{-4}$ |
> | Early | max over steps of 99th-percentile update | $1.540 \times 10^{-4}$ |
> | Early | max over steps of 99.9th-percentile update | $1.865 \times 10^{-4}$ |
> | Early | max over steps of maximum update | $2.346 \times 10^{-4}$ |
> | Early | # steps with max update $> 2\rho$ | $0$ |
> | Intermediate | max over steps of median update | $2.364 \times 10^{-5}$ |
> | Intermediate | max over steps of 90th-percentile update | $6.021 \times 10^{-5}$ |
> | Intermediate | max over steps of 99th-percentile update | $1.122 \times 10^{-4}$ |
> | Intermediate | max over steps of 99.9th-percentile update | $1.718 \times 10^{-4}$ |
> | Intermediate | max over steps of maximum update | $2.354 \times 10^{-4}$ |
> | Intermediate | # steps with max update $> 2\rho$ | $0$ |
> | Late | max over steps of median update | $1.452 \times 10^{-5}$ |
> | Late | max over steps of 90th-percentile update | $3.893 \times 10^{-5}$ |
> | Late | max over steps of 99th-percentile update | $6.902 \times 10^{-5}$ |
> | Late | max over steps of 99.9th-percentile update | $9.574 \times 10^{-5}$ |
> | Late | max over steps of maximum update | $1.943 \times 10^{-4}$ |
> | Late | # steps with max update $> 2\rho$ | $0$ |
>
> We hope these responses address your concerns and kindly ask you to reconsider your score.

---

> > ### Author Rebuttal · Reviewer_khej · 2026-04-03
> >
> > I thank the authors for their response. I will keep my score.

---

> > > ### Author Response · Authors · 2026-04-03
> > >
> > > Thank you very much for the careful reading and constructive feedback. We are very happy that our rebuttal addressed your concerns. Your comments substantially helped us improve the clarity and quality of the paper. We hope the reviewer will help us deliver this important message to this field.

---

### Official Review · Reviewer_zyUe · 2026-03-13

**Soundness:** 3
**Presentation:** 3
**Significance:** 2
**Originality:** 3
**Overall Recommendation:** 4
**Confidence:** 2

**Summary:**

This paper investigates the seemingly random nature of weight sign patterns in neural network compression, specifically targeting the storage bottleneck caused by sign bits in sub-bit regimes. The authors demonstrate that across MLPs, CNNs, and Transformers, sign matrices mimic i.i.d. Rademacher noise, making them stubbornly resistant to traditional structural compression despite remaining largely unchanged from their initial states during training. To explain this behavior, the paper introduces Sign Lock-In Theory, which characterizes sign flips as rare, decaying events driven by SGD dynamics. Building on this framework, the researchers propose gap initialization and outward-drift regularization as methods to stabilize these patterns, successfully preserving a compressible sign template throughout training without sacrificing model performance.

**Compliance With Llm Reviewing Policy:**

Affirmed.

**Final Justification:**

Although I am not an expert in this area and may not have fully captured all aspects of the paper in my original concerns, the authors’ detailed response has adequately addressed them. Therefore, I would like to increase my overall recommendation.

**Key Questions For Authors:**

See weaknesses

**Limitations:**

yes

**Strengths And Weaknesses:**

Strengths

1. Underexplored perspective on sub-bit compression with clear theoretical analysis.
The paper studies an often overlooked component of neural network parameters: the sign patterns of weights. It proposes the sign lock-in theory, which provides a theoretical explanation for why weight signs appear random yet remain stable during training. By modeling sign flips as boundary-crossing events and analyzing their stopping-time dynamics, the paper derives a geometric-tail bound on the number of effective sign flips.

2. Comprehensive experimental analysis of sign behavior.
The authors provide extensive empirical analysis across different architectures, including MLPs, CNNs, and Transformers. The experiments consistently show that weight signs exhibit noise-like spectral behavior while remaining highly persistent during training.

3. Clear and well-organized presentation.
The paper is well-written and structured in a logical manner. The motivation, methodology, and experimental results are presented clearly, making the overall framework easy to follow.

Weaknesses

1. Limited evaluation on modern large-scale models and tasks.
Although the paper presents experiments across several architectures, most evaluations are conducted on relatively small or controlled experimental setups. Additional validation on larger modern models and more realistic large-scale training datasets would strengthen the generality of the claims.

2. Practical compression benefits remain somewhat indirect and comparisons with extreme quantization methods are limited.
While the paper identifies the sign bottleneck and proposes techniques to stabilize sign structures, it does not present a complete end-to-end compression pipeline demonstrating clear improvements in storage efficiency. The connection between the proposed techniques and actual sub-bit compression gains could be clarified further. In addition, the compatibility of the proposed approach with existing extreme quantization or sub-bit compression methods is not fully explored, leaving its practical impact somewhat unclear.

---

> ### Author Rebuttal · Authors · 2026-03-30
>
> ## General comment
> Thank you for your thoughtful review. The main claim of our work is to discover the sign lock-in phenomenon and provide its theoretical explanation. End-to-end compression is therefore not the main claim of the paper, which is why it is presented in Appendix E.3, but we agree with the reviewer that it is difficult to find. We will improve it in the camera-ready version.
>
> ## W1 (limited evaluation on modern large-scale models and tasks).
>
> The current manuscript already contains large-scale evidence for the mechanism itself: **Fig. 4 and Appendix C.5 sweep scratch-trained Transformers from roughly $30$M to more than $10$B parameters, where both $\hat h$ and $\hat g$ decrease with scale.** Furthermore, the theoretical discussion in Appendix D.3, together with the empirical results in Figs. 13 and 14, already suggests from both theory and experiment that sign lock-in becomes stronger for larger batch sizes and wider models. In addition, we add two large-scale experiments to strengthen the paper.
>
> ### (1) Additional large-scale evidence for the sign lock-in phenomenon: 1B language model trained 300B tokens.
>
> We calculated the sign-flip histogram on the 1B-parameter Transformer model trained on 300B tokens.
> We found the same qualitative pattern as in our main experiments: the effective sign-flip count strongly follows a geometric tail. We will add additional large-scale experiments in the camera-ready version.
>
> ***Table (zyUe-1). Pythia-1B trained on 300B tokens: effective flip-count histogram and geometric-tail fit.*** *Empirical histogram of the effective outer-to-outer flip count $K_{\mathrm{eff}}$ and the corresponding zero-inflated geometric fit.*
>
> | $K_{\mathrm{eff}}$ | Count | Empirical prob. | Fitted prob. | Abs. error |
> |---:|---:|---:|---:|---:|
> | 0 | 168032 | 0.840160 | 0.840160 | 0.000000 |
> | 1 | 25405 | 0.127025 | 0.126960 | 0.000065 |
> | 2 | 5170 | 0.025850 | 0.026116 | 0.000266 |
> | 3 | 1120 | 0.005600 | 0.005372 | 0.000228 |
> | 4 | 230 | 0.001150 | 0.001105 | 0.000045 |
> | 5 | 37 | 0.000185 | 0.000227 | 0.000042 |
> | 6 | 5 | 0.000025 | 0.000047 | 0.000022 |
> | 7 | 1 | 0.000005 | 0.000010 | 0.000005 |
>
> ### (2) Additional large-scale validation of the sign-template method in Appendix E.3: ImageNet training.
>
> We applied the sign template method — the end-to-end compression pipeline introduced in Appendix E.3 — to training on ImageNet, a representative large-scale dataset. The results show that conventional compression methods suffer severe performance degradation as the bit rate approaches 1 bit per weight, due to the 1-bit wall effect, whereas the sign template method achieves substantial improvements in the 0.5–2.5 bpw range. Crucially, 1 bit is no longer a hard wall for our method.
>
> ***Table (zyUe-2). ImageNet Top-5 accuracy (%) of ResNet under compression via SVD with NormalFloat (NF) factor quantization.*** *Each weight matrix is decomposed as $W \approx U\Sigma V^{\top}$, and the factors $U, V$ are quantized with per-row NF quantization ($2^b$ levels at the quantiles of $\mathcal{N}(0,1)$). Best NF bit-width (NF2–NF8) is selected per budget. Bold indicates the best value per row.*
>
> | bpw | Sign Template (Ours) | Vanilla SVD (Baseline) | $\Delta$ |
> |---:|:---:|:---:|---:|
> | 0.50 | **5.3** | 0.5 | +4.8 |
> | 0.75 | **23.1** | 0.5 | +22.6 |
> | 1.00 | **42.0** | 0.5 | +41.5 |
> | 1.50 | **65.6** | 12.1 | +53.5 |
> | 2.00 | **74.7** | 65.1 | +9.6 |
> | 2.50 | **78.1** | 76.3 | +1.8 |
> | 3.00 | 79.8 | **80.9** | −1.1 |
> | 4.00 | 81.3 | **83.8** | −2.5 |
>
> ## W2 (practical compression benefits remain indirect).
>
> Thank you for raising this point. **Appendix E.3 already presents a complete end-to-end compression pipeline** with explicit bit accounting, matched-bit comparisons, and comparisons against multiple compression baselines. Thus, the practical compression benefit is already part of the current manuscript, although we agree that it should have been made much more visible in the main text. More broadly, our method is intended to be complementary rather than a head-on replacement for existing extreme quantization or low-bit methods. The key idea is to make sign information approximately free, so that the available bit budget can be spent on the magnitude component, which is much more compressible. This differs from standard binary / low-bit methods, which typically still store learned sign or ternary states explicitly together with auxiliary parameters. In this sense, our method is a sign-side prior / control mechanism that can be combined with magnitude-side compressors, rather than a competing alternative to all low-bit methods. Since sign lock-in may also arise in binary-QAT-style settings, our findings may provide ideas for improved optimization strategies on top of such methods.
>
> We hope these responses address your concerns and kindly ask you to reconsider your score.

---

> > ### Author Rebuttal · Reviewer_zyUe · 2026-04-02
> >
> > Thanks for the replies. All my concerns are addressed, and I will update my score.

---

> > > ### Author Response · Authors · 2026-04-03
> > >
> > > Thank you for the constructive and important review! Our work has been much polished.

---

### Official Review · Reviewer_hAUL · 2026-03-22

**Soundness:** 3
**Presentation:** 3
**Significance:** 3
**Originality:** 4
**Overall Recommendation:** 4
**Confidence:** 3

**Summary:**

This paper investigates the difficulty of compressing neural network weight signs below 1-bit per parameter. The authors identify a phenomenon termed sign lock-in, where learned weight signs closely resemble their random initial states and resist low-rank approximation. They observe experimentally that weight signs rarely flip during training because trajectories infrequently cross the zero boundary. To model these dynamics during stochastic gradient descent, the study develops a mathematical model using stopping-time analysis. The theory suggests that the number of effective sign flips follows a geometric tail distribution. Based on these insights, the paper proposes a gap-based initialization and an outer-drift regularizer. These interventions aim to preserve a highly compressible, low-rank sign template throughout training. Reductions in the sign flip rate were observed during experiments, and the compressibility of the sign matrices was maintained with minimal degradation in task performance.

**Compliance With Llm Reviewing Policy:**

Affirmed.

**Final Justification:**

The rebuttal addressed my main concerns regarding the bounded-update assumption, scaling to larger models, and expressivity preservation. One point remains partially addressed: the applicability of the proposed method to existing pretrained models (as opposed to from-scratch training) is still unclear. I encourage the authors to clarify this in the camera-ready version. The rebuttal reinforced my prior assessment. I maintain my score of Weak Accept (4).

**Key Questions For Authors:**

1.  Could the authors clarify how the bounded-update assumption behaves empirically when applied to optimizers with dynamic preconditioners like AdamW throughout various training phases?
2.  It is unclear if the proposed sign template method scales effectively to large vocabulary language models or standard large-scale benchmarks like ImageNet. Could the authors provide evidence or intuition regarding this scaling?
3.  Could the authors provide more intuition on how the low-rank template rank $r$ interacts with the overall expressivity of the network when scaling up to larger architectures?

**Limitations:**

The authors have adequately discussed several technical limitations within the main text, such as the potential to artificially induce a floating mode and the current focus on simple enforcement methods. However, the discussion of downstream risks and real-world consequences is insufficient. The authors could improve this section by offering a genuine reflection on how extreme model compression could affect deployment scenarios. Constructive additions could include discussing the environmental benefits of smaller models or the potential risks of deploying highly compressed models in sensitive applications where boundary-case performance might degrade.

**Strengths And Weaknesses:**

**Strengths:**

1. The identification of the sign lock-in phenomenon provides a convincing explanation for the one-bit wall in model compression.
2. A rigorous stopping-time analysis is developed to effectively model the rare boundary-crossing events.
3. The proposed interventions are directly motivated by the theory and offer a practical method to maintain compressible sign templates.
4. The empirical validation spans multiple architectures and scales up to billion-parameter models, which suggests broad relevance of the findings.

**Weaknesses:**

1. The theoretical bounds rely on strict bounded-update assumptions that might not perfectly reflect the empirical behavior of all adaptive optimizers without careful tuning.
2. The empirical evaluation of the sub-bit compression method is primarily demonstrated on relatively small-scale datasets, such as CharLM and DBPedia14.
3. An additional hyperparameter is introduced by the outer-drift regularizer, which requires careful balancing to avoid degrading the validation loss.
4. The figure on page 8 is mistakenly labeled as "Figure 7". Since it directly follows Figure 5 and is referred to as Figure 6 in the text, the caption should be updated accordingly.

---

> ### Author Rebuttal · Authors · 2026-03-30
>
> ## General comment
> Thank you for the careful and supportive review. We have conducted extensive additional experiments to address all of the reviewer's concerns.
>
> ## W1&Q1 (AdamW / bounded updates).
>
> We would like to clarify that our theory does not assume a strict almost-sure bounded-update condition for optimizers. Assumption 3.3 is explicitly formulated as a **high-probability bounded-update condition, with failure probability $\delta_{\mathrm{upd}}$**. Thus, the analysis is designed to reflect practical training behavior, where occasional atypical updates may occur, rather than requiring perfectly bounded dynamics at every step. We further report typical empirical measurements of the update magnitude in our **W5&Q2 response to Reviewer khej**. In our experiments, the update magnitudes are well-concentrated and remain far below the threshold $2\rho$, and we did not observe events that would need to be absorbed into the failure probability $\delta_{\mathrm{upd}}$. Regarding AdamW, the main theorem itself is optimizer-agnostic: Theorem 3.7 only needs Assumptions 3.3–3.4, while Proposition 3.5 is presented as an SGD-style sufficient condition. As noted in Remark 3.6, the sign lock-in theorem relies solely on these two assumptions, so any optimizer that satisfies comparable bounded-update and re-entry conditions arrives at the same conclusion. Furthermore, Proposition 3.5 can be extended to optimizers with a preconditioner by developing an analogous argument, as outlined in Remark D.9, confirming that AdamW also falls within the scope of the sign lock-in theory. **Finally, we would also like to emphasize that our experiments were conducted using AdamW, as described in Appendix C.**
>
> ## W2&Q2 (Scaling to larger models / benchmarks).
>
> We agree that end-to-end compression on ImageNet-scale would further strengthen the paper. The theoretical discussion in Appendix D.3 suggests from both theory and experiment that sign lock-in becomes stronger for larger batch sizes and wider models. In addition, we have now conducted two additional experiments: (1) the flip histogram for Pythia models, and (2) an **ImageNet** training run with end-to-end sign-template compression, reported in the **W1 response to Reviewer zyUe**.
>
> ## W3&Q3 (Expressivity in larger models).
>
> To directly test how a fixed rank-$r$ template interacts with expressivity at scale, we trained Transformer language models from 0.3M to 512M parameters, all sharing the same rank  ($r=2$) sign template.
>
> ### (1) Expressivity is empirically preserved.
>
> As shown in **Table (hAUL-1)**, the best-validation-PPL overhead of gap+reg decreases with model size and becomes negligible, empirically. This is because the rank-$r$ template constrains only the sign pattern while leaving all $mn$ magnitude parameters per weight matrix free; the number of continuous degrees of freedom is therefore unchanged.
>
> ***Table (hAUL-1). Best validation PPL on Transformer next-token prediction.***
>
> | #Params | Vanilla | Gap+Reg | Degradation PPL |
> |---:|---:|---:|---:|
> | 0.3M  | 5.22 | 5.61 | +0.39 |
> | 1.9M | 4.55 | 4.69 | +0.14 |
> | 4.8M | 4.67 | 4.64 | **−0.03** |
> | 10.8M  | 4.67 | 4.63 | **−0.04** |
> | 25.4M  | 4.70 | 4.69 | **−0.01** |
> | 59.3M  | 4.82 | 4.78 | **−0.04** |
> | 99.5M  | 4.87 | 4.78 | **−0.09** |
> | 512M  | 4.93 | 4.94 | +0.01 |
>
> ### (2) Regularization effect improves generalization performance.
>
> **Table (hAUL-2)** further shows that the template acts as a strong implicit regularizer. Vanilla training overfits severely at scale (e.g., PPL degrades by +13.22 at 99.5M), whereas gap+reg limits the degradation to +1.01. At 512M the contrast persists: +2.99 vs. +0.06. In summary, the low-rank sign template does not appear to limit network capacity in practice; it steers optimization toward better-generalizing solutions.
>
> ***Table (hAUL-2). Overfitting: best vs. final PPL on Transformer next-token prediction.***
>
> | #Params | Vanilla Best | Vanilla Final | Vanilla Overfit | Gap+Reg Best | Gap+Reg Final | Gap+Reg Overfit |
> |---:|---:|---:|---:|---:|---:|---:|
> | 0.3M | 5.22 | 5.22 | +0.00 | 5.61 | 5.56 | −0.05 |
> | 1.9M | 4.55 | 4.69 | +0.14 | 4.69 | 4.70 | +0.01 |
> | 4.8M | 4.67 | 5.96 | **+1.29** | 4.64 | 4.67 | **+0.03** |
> | 10.8M | 4.67 | 9.42 | **+4.75** | 4.63 | 4.94 | **+0.31** |
> | 25.4M | 4.70 | 12.58 | **+7.88** | 4.69 | 5.76 | **+1.07** |
> | 59.3M | 4.82 | 15.36 | **+10.54** | 4.78 | 5.79 | **+1.01** |
> | 99.5M | 4.87 | 18.09 | **+13.22** | 4.78 | 5.79 | **+1.01** |
> | 512M | 4.93 | 7.92 | **+2.99** | 4.94 | 5.00 | **+0.06** |
>
> ## W4 (Minor).
>
> The figure immediately after Fig. 5 is intended to be Fig. 6. We will fix this.
>
> ## Limitations / broader impact.
>
> Thank you. Sub-bit compression reduces energy and storage costs, but aggressive compression risks degrading boundary-case performance in safety-critical settings. We will expand this discussion in the camera-ready version.
>
> We hope these responses address your concerns and kindly ask you to reconsider your score.

---

> > ### Author Rebuttal · Reviewer_hAUL · 2026-04-02
> >
> > Thank you for the detailed rebuttal. All weaknesses and key questions have been adequately addressed. I have one follow-up question:
> >
> > The paper's empirical discovery analyzes pretrained models (e.g., TinyLlama, ResNet18, MLP-Mixer) and observes that learned signs are random-like yet persist from initialization. The proposed method, however, requires **from-scratch training** with a low-rank sign template, it does not apply to existing pretrained models via QAT-style fine-tuning.
> >
> > This distinction is important but not explicitly stated in the paper. Could the authors clarify this limitation in the camera-ready version? Specifically:
> >
> > - The method applies to models trained from scratch, not to existing pretrained models.
> > - Applying the sign template to pretrained models would require re-initializing signs, which destroys learned structure.
> >
> > This clarification would help readers understand the scope and applicability of the proposed approach.

---

> > > ### Author Response · Authors · 2026-04-02
> > >
> > > Thank you for your constructive suggestions. The main contributions of our work are the discovery of the sign lock-in phenomenon and its theoretical elucidation. In addition, we proposed the sign template method as a practical way to utilize sign lock-in phenomenon.
> > >
> > >  - Your understanding of the sign template method is correct. This method is designed for training from scratch. In practice, the pipeline can be further lightweighted by incorporating techniques such as distillation. Indeed, Figure 21(b), (d), and (f) present distilled versions of Figure 21(a), (c), and (e). However, we regard these results as still falling within the from-scratch training.
> > >
> > > - We agree with the reviewer’s suggestion, and we will add the limitations proposed by the reviewer in the camera-ready version.
> > >
> > > We sincerely appreciate this important suggestion. In response to the reviewer’s feedback, we conducted a large-scale additional experiment, which substantially improved the quality of this work. We would be very grateful if you could reconsider the score upward so that this study can better contribute to the field.

---

### Decision · Program_Chairs · 2026-04-30

**Decision:**

Accept (regular)

**Comment:**

This paper identifies and studies an interesting bottleneck in sub-bit compression: once magnitudes are aggressively compressed, the sign bit becomes a dominant and difficult-to-compress cost. The paper shows empirically that learned sign matrices are both random-like and highly persistent from initialization, develops a stopping-time theory of rare sign flips that explains this behavior, and proposes a practical from-scratch training method based on compressible sign templates and lock-in enhancement. Overall, I found the paper’s central idea clear, original, and technically meaningful.

The main concerns raised in review were about practical scope: limited large-scale end-to-end validation, the indirect connection to full compression pipelines, and the fact that the proposed sign-template method applies to training from scratch rather than to arbitrary pretrained models. These are legitimate limitations. However, the rebuttal appears to have addressed most of the concerns with additional large-scale evidence and clearer scoping, and I do not view the remaining issue as fatal. The paper’s main claim is the discovery and explanation of sign lock-in, and on that claim the submission is convincing.

Overall, I believe this paper makes a worthwhile contribution. It explains a previously underappreciated obstacle in sub-bit compression and offers both a theory and a concrete intervention that others can build on. I therefore recommend accept, with the expectation that the final version will state more explicitly that the proposed sign-template method is a from-scratch approach and not a direct post-training method for existing pretrained models.